# Let Me Explain, Again:
# Multiplicity in Local Sufficient Explanations

**Ryan Pilgrim**[*]                                         *pilgrimr@seas.upenn.edu*
*Innovation in Data Engineering and Science (IDEAS)*
*University of Pennsylvania*

**Beepul Bharti**                                          *bbharti1@jhu.edu*
*Mathematical Institute for Data Science (MINDS) and Data Science and AI Institute (DSAI)*
*Johns Hopkins University*

**Kyle Poe**                                               *kpoe@sas.upenn.edu*
*Innovation in Data Engineering and Science (IDEAS)*
*University of Pennsylvania*

**René Vidal**                                             *vidalr@seas.upenn.edu*
*Innovation in Data Engineering and Science (IDEAS)*
*University of Pennsylvania*

**Jeremias Sulam**                                         *jsulam1@jhu.edu*
*Mathematical Institute for Data Science (MINDS)*
*Johns Hopkins University*

**Reviewed on OpenReview:** *https://openreview.net/forum?id=d6FMg4hozX*

## Abstract

When asked to explain their decisions, humans can produce multiple complementary justifications. In contrast, several feature attribution methods for machine learning produce only one such attribution, despite the existence of multiple equally strong and succinct explanations. The explanations found by these methods thus offer an incomplete picture of model behavior. In this paper, we study the problem of explaining a machine learning model's prediction on a given input from the perspective of minimal feature subsets that are sufficient for the model's prediction, focusing on their non-uniqueness. We give a tour of perspectives on this non-uniqueness, in terms Boolean logic, conditional independence, approximate sufficiency, and degenerate conditional feature distributions. To cope with the multiplicity of these explanations, we propose a wrapper methodology that can adapt and extend methods that find a single explanation into methods for finding multiple explanations of similar quality. Our experiments benchmark the proposed meta-algorithm, which we call Let Me Explain Again (LMEA), against two multi-explanation method baselines on synthetic and real-world multiple-instance learning problems for image classification and demonstrate the ability of LMEA to augment two single-explanation methods.

## 1 Introduction

Predictive machine learning (ML) models are increasingly used in high-stakes decision-making contexts such as healthcare (Shailaja et al., 2018), employment (Freire & de Castro, 2021), credit scoring (Thomas et al., 2017), and criminal justice (Rudin et al., 2020). As the consequences of their use grow, so does the importance of understanding the decision processes behind model predictions. However, a full understanding of complex

---

[*]Work performed while at Johns Hopkins University.

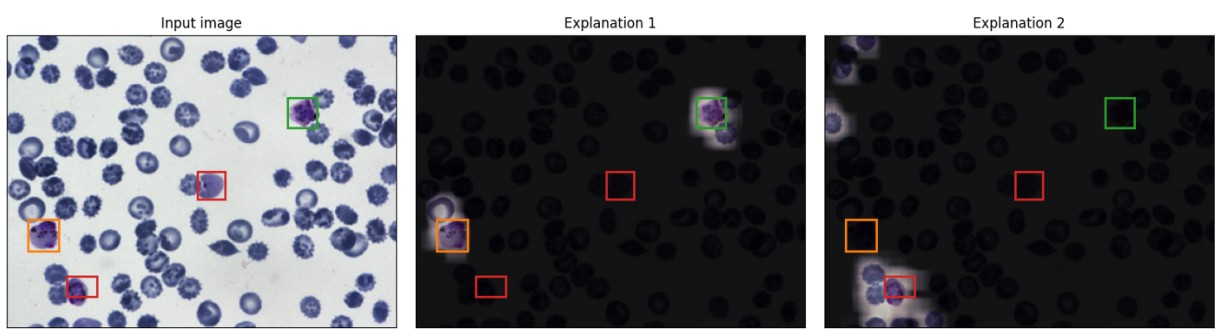

Figure 1: Multiple MSSs are common in the multiple instance learning (MIL) (Dieterich et al., 1997) setting. Pictured are example explanations for a classifier trained to predict the presence of certain malaria-infected cells (trophozoites) in images of blood smears from the BBBC041 dataset (Ljosa et al., 2012). The green box depicts the location of a trophozoite, while the red boxes depict the location of other kinds of infected cells. The orange box is labeled as "difficult," meaning it could not be classified by a human expert. *Left:* input image to be explained, for which the classifier predicts the presence of a trophozoite. *Middle:* an explanation recovered by a method for finding a single MSS. *Right:* an explanation found by running the same single-MSS method again using our proposed wrapper algorithm. Crucially, although both explanations are sufficient, the single-MSS approach misses the model's dependence on non-trophozoite cells, which our method reveals.

models, such as deep neural networks, remains elusive: these models have been, and continue to be, widely regarded as "black boxes" (Alain & Bengio, 2016, p. 1).

These concerns have spurred the development of local feature attribution methods, which aim to explain the prediction of an ML model on a specific input by identifying the most "important" features. Among the many diverse methods, an increasingly popular family of approaches focuses on identifying a minimal subset of features that, once fixed, are sufficient to determine the prediction (Chockler et al., 2021; 2024; Shih et al., 2018; Ignatiev et al., 2020; Darwiche & Hirth, 2020; Ribeiro et al., 2018; Wang et al., 2021; Amoukou & Brunel, 2022; Fong & Vedaldi, 2017; Fong et al., 2019; Luss & Dhurandhar, 2024; Chockler et al., 2025). Formally, these methods aim to explain the prediction of a model $f : \mathcal{X} \to \mathcal{Y}$ on a fixed input $\mathbf{x} \in \mathcal{X} \subsetneq \mathbb{R}^d$ by identifying a minimal subset $\mathcal{S} \subseteq [d]$ such that fixing $\mathbf{x}_{\mathcal{S}}$ suffices to determine the model output $f(\mathbf{x})$ either exactly or approximately, and either deterministically or with high probability.

In this work, we study this class of explanations, which we refer to as *minimal sufficient subsets* (MSS). Although MSSs have been formally defined in various ways and their properties have been studied to different extents, a key aspect that has received little attention is their *multiplicity*. Empirically, MSSs have been shown to be not unique (Carter et al., 2019; Camburu et al., 2020), but the underlying theoretical reasons for this phenomenon remain largely unexplored. Even more concerning is that, despite this fact, several methods that find (or can be easily adapted to find) MSSs (Fong & Vedaldi, 2017; Fong et al., 2019; MacDonald et al., 2019; Brinner & Zarrieß, 2023; Luss & Dhurandhar, 2024) return only one such subset.[1]

To understand why identifying only a single MSS can be problematic, consider a common downstream application of these methods: model debugging. *Post-hoc* explanations can be used to identify model dependence on semantically meaningless or irrelevant features (Carter et al., 2021; Wäldchen, 2022). In this context, methods that provide only one MSS will not necessarily identify the missing spurious features that influence a model's prediction (Chockler et al., 2025). This situation is depicted in Figure 1, which involves explaining a classifier for detecting infected cells (trophozoites) in blood smear images from the BBBC041 dataset (Ljosa et al., 2012). Seeing just the first explanation might lead the user to conclude that the non-trophozoite cells do not influence the classifier's prediction when, in fact, they do.

---

[1]A notable exception to these statements is the growing set of methods based on logic (Ignatiev & Marques-Silva, 2021; Huang et al., 2021; Izza et al., 2020), for which the reasons behind MSS multiplicity are well understood. Enumeration of MSSs is possible for these methods in certain scenarios (Ignatiev et al., 2020); however, the problems these methods solve are often computationally hard, and even the most recent methods scale poorly to neural networks with high-dimensional inputs (Marques-Silva, 2023; Bassan et al., 2025).

The multiplicity of MSSs raises important questions. Are there settings when MSSs are unique, or is this multiplicity inevitable (and when)? Furthermore, when multiple MSSs do exist, can we properly identify them all? In this work, we provide answers to these questions. To address the multiplicity of explanations, we propose a meta-algorithm that extends methods which find a single (or a few) MSSs, making them capable of systematically identifying multiple, different MSSs. In the ideal case where we have access to an oracle (that returns a single MSS), our algorithm provably recovers a subset of MSSs that intersects every MSS. In turn, our experiments demonstrate how augmenting existing methods that identify a single MSSs leads to more comprehensive explanations in real and synthetic computer vision settings.

## 1.1 Related work

MSSs have been formalized in diverse ways, drawing on ideas from probability (Amoukou & Brunel, 2022; Wang et al., 2021; Bharti et al., 2025), causality (Chockler et al., 2025), and logic (Ignatiev et al., 2019; 2020; Marques-Silva et al., 2020; Izza et al., 2020; Darwiche & Ji, 2022; Darwiche & Hirth, 2020). Across these formulations, it is often acknowledged that MSSs are generally non-unique. However, aside from works in logic-based explainability, few address this fact as a central concern. Among those that do, approaches can be categorized along several axes; namely, whether they (a) assume black-box (Carter et al., 2019; Chockler et al., 2025) or gradient (Carter et al., 2021; Byra & Skibbe, 2025) access to the model, (b) identify disjoint (Carter et al., 2019; 2021) or overlapping (Shitole et al., 2021; Chockler et al., 2025; Byra & Skibbe, 2025) sets of MSSs and (c) possess (Carter et al., 2019) or lack (Chockler et al., 2025; Shitole et al., 2021; Amoukou & Brunel, 2022) theoretical guarantees on the subset of MSSs found.

Works related to ours are those by Shitole et al. (2021), Chockler et al. (2025), Amoukou & Brunel (2022), MacDonald et al. (2019), Fong & Vedaldi (2017), and Fong et al. (2019). The method of Shitole et al. (2021) and MultiReX (Chockler et al., 2025) are two methods that can identify overlapping (or disjoint) sets of MSSs. The method proposed by Shitole et al. (2021) uses a forward beam search on a $7 \times 7$ image grid, and limit the overlap of the final MSS set by doing a greedy backward elimination on a submodular set intersection objective. MultiReX uses principles from actual causality (Halpern, 2019). While empirically effective, the MultiReX approach lacks theoretical guarantees about the subset of MSSs it recovers. Amoukou & Brunel (2022) propose a nonparametric algorithm for finding MSSs via a Random Forest (Breiman, 2001) and propose combining multiple explanations into a single attribution by exploring a subset of all MSSs. MacDonald et al. (2019) propose a method for finding minimal subsets of features that account for the prediction in a rate-distortion theoretic framework. Loosely speaking, their method finds a small set of features that approximately preserves the prediction when replacing the complement with Gaussian noise. Similarly, Fong & Vedaldi (2017); Fong et al. (2019) propose methods to find small, contiguous sets of features that maintain the prediction when the complement is replaced by background (e.g., zero or blurred input) values. Both (MacDonald et al., 2019) and (Fong et al., 2019) find a *single* MSS; we will show how each of these methods can be extended to find *multiple* MSSs in our experiments.

The works most relevant to ours are those of Bharti et al. (2025), Carter et al. (2019; 2021), and Byra & Skibbe (2025). Bharti et al. (2025) propose definitions for sufficiency and necessity of ML model explanations in the context of feature removal (Covert et al., 2021), connect these definitions to conditional independence and Shapley value explanations (Lundberg & Lee, 2017), and demonstrate via theory and experiments how augmenting the trade off between necessity and sufficiency leads to equally informative, yet different, explanations. In this work, we adopt a variation on their sufficiency definition. Carter et al. (2019), propose a backward-forward greedy algorithm that sequentially identifies multiple MSSs and demonstrate its effectiveness on MNIST digit classification, sentiment analysis, and genomics tasks. To improve the scalability of this algorithm, the authors extend their method to image classifiers in (Carter et al., 2021) by incorporating input gradients to guide the backward elimination, and show its application to discovering models' usage of spurious feature patterns (e.g., background pixels) in their predictions. Byra & Skibbe (2025) take an approach similar to (Fong et al., 2019), implementing the method of (Fong et al., 2019) using implicit neural representations (INR) to recover multiple sufficient explanations. To identify multiple explanations, they repeatedly solve an MSS problem, retraining an implicit network while preventing re-selection of features via a penalty term in their optimization objective. This approach involves reimplementation of the XP solver and determination of a penalty strength for which a suitable value is not obvious *a priori*.

While many works have noted the existence of multiple MSSs, several theoretical and qualitative reasons for the non-uniqueness of MSSs have not been made explicit. Furthermore, most of these works study the problem from the perspective of logic, and other perspectives on MSS non-uniqueness are comparatively under-explored. For example, while Bharti et al. (2025) study MSSs in a probabilistic framework, they do not address the reasons behind the non-uniqueness of MSS explanations in this probabilistic context. On the algorithmic side, most papers on the topic outside of the logic community propose specialized techniques to find multiple MSSs (e.g., (Carter et al., 2019; 2021; Shitole et al., 2021; Chockler et al., 2025)), and it is still unclear to what extent methods for finding a single MSS might be extended to recover multiple MSSs. While Byra & Skibbe (2025) take a first step toward answering this question, they only extend a single method.

## 1.2 Contributions

Our work addresses each of the aforementioned gaps by providing both a unified theoretical discussion of the MSS non-uniqueness problem and proposing a general framework for extending existing MSS-based techniques that find a single explanation. Concretely, our contributions are the following:

1. We present several mathematical and qualitative perspectives on multiple MSSs, including interpretations in terms of logic, symmetry, CI relationships, non-linearity, and approximation. Along the way, we weave in intuition-building examples and unify logical and probabilistic viewpoints on the multi-MSS phenomenon. To the best of our knowledge, such a comprehensive treatment of the reasons behind the non-uniqueness of MSSs is lacking in the literature.

2. We propose a meta-algorithm, Let Me Explain Again (LMEA), that generalizes existing strategies for finding multiple MSSs (Carter et al., 2019; 2021; Byra & Skibbe, 2025) and can adapt explanation methods for finding a single MSS into methods for finding multiple. Given access to an oracle that finds a single MSS, we show that LMEA recovers a disjoint subset of MSSs whose union intersects every MSS.

3. We show how LMEA can be used to extend two gradient-based methods—rate distortion explanations (RDE) (MacDonald et al., 2019) and extremal perturbations (XP) (Fong et al., 2019)—to retrieve multiple MSSs. Our proposed method provides easy-to-set hyperparameters and allows the extension of certain gradient-based perturbative MSS explanation methods with minimal to no modification of their off-the-shelf implementations, enhancing the applicability of LMEA.

4. We showcase LMEA on a series of multiple instance learning (MIL) (Dietterich et al., 1997) image classification problems, comparing LMEA against three baselines for finding multiple MSSs: MultiReX (Chockler et al., 2025), sufficient input subsets (SIS) (Carter et al., 2019), and beam search (BS) (Shitole et al., 2021). Our results demonstrate the ability of LMEA to extend single-MSS explanation methods into multiple-MSS explanation methods. Furthermore, our results show that extending RDE and XP with LMEA retrieves explanations that are competitive with those found by MultiReX, SIS, and BS in terms of overlap with human-annotated salient image regions.

## 2 Setting and background

**Notation.** Let $f : \mathcal{X} \to \mathcal{Y}$ be a model to be explained, where $\mathcal{X} \subseteq \mathbb{R}^d$ denotes the input space, and $\mathcal{Y}$ denotes the output space, which depends on the task. For instance, $\mathcal{Y} \subseteq \mathbb{R}^k$ in the regression setting, while $\mathcal{Y} = \mathcal{P}(\mathcal{L})$ or $\mathcal{Y} = \mathcal{L}$ in the classification setting, where $\mathcal{L}$ is a set of labels and $\mathcal{P}(\mathcal{L})$ is the set of probability distributions over $\mathcal{L}$. Sets will be written in calligraphic script, e.g., $\mathcal{S}$, and set complements will be denoted with a superscript $c$, e.g., $\mathcal{S}^c$. We will write $[d] \doteq \{1, 2, \ldots, d\}$. Random scalars will be written in uppercase, e.g., $X$, while realizations will be written in lowercase, e.g., $x$. Matrices will also be written in un-bolded uppercase, e.g., $A$; context will distinguish them from random variables. Vectors will be bolded, with random vectors in boldface uppercase font, e.g., $\mathbf{X}$, and deterministic vectors and realizations of random vectors in boldface lowercase font, e.g., $\mathbf{x}$. Subvectors will be denoted with subscripts, e.g., $\mathbf{x}_{\mathcal{S}} = (x_i)_{i \in \mathcal{S}}$. We will write $A_{\mathcal{S}}$ to denote the submatrix of $A$ formed by the columns indexed by $\mathcal{S}$. Constant vectors and matrices of all zeros (resp. ones) will be written as $\mathbf{0}$ (resp. $\mathbf{1}$), with dimension determined by context. All random variables

will be understood to be defined over an underlying sample space $\Omega$ with joint probability measure $\mathbb{P}$. Given a random variable $Z$, we will denote by $\mathbb{P}_Z$ the marginal distribution of $Z$, and $p_Z$ its corresponding density. When clear from context, we will drop the density subscripts, e.g., $p(z)$ will be understood to mean $p_Z(z)$. Expectations will be denoted by $\mathbb{E}$. To avoid measure-theoretic issues, we will assume throughout that the distribution of $\mathbf{X}$ admits either a probability mass function (pmf) or probability density function (pdf), $p(\mathbf{x})$. Finally, for any measurable function $\varphi$, we will use the following notation for conditional expectations with respect to $p(\mathbf{x}'_{\mathcal{S}^c} \mid \mathbf{x}_{\mathcal{S}})$:

$$\mathop{\mathbb{E}}_{\mathbf{X}_{\mathcal{S}^c}|\mathbf{x}_{\mathcal{S}}} [\varphi(\mathbf{X}_{\mathcal{S}^c}, \mathbf{x}_{\mathcal{S}})] \doteq \int p(\mathbf{x}'_{\mathcal{S}^c} \mid \mathbf{x}_{\mathcal{S}}) \varphi(\mathbf{x}'_{\mathcal{S}^c}, \mathbf{x}_{\mathcal{S}}) \, \mathrm{d}\mathbf{x}'_{\mathcal{S}^c}. \tag{1}$$

For discrete $\mathbf{X}$, one may replace the integral with a sum.

**Problem Setting.** The explanation problem we study is as follows: given a trained model $f : \mathcal{X} \to \mathcal{Y}$ that maps inputs in $\mathcal{X}$ to outputs in $\mathcal{Y}$, we seek a *minimal sufficient* set $\mathcal{S} \subseteq [d]$ such that the values of $\mathbf{x}$ on $\mathcal{S}$ approximately determine the prediction $f(\mathbf{x})$, in a sense made formal by Definition 1. When $f$ is a classification model, we will usually take $\mathcal{Y}$ to be a space of distributions $\mathcal{P}(\mathcal{L})$ and $f_y(\mathbf{x})$ to approximate the conditional label probability $\mathbb{P}(Y = y \mid \mathbf{X} = \mathbf{x})$ for each $\mathbf{x} \in \mathcal{X}$ and $y \in \mathcal{L}$. When $f$ is a regression model, we will take $f$ to approximate the conditional expectation $\mathbb{E}[\mathbf{Y} \mid \mathbf{X} = \mathbf{x}]$. Throughout, we will consider a nonnegative function $D : \mathcal{Y} \times \mathcal{Y} \to \mathbb{R}$ with $D(\mathbf{y}, \mathbf{y}') = 0$ if and only if $\mathbf{y} = \mathbf{y}'$, but we do not require $D$ to be symmetric. Choices of $D$ can be, for instance, the squared $\ell_2$ distance, KL divergence, or TV distance. To ensure that conditional expectations are well-defined, we will assume that the input to be explained, $\mathbf{x}$, has positive probability mass, or density, under $\mathbb{P}$, i.e., $p(\mathbf{x}) > 0$.

## 3 Minimal sufficient subset explanations

We begin by introduction the definitions of minimality and sufficiency that are central to our work. For sufficiency, we consider following definition, which strengthens the definition of Bharti et al. (2025) and takes inspiration from similar definitions of sufficiency (Chattopadhyay et al., 2022) and distortion (MacDonald et al., 2019).

**Definition 1** ($\varepsilon$-Sufficiency). Fix the model $f$, input instance $\mathbf{x}$, and $D$. The nonempty subset $\mathcal{S} \subseteq [d]$ is $\varepsilon$-*sufficient for $f$ at $\mathbf{x}$ with respect to $D$ if*

$$\mathop{\mathbb{E}}_{\mathbf{X}_{\mathcal{S}^c}|\mathbf{x}_{\mathcal{S}}} [D(f(\mathbf{X}_{\mathcal{S}^c}, \mathbf{x}_{\mathcal{S}}), f(\mathbf{x}))] \leq \varepsilon,$$

with $\varepsilon \geq 0$. When $f$, $\mathbf{x}$, or $D$ are clear from context, we will omit the corresponding qualifiers.

By setting $D$ to be an indicator, $D(\mathbf{u}, \mathbf{v}) = \mathbb{I}(\mathbf{u} \neq \mathbf{v})$, and taking $f : \mathbb{R}^d \to \mathcal{L}$ to be a classifier that outputs a class label, Definition 1 generalizes same-decision probability (Wang et al., 2021).

When $\varepsilon = 0$, we recover the logical definition of sufficiency, i.e. that $f(\mathbf{x}_{\mathcal{S}}, \mathbf{X}_{\mathcal{S}^c}) \overset{\text{a.s.}}{=} f(\mathbf{x})$, where $\mathbf{X}_{\mathcal{S}^c} \sim \mathbb{P}_{\mathbf{X}_{\mathcal{S}^c}|\mathbf{X}_{\mathcal{S}} = \mathbf{x}_{\mathcal{S}}}$. When $\varepsilon > 0$, on the other hand, Definition 1 recovers relaxed versions of this notion, with the choice of $D$ playing a role in how these are formally quantified.

As in several prior works, we seek a *minimal* sufficient subset (MSS) that explains the model prediction. Thus we present the following definition, which adopts the notion of minimality used in, e.g., prime implicant (PI) explanations (Shih et al., 2018).

**Definition 2** (Minimal $\varepsilon$-Sufficient Subset). A subset $\mathcal{S} \subseteq [d]$ is a minimal $\varepsilon$-sufficient subset ($\varepsilon$-MSS) if $\mathcal{S}$ is $\varepsilon$-sufficient and there exists no $\mathcal{T} \subsetneq \mathcal{S}$ that is $\varepsilon$-sufficient.

When $\varepsilon = 0$, we refer to a 0-MSS simply as an MSS. In what follows, terms like "smallest" or "minimal" will refer to minimality with respect to set inclusion.

## 4 The ubiquity of multiple minimal sufficient subsets

The non-uniqueness of MSSs is a multifaceted phenomenon that has not been deeply explored. Here, we present several perspectives on non-uniqueness, addressing Boolean logic, application-specific considerations,

conditional independence (CI), nonlinearity, approximate sufficiency, and degenerate conditional feature distributions. Taken together, we present and detail a number of instructive examples that demonstrate unique MSSs are often the exception rather than the rule.

### 4.1 `OR` logic and multiple MSSs in vision problems

We begin with the following toy setup which gives strong intuition on why multiple MSSs exist; it is an example of the widely noted multiplicity of (probabilistic) prime implicants in logic-based explanations (Wäldchen, 2022, p. 51), (Darwiche & Hirth, 2020; Ignatiev et al., 2020).[2]

**Example 1** (`OR`). Let $X_1, X_2 \overset{\text{i.i.d.}}{\sim} \text{Ber}(1/2)$ and $Y = X_1 \vee X_2$, where $\vee$ is the `OR` operation. Then, for $\mathbf{x} = (1, 1)$, the sets $\mathcal{S} = \{1\}$ and $\mathcal{S}' = \{2\}$ are both MSSs for the model $f(\mathbf{x}) = \mathbb{P}(Y = 1 \mid \mathbf{X} = \mathbf{x})$.

This `OR` logic, and generalizations thereof, are ubiquitous in machine learning. One very common area where the `OR` logic appears is in image classification, when there are multiple objects of a particular target class in an image, resulting in potentially multiple sufficient subsets. For example, consider the problem of malaria detection via images of human blood smears (Ljosa et al., 2012), where the goal is to classify the entire image as "infected" or "not infected". A single infected cell suffices to determine the prediction, although multiple may be present. Symmetries also easily give rise to multiple MSSs. If we consider classifying images based on the presence or absence of earrings in portraits, as we will do in Section 6, there will often be at least two earrings present, corresponding to the bilateral symmetry of the face. Seeing one (earring) should be all that a model needs to detect the presence of at least one (earring). A classifier trained to optimality on these tasks will therefore exhibit the type of `OR` logic outlined in Example 1, and will thus possess multiple MSSs on certain inputs. While each of these sufficient explanations may not be strictly minimal, practically, we find that methods to find MSSs will often recover each of them. These examples demonstrate that the `OR` is not merely a special or contrived case for the existence of multiple MSSs; this pattern, and therefore multiple MSSs, can be found in many real-world scenarios.

### 4.2 The role of context-specific independence

Example 1 is intuitive and relates logical notions of sufficiency to our probabilistic Definition 1, but the degenerate conditional distribution $p(y \mid \mathbf{x})$ yields a deterministic relationship between the features $\mathbf{X}$ and the label $Y$, which does not capture the more general situation where $Y$ is only partially determined by $\mathbf{X}$. Here we present a more general, but related, reason for non-uniqueness of MSSs under our probabilistic definition of sufficiency by connecting MSSs to CI relationships.

At a high level, CI relationships allow us to reason about the dependencies between random variables. Intuitively, two random variables $A$ and $B$ are conditionally independent given $C$, (mathematically written as $A \perp\!\!\!\perp B \mid C$) if $A$ provides no additional information about $B$ once we know $C$. More formally, $A \perp\!\!\!\perp B \mid C$ if $p(a \mid b, c) = p(a \mid c)$ for all $a, b, c$ with $p(b, c) > 0$ (assuming the necessary densities/pmfs and conditional probabilities are well defined).

CI relations are a useful way to rigorously characterize MSSs via probabilistic statements. For example, for the model $f$ that is the Bayes-optimal classifier, i.e., $f_y(\mathbf{x}) = \mathbb{P}(Y = y \mid \mathbf{X} = \mathbf{x})$, a subset $\mathcal{S}$ is an MSS if it is the minimal set satisfying the following CI relation (see Appendix A for proof):

$$\mathbf{X}_{\mathcal{S}^c} \perp\!\!\!\perp \mathbf{Y} \mid \mathbf{X}_{\mathcal{S}} = \mathbf{x}_{\mathcal{S}}. \tag{2}$$

In other words, $\mathcal{S}$ is the smallest set such that the remaining features $\mathbf{X}_{\mathcal{S}^c}$ are CI of $\mathbf{Y}$ given the observed features $\mathbf{x}_{\mathcal{S}}$. Note that a similar statement can be made for arbitrary models $f$ (Teneggi et al., 2023); however, we stick to the Bayes-optimal model for ease of exposition.

Due to the dependence on the specific values $\mathbf{x}_{\mathcal{S}}$ taken by $\mathbf{X}_{\mathcal{S}}$ (the *context*) in the conditioning event $\mathbf{X}_{\mathcal{S}} = \mathbf{x}_{\mathcal{S}}$, this CI relation in (2) is said to be *context-specific* (Boutilier et al., 1996). In characterizing an MSS this way, we can draw connections to the well-studied *Markov blankets* of $\mathbf{Y}$ (Pearl, 1988, p. 97), which

---

[2]This example is also similar to (Carter et al., 2019, Example 2), albeit using a different definition of sufficiency.

are subsets $\mathcal{S}$ such that

$$\mathbf{X}_{\mathcal{S}^c} \perp\!\!\!\perp \mathbf{Y} \mid \mathbf{X}_{\mathcal{S}}. \tag{3}$$

The smallest such subset $\mathcal{S}$ is called the *Markov boundary* of $\mathbf{Y}$ (Pearl, 1988, p. 97). One might hope that standard conditions for Markov boundary uniqueness also imply the uniqueness of their context-specific counterpart (i.e., MSSs), thereby eliminating the issue of multiple explanations. For example, if the joint distribution $p(\mathbf{x}, \mathbf{y})$ satisfies strict positivity, i.e., $p(\mathbf{x}, \mathbf{y}) > 0$ for all values of $\mathbf{x}$ and $\mathbf{y}$, then the Markov boundary of $\mathbf{Y}$ is unique (Pearl, 1988, Theorem 1, Theorem 4). We illustrate that such conditions unfortunately do not carry over to the context-specific independence represented by MSSs and Equation 2.

Mathematically, the `OR` example (Example 1) states that either $x_1 = 1$ or $x_2 = 1$ is sufficient because

$$\mathbb{P}(Y = 1 \mid \mathbf{X} = (1, 1)) = \mathbb{P}(Y = 1 \mid X_1 = 1) = \mathbb{P}(Y = 1 \mid X_2 = 1) = 1.$$

Now, since $\mathbb{P}(Y = 0 \mid \mathbf{X} = (1, 1)) = 0$, strict positivity of $p(\mathbf{x}, y)$ is violated. Revisiting the earlier discussion of strict positivity implying a unique Markov boundary, one might wonder whether "fixing" this degeneracy leads to a unique solution. Unfortunately, this is not the case. We demonstrate that the following example, presented in (Klein & Shimony, 2004) as an illustrative example to understand context-specific CI relationships, also serves as a great example of why strict positivity of $p(\mathbf{x}, y)$ does *not* imply the existence of unique MSSs.

**Example 2** (Leaky `OR`). *Let* $X_1, X_2 \overset{\text{i.i.d.}}{\sim} \text{Ber}(1/2)$, $N \sim \text{Ber}(\delta)$ *with* $0 < \delta < 1/2$, *and* $Y = (X_1 \vee X_2) \oplus N$, *where* $\oplus$ *denotes* `XOR`. *Then, for* $\mathbf{x} = (1, 1)$ *and* $f(\mathbf{x}) = \mathbb{P}(Y = 1 \mid \mathbf{X} = \mathbf{x})$, *the sets* $\mathcal{S} = \{1\}$ *and* $\mathcal{S}' = \{2\}$ *are both MSSs.*

Here, $p(\mathbf{x}, y) = p(\mathbf{x})p(y \mid \mathbf{x}) \geq \delta/4 > 0$ for all $(\mathbf{x}, y) \in \mathcal{X} \times \mathcal{Y}$, so strict positivity is satisfied. However,

$$\mathbb{P}(Y = 1 \mid \mathbf{X} = (1, 1)) = \mathbb{P}(Y = 1 \mid X_1 = 1) = \mathbb{P}(Y = 1 \mid X_2 = 1) = 1 - \delta,$$

so that $X_1 = 1$ and $X_2 = 1$ each determine $Y$ up to the noise (Klein & Shimony, 2004).[3] In conclusion, strict positivity of $p(\mathbf{x}, y)$ does not imply uniqueness for MSSs as it does for Markov Blankets.

### 4.3 Nonlinearity and approximate sufficiency

So far, Example 1 and Example 2 showcase that the existence of multiple MSSs can be a result of the relationship between $\mathbf{X}$ and $Y$. This is true more generally: the existence of multiple MSSs typically depends on the interplay between the feature distribution $p(\mathbf{x})$ and the model $f$. However, in some settings, the existence of multiple MSSs depends on one of these independently of the other. In this subsection and the next, we present results that are novel, to the best of our knowledge, and that elaborate upon this point.

We begin by illustrating that a unique MSS can exist under an idealized setting. In particular, the following result shows that for generalized linear models with certain nonlinearities, the MSS is unique *regardless* of the underlying feature distribution $p(\mathbf{x})$.

**Proposition 1** (Generalized linear models have unique MSSs). *Suppose that* $\mathbf{X}$ *has a continuous and strictly positive density, i.e.,* $p(\mathbf{x}) > 0$ *for all* $\mathbf{x} \in \mathcal{X} = \mathbb{R}^d$, *and let* $D$ *be continuous. Let* $f(\mathbf{x}) = g(A\mathbf{x} + \mathbf{b})$, *where* $A \in \mathbb{R}^{m \times d}$, $\mathbf{b} \in \mathbb{R}^m$, *and* $g : \mathbb{R}^m \to \mathbb{R}^k$ *for some* $m$. *If* $g$ *is continuous and injective on* $\text{span}(A) + \mathbf{b}$, *where* $\text{span}(A)$ *is the column span of* $A$, *then for all* $\mathbf{x} \in \mathcal{X}$ *the MSS for* $f$ *at* $\mathbf{x}$ *with respect to* $D$ *is unique and equals the nonzero column indices of* $A$. *Furthermore, if* $g = \text{softmax}$, *then for all* $\mathbf{x} \in \mathcal{X}$, *the MSS is also unique and equals* $\{i \in [d] : \forall c \in \mathbb{R}, \mathbf{a}_i \neq c\mathbf{1}\}$, *where* $\mathbf{1}$ *is the constant vector of all ones and* $\mathbf{a}_i$ *is the* $i^{th}$ *column of* $A$.

The proof is deferred to Appendix B. We remark that Proposition 1 encompasses linear regression models and linear classifiers (both binary and multi-class), since the logistic function is injective. Note that Proposition 1

---

[3]This argument further reveals that the weaker intersection property of CI (Pearl, 1988, Theorem 1), which also suffices to guarantee a unique Markov boundary (Pearl, 1988, Theorem 4), fails to guarantee a unique MSS, since the intersection property follows from strict positivity (Drton et al., 2009, Proposition 3.1.3). The same reasoning shows that the intersection property of context-specific independence (Corander et al., 2019) also cannot guarantee a unique MSS.

stands in contrast with logic-based explanations, for which linear models may possess many MSSs (Marques-Silva et al., 2020). This difference, however, relies on the assumptions of Proposition 1. If we relax these by allowing approximate ($\varepsilon > 0$) sufficiency or degenerate conditional distributions, then even linear models can have multiple MSSs, as Examples 3 and 4 demonstrate.

**Example 3.** Let $\mathbf{X} \sim \mathcal{N}(\mathbf{0}, I)$, $\mathbf{X} \in \mathbb{R}^3$, and let $f(\mathbf{x}) = A\mathbf{x}$, where $A \in \mathbb{R}^{3 \times 3}$ has normalized columns $\mathbf{a}_i$, $i \in [3]$. Write $A_{\mathcal{S}}$ to denote the submatrix of $A$ formed by the columns indexed by $\mathcal{S}$. Setting $D$ to the squared Euclidean metric in Definition 1, an $\varepsilon$-MSS $\mathcal{S}$ for $f$ at $\mathbf{x} = \mathbf{0}$ satisfies

$$
\begin{aligned}
\mathop{\mathbb{E}}_{\mathbf{X}_{\mathcal{S}^c}|\mathbf{x}_{\mathcal{S}}} \left[ \|A_{\mathcal{S}^c}\mathbf{X}_{\mathcal{S}^c} + A_{\mathcal{S}}\mathbf{x}_{\mathcal{S}} - A\mathbf{x}\|^2 \right] &= \mathbb{E}\left[\|A_{\mathcal{S}^c}\mathbf{X}_{\mathcal{S}^c}\|^2\right] && (\mathbf{x} = \mathbf{0} \text{ and } \mathbf{X}_{\mathcal{S}^c} \perp\!\!\!\perp \mathbf{X}_{\mathcal{S}}) \\
&= \mathrm{Tr}(A_{\mathcal{S}^c}^\top A_{\mathcal{S}^c}) && (\mathrm{Tr} \text{ commutes with } \mathbb{E} \text{ and } \mathbb{E}[\mathbf{X}_{\mathcal{S}^c}\mathbf{X}_{\mathcal{S}^c}^\top] = I) \\
&= |\mathcal{S}^c| && ([A_{\mathcal{S}^c}^\top A_{\mathcal{S}^c}]_{ii} = \|\mathbf{a}_i\|_2^2 = 1) \\
&\leq \varepsilon.
\end{aligned}
$$

Setting $\varepsilon = 1$, we have three MSSs: $\mathcal{S}_1 = \{2, 3\}$, $\mathcal{S}_2 = \{1, 3\}$, and $\mathcal{S}_3 = \{1, 2\}$.

In many real-world applications, it is much more natural to seek an $\varepsilon$-MSS with $\varepsilon > 0$. 0-MSSs may be so large (sometimes being the entire set of features) that they do not provide much practical insight or interpretability (Wäldchen et al., 2021). Therefore, asking for $\varepsilon$-MSSs with $\varepsilon > 0$ often renders Definition 1 more useful and practically applicable. As Example 3 illustrates, when asking for such MSSs, unique ones should not be expected.

### 4.4 Degenerate conditional feature distributions

In the previous section we saw that MSS multiplicity can depend on the properties of the model $f$. We end by showing that the feature distribution $\mathbb{P}_{\mathbf{X}}$ can *also* induce multiple MSSs independently of $f$, as the following example demonstrates.

**Example 4.** Let $\mathbf{X}$ be discrete with uniform probability over the set $\{-1/2, 1/2\}^d \cup \{\mathbf{1}\}$. Then fixing any $X_i = 1$ fully determines the remaining $\mathbf{X}_{[d]\setminus\{i\}}$:

$$
\mathbb{P}(\mathbf{X}_{[d]\setminus\{i\}} = \mathbf{1} \mid X_i = 1) = \frac{\mathbb{P}(\mathbf{X}_{[d]\setminus\{i\}} = \mathbf{1}, X_i = 1)}{\mathbb{P}(X_i = 1)} = 1,
$$

and thus

$$
\mathbb{E}[D(f(\mathbf{X}_{[d]\setminus\{i\}}, x_i), f(\mathbf{x})) \mid X_i = 1] = 0.
$$

I.e., for any $i \in [d]$, $\{i\}$ is a MSS for any $f$ at $\mathbf{x} = \mathbf{1}$.

In conclusion, we have discussed multiple perspectives on MSS non-uniqueness, demonstrating that disjunctive logic, symmetries and repetitions, intersection properties of context-specific independence, nonlinearity, approximate sufficiency, and degeneracies in the feature distribution all contribute to the presence of multiple MSSs. Taken together, these facts suggest that a unique MSS is the exception, rather than the rule. Therefore, rather than seeking a single MSS, it is better to seek multiple. To accomplish this, we now propose a simple method to extend techniques for finding a single MSS to strategies for finding several.

## 5 Let me explain, again

Since we do not know *a priori* when multiple minimal sufficient subsets will exist in a given scenario, it is sensible to look for multiple. However, some methods (e.g., (Fong & Vedaldi, 2017; Fong et al., 2019; MacDonald et al., 2019; Luss & Dhurandhar, 2024)) only find a single MSS. To extend such methods, we propose a strategy for identifying multiple MSSs. Similarly to prior work (Carter et al., 2019; 2021; Byra & Skibbe, 2025), this strategy finds explanations in an iterative fashion, constraining new explanations to be distinct from those found previously.

### 5.1 Meta-algorithm

The meta-algorithm we study here, which we term Let Me Explain Again (LMEA), finds multiple MSSs via repeated calls to a single-MSS explanation method, which we will call a *minimal sufficiency oracle (MSO)*.

In simple terms, LMEA proceeds as follows. We start with the full active feature pool $\mathcal{A} = [d]$, and iteratively search for MSSs by querying the MSO, which returns a set $\mathcal{S}$. We then add $\mathcal{S}$ to a running set $\mathcal{E}$ of MSSs found so far and remove $\mathcal{S}$ from the active feature pool $\mathcal{A}$. This process is repeated until the MSO returns an empty set or a user-specified maximum number of iterations $N$ is reached. In the algorithm, "removing" $\mathcal{S}$ is modeled theoretically by the set-difference operation $\mathcal{A} \setminus \mathcal{S}$. In practice, however, removal from the active feature pool $\mathcal{A}$ may be implemented in different ways depending on the MSO (see Section 6.3). The theoretical guarantees we provide shortly assume the set-difference $\mathcal{A} \setminus \mathcal{S}$ definition of removal.

By substituting various MSO approximators (as opposed to specific, fixed choices), LMEA generalizes the algorithms proposed by Carter et al. (2019;

---

**Algorithm 1** Let Me Explain Again (LMEA)

**Require:** Model $f$, input $\mathbf{x} \in \mathbb{R}^d$, sufficiency level $\varepsilon$, MSO, maximum explanations to return $N$

$\mathcal{A} \leftarrow [d]$
$\mathcal{E} \leftarrow \emptyset$
**for** $k = 1, \ldots, N$ **do**
    $\mathcal{S} \leftarrow \mathsf{MSO}(f, \mathbf{x}, \mathcal{A})$
    **if** $\mathcal{S} = \emptyset$ **then**
        **break**
    **end if**
    $\mathcal{E} \leftarrow \mathcal{E} \cup \{\mathcal{S}\}$
    $\mathcal{A} \leftarrow \mathcal{A} \setminus \mathcal{S}$
**end for**
**return** $\mathcal{E}$

---

2021), which operate via greedy search, and Byra & Skibbe (2025), which is similar to LMEA with extremal perturbations (XP) (Fong et al., 2019) as the MSO. We note that Algorithm 1 is not tied to our particular definition of sufficiency (Definition 1), and in Section 6, we show how LMEA can be applied to MSOs that optimize for different notions of sufficiency.

### 5.2 Termination and completeness

To provide theoretical guarantees on the termination time of LMEA and on properties of its final solution $\mathcal{E}$, we make the following assumption.

**Assumption 1.** The MSO correctly identifies an $\varepsilon$-MSS $\mathcal{S}$ for $f$ at $\mathbf{x}$, for all values of $\emptyset \neq \mathcal{A} \subseteq [d]$, provided an $\varepsilon$-MSS $\mathcal{S} \subseteq \mathcal{A}$ exists. If an $\varepsilon$-MSS does not exist, then the MSO returns the empty set $\emptyset$.

Under this assumption, Algorithm 1 terminates and has linear query complexity.

**Lemma 1.** *LMEA terminates after at most $d$ calls to the MSO, where $d$ is the dimension of $\mathbf{x}$.*

The proof is provided in Appendix C. Furthermore, beyond LMEA simply terminating, we can also show that LMEA inherits the correctness property of one of the algorithms it generalizes; the following result is analogous to an easy corollary of (Carter et al., 2019, Proposition 2) under certain assumptions.[4]

**Proposition 2.** *Under Assumption 1, LMEA with $N = d$ returns a set of explanations $\mathcal{E}$ whose union, $\mathcal{U} \dot{=} \bigcup_{\mathcal{S} \in \mathcal{E}} \mathcal{S}$, intersects each $\varepsilon$-MSS for $f$ at $\mathbf{x}$: for all $\varepsilon$-MSSs $\mathcal{S}$, $\mathcal{S} \cap \mathcal{U} \neq \emptyset$.*

In words, Proposition 2 demonstrates that each $\varepsilon$-MSS will have at least one feature in common with some explanation in the returned set $\mathcal{E}$ of MSSs. When there are multiple $\varepsilon$-MSSs and they are all disjoint, Proposition 2 implies that LMEA will recover them all. However, $\varepsilon$-MSSs may overlap. In high-dimensional feature spaces (such as high-resolution images), the degree of this overlap may become perceptually negligible, as $\mathcal{U}$ may intersect each explanation $\mathcal{S} \in \mathcal{E}$ at one out of hundreds of features comprising $\mathcal{S}$. In the worst case, when each MSS intersects every other, LMEA will only recover one MSS. To overcome this limitation, we believe that a principled backtracking approach could be incorporated, but that to do so, some additional structure on the set of MSSs would need to be assumed. We leave this for future work. Nevertheless, when

---

[4]Specifically, if we assume that each MSS $\mathcal{S}$ is such that every $\mathcal{T} \supseteq \mathcal{S}$ is also sufficient (with the definitions and context of (Carter et al., 2019)) then an analog of Proposition 2 follows from their Proposition 2 in the context of their paper.

multiple disjoint $\varepsilon$-MSSs are expected, the simplicity of LMEA allows it to perform well in practice, as we show in our experiments.

## 6 Experiments

To demonstrate the practical applicability of LMEA, we present experiments on three multiple instance learning (MIL) (Dietterich et al., 1997) image classification tasks, for which the presence of a single object in the image suffices for a positive prediction, and for which there may be many such objects. For each of these tasks, we run LMEA on images from the test set to recover as many explanations as possible. The explanations output by LMEA are evaluated against ground truth labels (either bounding boxes or segmentation masks) that indicate the salient regions of the image for the prediction, implying sufficiency according to human annotators. Code for our experiments is provided at `https://github.com/r-zip/LMEA`.

### 6.1 MSOs

**Rate distortion explanations (RDE).** One of our LMEA MSOs combines ideas from rate-distortion explanations (RDE) (MacDonald et al., 2019) and extremal perturbations (XP) (Fong et al., 2019). We choose RDE because its distortion criterion closely matches our sufficiency criterion in Definition 1, making it a good approximation to an MSO. More specifically, this MSO solves a problem of the form

$$\mathbf{s}^\star = h\left(\operatorname*{argmin}_{\mathbf{s} \in [0,1]^d} L(\mathbf{s})\right) \quad \text{with} \quad L(\mathbf{s}) \doteq \frac{1}{2}\,\mathbb{E}\left[\left\|f(h(\mathbf{s}) \odot \mathbf{x} + (\mathbf{1} - h(\mathbf{s})) \odot \widetilde{\mathbf{X}}) - f(\mathbf{x})\right\|^2\right] + \lambda \|h(\mathbf{s})\|_1, \quad (4)$$

where $\widetilde{\mathbf{X}} \overset{\text{i.i.d.}}{\sim} \mathcal{N}(\boldsymbol{\mu}, \boldsymbol{\Sigma})$ for some choice of $\boldsymbol{\mu} \in \mathbb{R}^d$, $\boldsymbol{\Sigma} \in \mathbb{R}^{d \times d}$ (MacDonald et al., 2019), and $h$ is a smoothing function for the mask $\mathbf{s}$. Empirically, this smoothing makes the resulting explanations more reliable, especially in later iterations of LMEA. Following Fong et al. (2019), we set $h$ to be the smoothmax function. The solution $\mathbf{s}^\star$ to Equation (4) can be thought of as a soft version of the characteristic vector $\mathbf{1}_{\mathcal{S}^\star}$ corresponding to a small $\varepsilon$-sufficient set $\mathcal{S}^\star$.[5] We solve (4) for a number of $\lambda$'s and pick the solution for the largest $\lambda$ (and thus the sparsest mask $\mathbf{s}^\star$) that produces an $\varepsilon$-sufficient result, where $\varepsilon$ is chosen as specified in Section 6.3.

**Extremal perturbations (XP).** We also study the use of extremal perturbations (XP) (Fong et al., 2019) as an MSO. XP solves the following optimization problem:

$$\mathbf{s}^\star = h\left(\operatorname*{argmax}_{\mathbf{s} \in [0,1]^d} f_{y^\star}(g(\mathbf{s}, \mathbf{x})) - \lambda R_\alpha(\mathbf{s})\right),$$

where $y^\star = \operatorname{argmax}_y f_y(\mathbf{x})$ is the predicted class, $g(\mathbf{s}, \mathbf{x})$ is a function that computes a perturbed version of the input $\mathbf{x}$ based on the mask $\mathbf{s}$, $R_\alpha(\mathbf{s})$ is a regularizer used to control the sparsity of the attribution mask $\mathbf{s}^\star$, $\alpha$ is a sparsity constraint parameter, and $\lambda$ is the corresponding regularization strength. The perturbation function $g$ can be a blurred version of the original image with local spread of the Gaussian blurring kernel determined by the smoothed mask value (so that $h(\mathbf{s})_i = 1$ corresponds to no blur and $h(\mathbf{s})_i = 0$ corresponds to maximal blur), or the "fade-to-black" perturbation $g(\mathbf{s}, \mathbf{x}) = h(\mathbf{s}) \odot \mathbf{x}$, where $h$ is the smoothmax function (Fong et al., 2019). Finally, for integer-valued $\alpha d$, $R_\alpha(\mathbf{s}) = \|\text{vecsort}(\mathbf{s}) - \mathbf{1}_{[\alpha d]}\|^2$ is an area constraint regularizer that encourages the mask $\mathbf{s}$ to be nearly binary and $\alpha d$-sparse. Although both XP and RDE find MSSs by solving continuous optimization problems, they differ in their (i) notions of sufficiency, (ii) methods of controlling sparsity, and (iii) perturbations of the input $\mathbf{x}$.

For the XP MSO, LMEA is similar to the method of Byra & Skibbe (2025) with an infinite Sørensen-Dice (Dice, 1945; Sørensen, 1948) penalty coefficient. Byra & Skibbe (2025) re-implement XP using coordinate-based implicit networks to represent the attribution masks. These networks can learn the non-linear and continuous relationships between the input features and their importance for the model's prediction. To generate $N$ explanations, their method requires retraining this network $N$ times, each time forcing the network to learn a new mask $\mathbf{s}$ that is different from the previous masks. While a powerful approach, it

---

[5]A characteristic vector $\mathbf{1}_{\mathcal{S}}$ is such that $(\mathbf{1}_{\mathcal{S}})_i = 1$ if $i \in \mathcal{S}$ and $(\mathbf{1}_{\mathcal{S}})_i = 0$ otherwise.

is also computationally expensive. We take a different, more efficient approach, and re-use (without modification) the implementation of XP provided in the TorchRay package, using a trick outlined in Section 6.3 to restrict the active set $\mathcal{A}$.[6] Similarly to RDE, we pick the explanation with the smallest area parameter that is $\varepsilon$-sufficient, where $\varepsilon$ is set as specified in Section 6.3.

## 6.2 Baselines

**Beam search (BS).** The method of Shitole et al. (2021) identifies multiple MSSs in an image by performing a systematic beam search in the space of image patches. A combinatorial search becomes infeasible if each image pixel is treated as a patch. To address this, the beam search approach divides the image into a coarser set of non-overlapping patches by resizing the input image to $224 \times 224$ and then partitioning it into a $7 \times 7$ grid. There are several hyperparameters used in the beam search; we provide details of the settings we use in Appendix E. The patch size $r$ is hard-coded in some places in the authors' implementation,[7] and implementing custom grid shapes is thus nontrivial. The neural networks for our shapes and malaria experiments (described in Sections 6.5.1 and 6.6, respectively) were trained with only one image size each, so the $224 \times 224$-pixel inputs required by BS are out-of-distribution for our models. This makes fair comparison with the other methods (which do not have a strict shape requirement) challenging. We therefore compare to this method only on the experiment described in Section 6.7, the model for which was trained on $224 \times 224$ images. For consistent comparison to LMEA and our other baselines, we constrain the set of explanations returned by BS to be approximately disjoint, as described further in Appendix E.2.

**MultiReX.** MultiReX (Chockler et al., 2025)[8] finds multiple explanations via a stochastic search procedure, based on a pre-computed ranking of pixels by causal responsibility for the classifier output. Our experiments test LMEA against this existing multi-MSS method, including both the number of explanations recovered and their quality. More details on MultiReX and its hyperparameters are given in Appendix E.1. For consistent comparison to LMEA, we set MultiReX's hyperparameters so that it returns a diverse set of explanations as measured by pairwise overlap. Specifically, we enforce a worst-case pairwise Sørensen-Dice score (Sørensen, 1948; Dice, 1945) between MSSs of 0.1.

**Patch-wise SIS (PSIS).** We also implement sufficient input subsets (SIS) (Carter et al., 2019). The original SIS paper proposed backward elimination on individual pixels. However, this approach requires $\mathcal{O}(d^2)$ forward passes through the model $f$ (Carter et al., 2021) and results in explanations that lack spatial contiguity, making them harder to interpret. To compare to this method in our high-dimensional image classification setup, we therefore implement SIS to do backward selection on a $14 \times 14$ grid of image patches, similar to the $7 \times 7$ grid used by Shitole et al. (2021). We refer to this method as patch-wise SIS (PSIS) and note that SIS can be interpreted as a special case of LMEA with a simple MSO and a slightly different notion of sufficiency. For fair comparison, we set the maximum number of explanations for SIS to $N = 20$, which matches the settings used for the other baselines. For the results in Tables 1–3, we follow the original and Google Research implementations of SIS[9] for vision and set the background $\mathbf{b}$ to a (constant) shared mean pixel value computed over the training set.

## 6.3 LMEA implementation details

Here we discuss two important implementation details of LMEA. Further implementation details are provided in Appendix H.1.

**Sparsity-based postprocessing (SBP).** Under Assumption 1, Algorithm 1 finds a set of true MSSs for $f$ at $\mathbf{x}$, but in practice, MSOs will not satisfy this assumption and will instead return some sets that are not MSSs under Definition 2. However, in certain situations, such as the experiments in Sections 6.5.1, 6.6, and 6.7, we may have prior knowledge about the classification problem that helps to mitigate this issue. Specifically, when MSSs are expected to be small and of approximately the same size, we propose to filter the

---

[6]The TorchRay package is available at https://github.com/facebookresearch/TorchRay.

[7]We use the implementation at `https://github.com/viv92/structured-attention-graphs`.

[8]We use the authors' implementation: https://github.com/ReX-XAI/ReX.

[9]The original is available at `https://github.com/b-carter/SufficientInputSubsets`, while the Google research version is available at `https://github.com/google-research/google-research/tree/master/sufficient_input_subsets`.

set of explanations $\mathcal{E}$ returned by Algorithm 1 based on sparsity, which we term sparsity-based postprocessing (SBP). After LMEA has terminated, we compute the minimum explanation size $K_{\min} \doteq \min_{\mathcal{S}' \in \mathcal{E}} |\mathcal{S}'|$, and we keep only the explanations that fall within a certain multiple of this size, say those $\mathcal{S}$ with $|\mathcal{S}| \leq (1+\delta)K_{\min}$ for some $\delta > 0$. In the results that follow, we set $\delta = 1$. For the MultiReX, PSIS, and BS baselines, we also apply this rule and report results both with and without SBP for consistent comparisons. SBP is an optional component of LMEA that can be applied when appropriate for the problem, and omitted otherwise.

**Specifying the active set $\mathcal{A}$.** It is convenient to be able to re-use MSO implementations within the LMEA framework. However, note that it is not guaranteed that a given implementation, e.g., the XP implementation discussed in Section 6.1, will accept a restricted feature pool $\mathcal{A}$ as an argument as specified in Algorithm 1 (aside from hyperparameters, such methods typically accept as input only the model $f$ and features $\mathbf{x}$, since these methods find a *single* MSS). Therefore, we propose a technique to wrap existing implementations of MSOs that optimize soft attribution masks by solving similar optimization problems to Equation 4, making LMEA more generally applicable to off-the-shelf MSOs.

To illustrate the idea, assume that $f$ is differentiable and that the MSO optimizes $L(\mathbf{s})$ in Equation (4) with $h(\mathbf{x}) = \mathbf{x}$ via projected gradient descent (PGD). Defining $\widetilde{\mathbf{X}}_{\mathbf{s}} \doteq \mathbf{s} \odot \mathbf{x} + (1-\mathbf{s}) \odot \widetilde{\mathbf{X}}$ and applying the chain rule, the gradient of the $n$-sample Monte Carlo estimate of the loss $L(\mathbf{s}^t)$ at iteration $t$ is

$$\nabla L(\mathbf{s}^t) = \frac{1}{2n} \sum_{n=1}^{n} \operatorname{diag}\big(\mathbf{x} - \widetilde{\mathbf{x}}_{\mathbf{s}^t}^{(n)}\big) J_f\big(\widetilde{\mathbf{x}}_{\mathbf{s}^t}^{(n)}\big)^\top \left( f\big(\widetilde{\mathbf{x}}_{\mathbf{s}^t}^{(n)}\big) - f(\mathbf{x}) \right) + \lambda \mathbf{1},$$

where $J_f\big(\widetilde{\mathbf{x}}_{\mathbf{s}^t}^{(n)}\big)$ is the Jacobian of $f$ evaluated at $\widetilde{\mathbf{x}}_{\mathbf{s}^t}^{(n)}$. If at each iteration $t$ we intervene on the columns of the Jacobian $J_f\big(\widetilde{\mathbf{x}}_{\mathbf{s}^t}^{(n)}\big)$ corresponding to the already selected features $\mathcal{A}^c$ by setting these columns to zero, i.e., $[J_f\big(\widetilde{\mathbf{x}}_{\mathbf{s}^t}^{(n)}\big)]_{\mathcal{A}^c} \leftarrow \mathbf{0}$, we then obtain the PGD update

$$\mathbf{s}_{\mathcal{A}^c}^{t+1} \leftarrow \operatorname{proj}_{[0,1]^d}\big(\mathbf{s}_{\mathcal{A}^c}^t - \eta\lambda\mathbf{1}\big)$$
$$\mathbf{s}_{\mathcal{A}}^{t+1} \leftarrow \operatorname{proj}_{[0,1]^d}\big(\mathbf{s}_{\mathcal{A}}^t - \eta[\nabla L(\mathbf{s}^t)]_{\mathcal{A}}\big),$$

where $\eta$ is the learning rate. Thus $\lim_{t \to \infty} \mathbf{s}_{\mathcal{A}^c}^t = \mathbf{0}$, while the gradient updates to the coordinates in $\mathcal{A}$ are left undisturbed. The above idea can be easily implemented in PyTorch (Paszke et al., 2019) via a simple wrapper over the model $f$ that disconnects certain variables from the computation graph (sample code is provided in Appendix H.1), rather than directly intervening in the optimization process. This approach allows obtaining new explanations simply by passing a wrapped model to the MSO, without altering the MSO implementation itself.[10] To adapt this idea to smoothing functions $h \neq \operatorname{Id}$ in our experiments, we perform a morphological dilation (see, e.g., (Haralick et al., 1987)) on each explanation $\mathcal{S}$ before removing it from $\mathcal{A}$ in Algorithm 1. Further discussion on the motivation behind this dilation, guidance on setting related hyperparameters, and dilation-related implementation details for our experiments are provided in Appendix I.

## 6.4 Evaluation

For the remainder of the paper, we will refer to LMEA, MultiReX, PSIS, and BS as *multi-explanation methods (MEMs)*, since they each seek multiple MSSs. The goal of MEMs (including LMEA) is to find multiple MSSs; the number of MSSs recovered by these methods is thus a natural metric of interest. Ideally, each MEM or MSO would find a subset of the set of true MSSs (according to Definition 2). In this case, comparing the sizes and diversities of the subsets of MSSs produced by each method might suffice, and we might simply prefer MEMs that find greater numbers of MSSs. In practice, however, the methods studied measure sufficiency with respect to some background value $\mathbf{b}$ or some mismatched distribution $\mathbb{Q} \neq \mathbb{P}_{\mathbf{X}}$ over $\mathbf{X}$, as described in Sections 6.1–6.2. Furthermore, many methods for finding MSSs (e.g., RDE (MacDonald et al., 2019)) resort to relaxations of the true problem and are not guaranteed to converge to global optima. The MSSs recovered by such methods are thus not necessarily minimal nor sufficient according to Definitions 1–2.

---

[10] A similar argument applies to XP. XP uses an area constraint penalty and logarithmic parametrization of $\mathbf{s}^t$, so we are not guaranteed to drive $\mathbf{s}_{\mathcal{A}^c}^t$ to zero with this method; however, we observe that LMEA with the XP MSO rarely re-selects regions in our experiments.

It is therefore of interest also to evaluate some notion of the "correctness" of the MSSs found by each method, but in practice, the true set of MSSs is unknown. Toward a tractable, approximate notion of correctness, we measure how well these MSSs coincide with regions of the input that are salient for the prediction, as judged by human annotators. When the truly discriminative input regions are known *a priori* and the model achieves perfect held-out accuracy (as in our synthetic experiment in Section 6.5.1), we expect the model to rely on these regions alone for its classification. Our experiments on real data in Sections 6.6–6.7 depart from this ideal. However, we still expect the models for those experiments, which achieve high held-out accuracy, to rely substantially on the labeled ground truth regions, making overlap between MSSs and ground truth labels a useful proxy for sufficiency according to Definition 1.

We now introduce several metrics to quantify performance in light of the preceding discussion.

**Number of MSSs recovered.** First, we report the total number of explanations returned by each method, which is denoted "Expl." in the tables that follow. All else being equal, we consider a higher value for this metric to be better.

**MSS overlap with ground truth regions.** Next, we introduce metrics to determine the degree to which explanations overlap with the known informative regions of the image for the given task. We report intersection-over-union (IoU) scores between the union of MEM explanation mask regions $\mathcal{R}_E \doteq \bigcup_{\mathcal{S} \in \mathcal{E}} \mathcal{S}$ and the union of the ground truth segmentation mask regions $\mathcal{R}_G \doteq \bigcup_{\mathcal{T} \in \mathcal{M}} \mathcal{T}$, where $\mathcal{M}$ is the set of all ground truth segmentation masks for $\mathbf{x}$:

$$\text{IoU}(\mathcal{R}_E, \mathcal{R}_G) \doteq \frac{|\mathcal{R}_E \cap \mathcal{R}_G|}{|\mathcal{R}_E \cup \mathcal{R}_G|}. \tag{5}$$

A score of 1 indicates perfect overlap, while a score of 0 indicates no overlap. Towards a notion of recall, we use the intersection-over-ground-truth (IoGT) metric, representing the same score instead normalized by the area of the ground truth segmentation:

$$\text{IoGT}(\mathcal{R}_E, \mathcal{R}_G) \doteq \frac{|\mathcal{R}_E \cap \mathcal{R}_G|}{|\mathcal{R}_G|}. \tag{6}$$

If IoGT is 1, then all ground truth pixels belong to some explanation output by the MEM explanation, while an IoGT of 0 indicates that none were recovered. Similarly, we use the intersection-over-explanation (IoE) metric as a proxy for precision. The IoE is the intersection normalized by the explanation mask area:

$$\text{IoE}(\mathcal{R}_E, \mathcal{R}_G) \doteq \frac{|\mathcal{R}_E \cap \mathcal{R}_G|}{|\mathcal{R}_E|}. \tag{7}$$

If IoE equals 1, then each pixel in each of the MEM explanations belongs to a ground truth object annotation. If it equals zero, then none belong to any ground truth object annotation. We report IoU and IoGT values without comparison across single-MSS methods (i.e., MSOs), since individual explanations may overlap a single salient object or some fraction thereof, which may constitute a small portion of $\mathcal{R}_G$.

One may wonder why we do not set some thresholds on these intersection metrics and evaluate based on the precision/recall scores at these thresholds, as is done in, e.g., object detection (Padilla et al., 2021). Because we are explaining the prediction process of a classifier (as opposed to an object detection or segmentation model) it is unreasonable to expect that each explanation will overlap exactly with the ground truth annotations. For example, if an image is classified as "dog," the pixels corresponding to the head or tail may be sufficient for the classifier to make its decision, explanations which would have small IoGT and IoU values. Furthermore, if two dogs are in an image, a classifier might use the head of one and the tail of another to make the prediction "dog," which is valid but nevertheless does not correspond to a single ground truth bounding box. We therefore simply report the raw scores for these quantities, which consider unions of individual explanations and ground truth labels.

**Additional metrics.** Finally, we report auxiliary metrics to further quantify MEM and MSO performance. Since MSSs defined in terms of *set size*, rather than *set inclusion*, are often of interest (see, for instance, (MacDonald et al., 2019)), we report the sparsity of the binarized attribution masks, that is, the size of each

MSS. To quantify the correspondence between the number of MSSs recovered and the number of ground truth objects in the image, we report "Expl./Obj.," representing the number of explanations per image that each MEM produces, normalized by the number of salient objects per image. For this metric, we refrain from specifying whether lower or higher values are better, since the dependence of the number of true MSSs on the number of salient objects in the image is unknown. Finally, for some samples, RDE and XP are unable to find any explanation. To quantify this, we also report "Missing," which is the number of instances for which each method failed to find any explanation. Ideally, this metric will be zero, and lower is better. Samples with missing explanations are excluded from metrics calculations, with the exception of "Expl./Obj." and "Expl."

**Further evaluation details.** For each dataset, we draw a held-out set from the training set (separate from the existing validation and test sets), which we use for MSO hyperparameter selection. For comparison between LMEA and MultiReX, we run each explanation on every positive sample from the test set that has a confident positive prediction, that is, an estimated positive probability of at least 0.8.

**Interpretation of performance metrics.** We seek to evaluate LMEA along two different axes: (i) how well it extends MSOs to find multiple MSSs, and (ii) how it performs relative to existing MEMs. For comparisons along axis (i), we present metrics for each wrapped method (RDE and XP) alongside LMEA with that method as its MSO. Ideally, LMEA will have higher values of "Expl." and similar values of "IoU," "IoGT," and "IoE" compared to these wrapped methods alone. For comparisons along axis (ii), we present metrics for LMEA and the baselines introduced in Section 6.2 side-by-side. For each metric, we provide an up arrow ($\uparrow$) or down arrow ($\downarrow$) in our tables to indicate whether lower or higher values of that metric are better for this MEM performance comparison.

### 6.5 Synthetic experiments

We perform a series of synthetic experiments to evaluate various properties of LMEA's performance. In the following subsection, we present an experiment demonstrating the performance of LMEA against our baselines on a synthetic vision example. Further experiments that study (i) LMEA's ability to recover the complete set of MSSs when these MSSs overlap and (ii) the impact of violating Assumption 1 on LMEA's ability to recover MSSs are deferred to Appendix D.

#### 6.5.1 Synthetic experiment: shapes dataset

First, we investigate the recovery rate of MSSs when the data generating process is fully known. To do this, we use Teneggi et al. (2022)'s synthetic shapes dataset, which consists of images of colorful shapes on a grid over a white background. Each shape (circle, square, triangle, "X") occupies a $10 \times 10$ patch of the size-$100 \times 120$ image, and there is a random number of such shapes per image. The dataset is labeled such that $y^{(i)} = 1$ if there is at least one "X" shape present in image $\mathbf{x}^{(i)}$, else $y^{(i)} = 0$. We expect that the explanations recovered by each MEM should consist of some portion of an "X," and that the pixels of each "X" should be included in the union of all explanations output by each MEM.

Following a similar setup to Teneggi et al. (2022), we train a small convolutional neural network (a layerwise-downsized version of (Chollet, 2021, Listing 8.1)) on this dataset to 100% test accuracy. For the RDE MSO, we set the size of the smoothmax kernel to $11 \times 11$ pixels. For full details on the setup, including classifier training and RDE hyperparameters, see Appendix H.1. For this method, LMEA is run for a maximum of ten iterations, meaning it can find up to ten explanations.[11] Randomly sampled runs of LMEA and MultiReX on this dataset are pictured in Figure 2.[12] The numerical results for this dataset are reported in Table 1. For this dataset, we use the provided bounding box annotations for evaluation. We find that, most of the time, MultiReX selects pieces of each "X" as an explanation, while LMEA with the RDE MSO selects an entire "X" or multiple "X"s as an explanation at each step, and LMEA with the XP MSO tends to select a single, complete "X" each time. PSIS tends to select patches of each "X". Because they select smaller regions, MultiReX and PSIS tend to find more explanations on this dataset. However, LMEA has higher

---

[11] MultiReX does not have an analagous hyperparameter. For details on MultiReX's hyperparameter settings, which are constant across experiments, see Appendix E.

[12] For details on this random sampling procedure, see Appendix G.

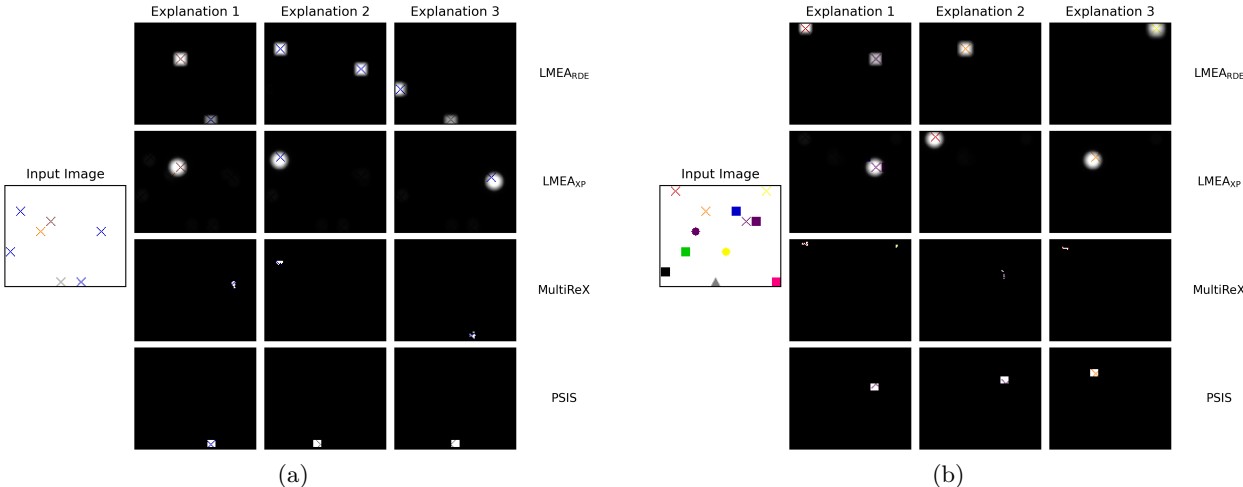

Figure 2: Sample runs of LMEA and baselines on images from the shapes dataset, where the model is trained to predict the presence of at least one "X". Black regions are unselected by each MEM. *Left*: without SBP. *Right*: with SBP. *Top row of each subfigure:* LMEA with RDE MSO. *Second row of each subfigure:* LMEA with XP MSO. *Bottom two rows of each subfigure:* MultiReX and PSIS. Further examples are provided in Appendix F.

Table 1: Results on the Shapes dataset. The table is grouped into single-MSS methods (MSOs) and multi-MSS methods (LMEA, MultiReX, and PSIS). The RDE and XP columns indicate the performance of those MSOs on their own, while the $\text{LMEA}_{\text{RDE}}$ and $\text{LMEA}_{\text{XP}}$ columns indicate the performance of LMEA with those choices of MSO. The table reports results both with and without the SBP procedure. Where applicable, each metric is reported alongside its standard deviation (in parentheses), and the result with the best mean value in each (single-/multi-MSS) category is bolded. The Expl./Obj. metric is not bolded, and the IoU and IoGT metrics are not bolded for single-MSS methods, for reasons explained in Section 6.4.

| | Metric | | Single-MSS | | Multi-MSS | | | |
| --- | --- | --- | --- | --- | --- | --- | --- | --- |
| | | | RDE | XP | $\text{LMEA}_{\text{RDE}}$ | $\text{LMEA}_{\text{XP}}$ | MultiReX | PSIS |
| **Without SBP** | Expl./Obj. | | 0.31 (0.27) | 0.31 (0.27) | 0.63 (0.16) | 0.93 (0.10) | 1.85 (0.74) | 3.21 (1.24) |
| | Expl. | ↑ | 1.00 (0.00) | 1.00 (0.00) | 3.51 (1.15) | 5.74 (2.01) | 9.55 (2.71) | **16.87** (4.35) |
| | IoU | ↑ | 0.30 (0.15) | 0.19 (0.08) | **0.55** (0.05) | 0.36 (0.02) | 0.14 (0.09) | 0.33 (0.10) |
| | IoGT | ↑ | 0.41 (0.29) | 0.32 (0.26) | **0.95** (0.07) | 0.92 (0.08) | 0.34 (0.10) | 0.92 (0.09) |
| | IoE | ↑ | **0.60** (0.09) | 0.39 (0.06) | 0.57 (0.05) | 0.37 (0.02) | 0.23 (0.20) | 0.34 (0.11) |
| | Sparsity | ↓ | **1.64** (0.54) | 1.85 (0.05) | 1.87 (0.59) | 1.81 (0.22) | 1.11 (2.42) | **0.66** (0.54) |
| | Missing | ↓ | **0.00** | **0.00** | **0.00** | **0.00** | **0.00** | **0.00** |
| **With SBP** | Expl./Obj. | | — | — | 0.59 (0.20) | 0.93 (0.11) | 1.01 (0.57) | 2.76 (0.51) |
| | Expl. | ↑ | — | — | 3.10 (1.10) | 5.72 (2.02) | 4.54 (1.63) | **16.08** (4.75) |
| | IoU | ↑ | — | — | **0.50** (0.10) | 0.36 (0.02) | 0.18 (0.10) | 0.39 (0.05) |
| | IoGT | ↑ | — | — | 0.84 (0.21) | **0.92** (0.09) | 0.19 (0.12) | 0.89 (0.09) |
| | IoE | ↑ | — | — | 0.57 (0.06) | 0.37 (0.03) | **0.86** (0.12) | 0.42 (0.05) |
| | Sparsity | ↓ | — | — | 1.76 (0.51) | 1.81 (0.15) | **0.16** (0.07) | 0.54 (0.17) |
| | Missing | ↓ | — | — | **0.00** | **0.00** | **0.00** | **0.00** |

maximum IoGT scores (over MSOs), indicating that it does a better job of recovering all relevant pixels for those MSOs.

These results are expected, since MultiReX operates by masking images with a (scalar) background value, which in these experiments corresponds to the white background. That is, its naive masking strategy is

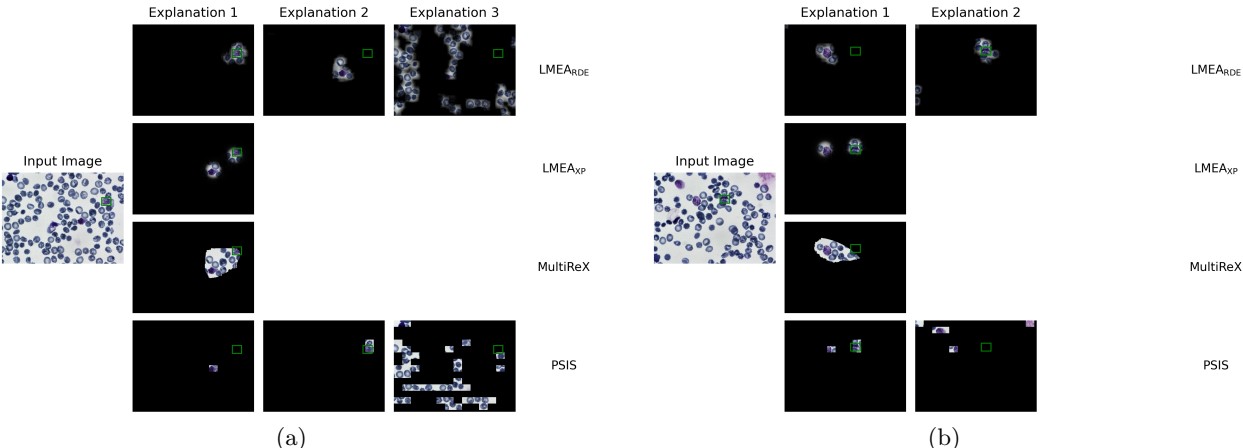

Figure 3: Sample runs of LMEA and baselines on images from the BBBC041 dataset, where the classifier is trained to predict the presence of at least one trophozoite in the image. Green squares indicate the bounding box labels corresponding to infected (trophozoite) cells. *Left*: without SBP. *Right*: with SBP. *Top row of each subfigure:* LMEA with RDE MSO. *Middle row of each subfigure:* LMEA with XP MSO. *Bottom two rows of each subfigure:* MultiReX and PSIS. Further examples are provided in Appendix F.

a perfect match for this simple dataset. RDE, on the other hand, uses a mismatched background value corresponding to the empirical mean image computed over the training set. RDE and XP are also constrained to select larger regions due to their smoothing kernels. For this dataset, SBP substantially improves MultiReX's IoE score. Aside from reducing the number of explanations found, it does not otherwise hurt performance. The results show that LMEA succeeds in augmenting RDE and XP to find multiple explanations, as measured by Expl., Expl./Obj. and IoGT metrics.

## 6.6 BBBC041 dataset

Next, we examine the performance of LMEA on the task of detecting malaria-infected cells in blood smears. For this experiment we use the BBBC041 dataset (Ljosa et al., 2012).[13] The dataset comes with expert annotations for each cell type, including healthy cells (red blood cells, leukocytes) and infected cells (rings, trophozoites, schizonts, gametocytes). Following Teneggi et al. (2022), we consider the problem of classifying each blood smear image based on whether or not at least one trophozoite cell is present. We use Teneggi et al. (2022)'s dataset for this task (derived from BBBC041) which has labeled $y^{(i)} = 1$ if a trophozoite is present in the image $\mathbf{x}^{(i)}$, else $y^{(i)} = 0$, and we follow their model architecture and training setup. Specifically, we fine-tune a ResNet18 network (He et al., 2016) that was pre-trained on ImageNet (Deng et al., 2009). The trained model achieves a test accuracy of 93.42%. For further details on the experimental setup, see Appendix H.1.

For this dataset, LMEA is run for a maximum of five iterations, meaning it can find up to five explanations. We provide examples of LMEA, MultiReX, and PSIS explanations in Figure 3. We find that all methods tend to select trophozoite cells, but often also include other cells in their explanations. These other cells are usually visually similar or belong to a neighborhood surrounding an infected cell.

Quantitative results for this dataset are provided in Table 2. Without SBP, LMEA is competitive with baselines across overlap metrics. However, LMEA with the XP MSO finds explanations that are more specific to known salient ground truth regions, as measured by IoE. Also, without SBP, LMEA$_{XP}$'s explanations are much sparser on average than the baselines. With SBP, LMEA$_{XP}$ and PSIS perform comparably on IoU and IoE, while LMEA with either MSO performs better than either baseline on IoGT. MultiReX produces larger explanations than the other methods. Overall, LMEA increases the number of explanations found by

---

[13]The original dataset is available here: https://bbbc.broadinstitute.org/BBBC041, but we use a re-split version of the pre-processed dataset provided by Teneggi et al. (2022) as described in Appendix H.3.

Table 2: Results on the BBBC041 dataset. The table is grouped into single-MSS methods (MSOs) and multi-MSS methods (LMEA, MultiReX, and PSIS). The RDE and XP columns indicate the performance of those MSOs on their own, while the $\text{LMEA}_{\text{RDE}}$ and $\text{LMEA}_{\text{XP}}$ columns indicate the performance of LMEA with those choices of MSO. The table reports results both without and with the SBP procedure. Where applicable, each metric is reported alongside its standard deviation (in parentheses), and the result with the best mean value in each (single-/multi-MSS) category is bolded. Samples where LMEA/RDE failed to find any explanation were excluded from the corresponding metric calculations, with the exception of the Expl./Obj. metric. The Expl./Obj. metric is not bolded, and the IoU and IoGT metrics are not bolded for single-MSS methods, for reasons explained in Section 6.4.

| | Metric | | Single-MSS | | Multi-MSS | | | |
| | | | RDE | XP | $\text{LMEA}_{\text{RDE}}$ | $\text{LMEA}_{\text{XP}}$ | MultiReX | PSIS |
|---|---|---|---|---|---|---|---|---|
| **Without SBP** | Expl./Obj. | | 0.50 (0.32) | 0.50 (0.32) | 1.26 (0.80) | 0.77 (0.54) | 0.71 (0.47) | 1.95 (1.78) |
| | Expl. | ↑ | 1.00 (0.00) | 0.96 (0.19) | 3.21 (1.06) | 2.07 (1.06) | 1.71 (0.59) | **4.58** (1.89) |
| | IoU | ↑ | 0.27 (0.15) | 0.27 (0.16) | 0.11 (0.09) | **0.31** (0.14) | 0.11 (0.12) | 0.07 (0.05) |
| | IoGT | ↑ | 0.57 (0.32) | 0.58 (0.34) | **0.90** (0.12) | 0.80 (0.21) | 0.69 (0.30) | 0.85 (0.10) |
| | IoE | ↑ | 0.35 (0.18) | **0.39** (0.20) | 0.12 (0.10) | **0.36** (0.17) | 0.13 (0.17) | 0.07 (0.06) |
| | Sparsity | ↓ | 3.69 (0.96) | **3.63** (2.47) | 9.21 (8.38) | **3.91** (2.43) | 17.23 (14.29) | 10.94 (14.08) |
| | Missing | ↓ | **0.00** | 3.53 | **0.00** | 3.53 | **0.00** | **0.00** |
| **With SBP** | Expl./Obj. | | — | — | 0.75 (0.49) | 0.73 (0.57) | 0.53 (0.31) | 1.04 (0.90) |
| | Expl. | ↑ | — | — | 2.01 (0.84) | 1.78 (0.99) | 1.21 (0.41) | **2.62** (1.46) |
| | IoU | ↑ | — | — | 0.26 (0.12) | **0.29** (0.14) | 0.13 (0.14) | 0.27 (0.13) |
| | IoGT | ↑ | — | — | **0.74** (0.24) | 0.72 (0.28) | 0.54 (0.35) | 0.53 (0.23) |
| | IoE | ↑ | — | — | 0.29 (0.15) | 0.37 (0.18) | 0.20 (0.24) | **0.41** (0.20) |
| | Sparsity | ↓ | — | — | 3.97 (1.15) | 3.54 (2.16) | 12.67 (10.37) | **1.74** (0.99) |
| | Missing | ↓ | — | — | **0.00** | 3.53 | **0.00** | **0.00** |

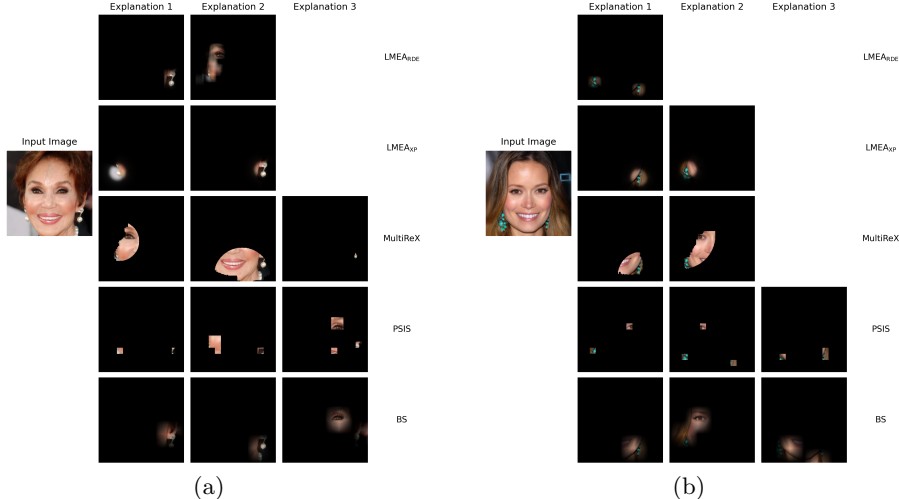

Figure 4: Randomly sampled runs of LMEA and baselines on images from the CelebAMask-HQ dataset, where the classifier is trained to predict the presence of the "wearing earrings" label. *Left*: without SBP. *Right*: with SBP. *Top row of each subfigure:* LMEA with RDE MSO. *Middle row of each subfigure:* LMEA with XP MSO. *Bottom three rows of each subfigure:* MultiReX, PSIS, and BS. Further examples are provided in Appendix F.

the wrapped MSOs, as measured by Expl., Expl./Obj., and IoGT metrics. The results also indicate that SBP allows LMEA to extend RDE to recover multiple explanations without sacrificing much specificity, as measured by IoE, and that SBP improves the sparsity and specificity of PSIS explanations.

## 6.7  CelebAMask-HQ dataset

We also apply LMEA to natural images, namely portraits of celebrity faces. We use the CelebAMask-HQ dataset (Lee et al., 2020) for this task. This dataset comes with segmentation annotations for a number of attributes, including physical characteristics and attire. Some of these, such as "wearing earrings" come in pairs. For this experiment, we choose this "wearing earrings" annotation and study whether LMEA can recover both earring regions: intuitively, each explanation should contain (a portion of) one earring if the classifier solves the task perfectly, i.e., if the explanation is an MSS.

Similar to the setup in Section 6.6, we fine-tune a ResNet18 (He et al., 2016) that was pre-trained on ImageNet (Deng et al., 2009). Due to label accuracy issues for the "wearing earrings" label (Wu et al., 2023), we use the presence of earring segmentation masks as labels instead of the original dataset labels. The trained model achieves a test accuracy of 89.51%, again with respect to the labels derived from the presence/absence of earring segmentation masks. For full details on model training and MEM hyperparameters, see Appendix H.1. For this method, LMEA is run for a maximum of three iterations, meaning it can find up to three explanations. We provide example runs of each MEM for this dataset in Figure 4, and the numerical results in Table 3. Without SBP, we can see that LMEA finds explanations that are more specific to the ground truth regions than baselines, as measured by IoU and IoE metrics. With SBP, LMEA is competitive on IoU and IoE metrics; however, LMEA underperforms BS on IoGT, both with and without SBP. As on the other datasets, the Expl., Expl./Obj., and IoGT metrics show that LMEA improves the ability of XP and RDE to find multiple MSSs (although the IoGT improvement is less pronounced on this dataset). Furthermore, SBP improves the IoU and IoE scores of PSIS without worsening other metrics besides Expl.

## 6.8  Additional experiments

As described in Section 6.2 and Appendix E, MultiReX, PSIS, and BS in the preceding experiments were each configured to use one of their respective default settings for the background value used to measure sufficiency.

Table 3: Results on the CelebAMask-HQ dataset. The table is grouped into single-MSS methods (MSOs) and multi-MSS methods (LMEA, MultiReX, PSIS, and BS). The RDE and XP columns indicate the performance of those MSOs on their own, while the $\text{LMEA}_{\text{RDE}}$ and $\text{LMEA}_{\text{XP}}$ columns indicate the performance of LMEA with those choices of MSO. The table reports results both without and with the SBP procedure. Where applicable, each metric is reported alongside its standard deviation (in parentheses), and the result with the best mean value in each (single-/multi-MSS) category is bolded. Samples where LMEA/RDE failed to find any explanation were excluded from the corresponding metric calculations, with the exception of the "Expl./Obj." metric. The Expl./Obj. metric is not bolded, and the IoU and IoGT metrics are not bolded for single-MSS methods, for reasons explained in Section 6.4.

| | Metric | | Single-MSS | | Multi-MSS | | | | |
| | | | RDE | XP | $\text{LMEA}_{\text{RDE}}$ | $\text{LMEA}_{\text{XP}}$ | MultiReX | PSIS | BS |
|---|---|---|---|---|---|---|---|---|---|
| **Without SBP** | Expl./Obj. | | 0.62 (0.22) | 0.62 (0.22) | 1.11 (0.46) | 1.15 (0.43) | 1.24 (0.50) | 2.56 (1.37) | 1.75 (0.77) |
| | Expl. | ↑ | 0.99 (0.09) | 1.00 (0.05) | 1.90 (0.65) | 1.98 (0.53) | 2.29 (0.97) | **4.56** (2.56) | 3.17 (1.38) |
| | IoU | ↑ | 0.14 (0.11) | 0.12 (0.08) | 0.10 (0.09) | **0.12** (0.10) | 0.03 (0.03) | 0.05 (0.05) | 0.06 (0.05) |
| | IoGT | ↑ | 0.55 (0.30) | 0.51 (0.28) | 0.71 (0.25) | 0.76 (0.24) | 0.40 (0.30) | 0.76 (0.24) | **0.88** (0.19) |
| | IoE | ↑ | **0.19** (0.16) | 0.16 (0.14) | 0.12 (0.12) | **0.13** (0.12) | 0.03 (0.03) | 0.06 (0.05) | 0.07 (0.06) |
| | Sparsity | ↓ | **4.21** (1.78) | 4.25 (2.37) | 6.19 (4.39) | **5.24** (3.59) | 9.06 (8.91) | 6.86 (9.28) | 5.63 (2.11) |
| | Missing | ↓ | 0.88 | **0.22** | 0.88 | 0.22 | **0.00** | **0.00** | **0.00** |
| **With SBP** | Expl./Obj. | | — | — | 0.88 (0.39) | 0.94 (0.30) | 0.81 (0.33) | 1.75 (1.18) | 1.59 (0.72) |
| | Expl. | ↑ | — | — | 1.49 (0.53) | 1.68 (0.48) | 1.39 (0.54) | **3.16** (2.13) | 2.88 (1.34) |
| | IoU | ↑ | — | — | 0.13 (0.10) | 0.14 (0.10) | 0.03 (0.04) | **0.15** (0.11) | 0.07 (0.05) |
| | IoGT | ↑ | — | — | 0.66 (0.28) | 0.72 (0.27) | 0.26 (0.28) | 0.53 (0.28) | **0.84** (0.21) |
| | IoE | ↑ | — | — | 0.16 (0.15) | 0.17 (0.13) | 0.06 (0.11) | **0.20** (0.14) | 0.07 (0.06) |
| | Sparsity | ↓ | — | — | 4.56 (2.11) | 4.07 (2.24) | 6.02 (4.82) | **2.23** (2.80) | 5.36 (1.93) |
| | Missing | ↓ | — | — | 0.88 | 0.22 | **0.00** | **0.00** | **0.00** |

To investigate the impact of these background values on the results, we also conducted an experiment with LMEA and all MEM baselines configured to use the same mean background used by LMEA. These results and corresponding discussion are provided in Appendix K. Overall, while the MEM baselines can sometimes find many more MSSs with this mean background value, LMEA tends to match or exceed the baselines in terms of specificity to the ground truth annotations, as measured by IoU and IoE.

### 6.9 Summary

The experiments show that LMEA recovers more MSSs than the wrapped MSO, and that the explanations output by LMEA are similar in quality to the MSO explanation in terms of sparsity and overlap with ground truth labels, when SBP is applied. We also find that LMEA is competitive with two multiple-MSS explanation baselines, MultiReX and PSIS, in terms of IoU, IoE, and IoGT (this latter metric to a lesser extent when compared to BS). Our results further demonstrate that SBP improves the specificity of both LMEA and PSIS without sacrificing other metrics besides the number of explanations found.

## 7 Limitations

The main advantage of LMEA is its simplicity, but this simplicity comes with certain limitations. It is not guaranteed that $\varepsilon$-MSSs will be disjoint, and, as discussed in Section 5.2, LMEA's performance will likely suffer in settings where $\varepsilon$-MSSs overlap substantially. However, for MSOs based on optimizing a smooth attribution mask, such as XP and RDE, overlapping explanations can be obtained by reducing the morphological dilation radius mentioned in Section 6.3, or omitting the dilation step entirely. The diversity of the resulting explanations can then be controlled by a postprocessing pruning step, as in (Shitole et al., 2021).

Algorithm 1 assumes that there is a way to specify the active set $\mathcal{A}$ to the MSO, but this is atypical. In this work, we present a practical and effective approach to avoid this limitation in Section 6.3 for two gradient-based MSOs; however, in general some reimplementation of the MSO may be required to restrict the active feature pool $\mathcal{A}$. Furthermore, for some MSOs, there may be more natural ways to find multiple MSSs. For instance, some logic-based explanations such as (Ignatiev et al., 2020) already support enumeration of MSSs.

To assess the quality of the MSSs found by different methods, our experiments in Section 6 use human-annotated salient regions of the image in lieu of the (unknown) true underlying MSSs according to Definition 2. While evaluation against these human annotations is a useful proxy for correctness, it is limited, since features outside of these regions may influence a model's predictions. Such features might therefore belong to some set of true MSSs, and yet result in lower IoU, IoE, and IoGT scores. Future work should explore evaluation protocols that mitigate this limitation without knowledge of the true underlying MSSs, which are intractable to compute.

Finally, our experiments involve MIL datasets which we know *a priori* to have several non-overlapping MSSs, and these may not be reflective of other machine learning scenarios. That said, our synthetic experiments in Appendix D.1 indicate that LMEA's performance degrades gracefully with the degree of MSS overlap, and in other structured domains such as text, we also anticipate that LMEA will succeed in extending existing single-MSS methods. These settings are left as a matter of future work.

## 8 Conclusion

In this work, we presented several complementary perspectives on the non-uniqueness of MSSs, including the multiplicity of (probabilistic) prime implicants, the inability of intersection properties of (context-specific) CI to guarantee uniqueness, the breakdown of the MSS uniqueness property for linear models once exact sufficiency ($\varepsilon = 0$) is relaxed to approximate sufficiency ($\varepsilon > 0$), and the potential for degenerate distributions to cause multiple MSSs regardless of the model $f$.

Motivated by the ubiquitous nature of MSS multiplicity, we proposed a meta-algorithm, LMEA, that generalizes previously studied approaches (Carter et al., 2019; 2021; Byra & Skibbe, 2025) and showed that

like (Carter et al., 2019), it recovers a representative set of all $\varepsilon$-MSSs. Experiments on three MIL datasets benchmarked LMEA against prior approaches and demonstrated the utility of LMEA for extending two MSOs that operate via gradient descent on attribution masks.

**Acknowledgments**

This work was supported by NSF grants 2212457 and 2031985, Simons Foundation grant 814201, NSF CAREER Award CCF 2239787, and NIH award R01CA287422. The authors thank Ryan Chan and Jacopo Teneggi for helpful conversations as well as Leandro Palma for feedback on an early draft of this work.

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

# A   Proof that MSSs correspond to context-specific CI relationships

In Section 4.2, we claimed that, if $f$ represents the true conditional probability mass function (pmf),

$$f_y(\mathbf{x}) = f_y^\star(\mathbf{x}) := p(y \mid \mathbf{x}),$$

then any MSS $\mathcal{S}$ for $f$ at $\mathbf{x}$ satisfies $\mathbf{X}_{\mathcal{S}^c} \perp\!\!\!\perp \mathbf{Y} \mid \mathbf{X}_{\mathcal{S}} = \mathbf{x}_{\mathcal{S}}$. While this result is straightforward, the formal statement and proof are included here for completeness.

The following result, which is a simple consequence of the properties of $D$ and the expectation operator $\mathbb{E}_{\mathbf{X}_{\mathcal{S}^c} \mid \mathbf{x}_{\mathcal{S}}}$, formalizes what is meant by the above CI notation.

**Claim 1.** *Let $\mathcal{S} \subseteq [d]$ be sufficient for $f = f^\star$ at $\mathbf{x}$. Then $\mathbf{X}_{\mathcal{S}^c} \perp\!\!\!\perp \mathbf{Y} \mid \mathbf{X}_{\mathcal{S}} = \mathbf{x}_{\mathcal{S}}$, that is,*

$$p(y \mid \mathbf{x}) \stackrel{\text{a.s.}}{=} p(y \mid \mathbf{X}_{\mathcal{S}^c}, \mathbf{x}_{\mathcal{S}}),$$

*where "a.s." denotes almost-sure equality with $\mathbf{X}_{\mathcal{S}^c} \sim \mathbb{P}_{\mathbf{X}_{\mathcal{S}^c} \mid \mathbf{X}_{\mathcal{S}} = \mathbf{x}_{\mathcal{S}}}$.*

*Proof.* Define $\Delta \doteq \Delta(\mathbf{X}_{\mathcal{S}^c}) \doteq D(f(\mathbf{X}_{\mathcal{S}^c}, \mathbf{x}_{\mathcal{S}}), f(\mathbf{x}))$, where $\mathbf{X}_{\mathcal{S}^c} \sim \mathbb{P}_{\mathbf{X}_{\mathcal{S}^c} \mid \mathbf{X}_{\mathcal{S}} = \mathbf{x}_{\mathcal{S}}}$. Recall that we assume $\mathbf{X}$ has a density or pmf $p(\mathbf{x})$, so that $\mathbb{E}_{\mathbf{X}_{\mathcal{S}^c} \mid \mathbf{x}_{\mathcal{S}}}[\Delta]$ is a well defined real number. By our assumption that $D$ is nonnegative in Section 2, we have $\Delta \geq 0$. Sufficiency of $\mathcal{S}$ implies $\mathbb{E}_{\mathbf{X}_{\mathcal{S}^c} \mid \mathbf{x}_{\mathcal{S}}}[\Delta] = 0$. Thus $\Pr(\Delta = 0 \mid \mathbf{X}_{\mathcal{S}} = \mathbf{x}_{\mathcal{S}}) = 1$ (Chung, 2001, Exercise 3.2.1, p. 46). Recall from Section 2 that $D(\mathbf{u}, \mathbf{v}) = 0 \iff \mathbf{u} = \mathbf{v}$, so that

$$\Pr\left(D(f(\mathbf{X}_{\mathcal{S}^c}, \mathbf{x}_{\mathcal{S}}), f(\mathbf{x})) = 0 \mid \mathbf{X}_{\mathcal{S}} = \mathbf{x}_{\mathcal{S}}\right) = \Pr\left(f(\mathbf{X}_{\mathcal{S}^c}, \mathbf{x}_{\mathcal{S}}) = f(\mathbf{x}) \mid \mathbf{X}_{\mathcal{S}} = \mathbf{x}_{\mathcal{S}}\right) = 1.$$

By hypothesis, $f_y(\mathbf{x}) = p(y \mid \mathbf{x})$ for all $y \in \mathcal{Y}$. Thus, taking $\mathbf{X}_{\mathcal{S}^c} \sim \mathbb{P}_{\mathbf{X}_{\mathcal{S}^c} \mid \mathbf{X}_{\mathcal{S}} = \mathbf{x}_{\mathcal{S}}}$,

$$p(y \mid \mathbf{x}) \stackrel{\text{a.s.}}{=} p(y \mid \mathbf{X}_{\mathcal{S}^c}, \mathbf{x}_{\mathcal{S}}) \quad \forall y \in \mathcal{Y}. \qquad \square$$

# B   Proof of Proposition 1

First, we will need the following simple lemma.

**Lemma 2.** *Suppose that $\mathcal{X} = \mathbb{R}$ and that $\mathbf{X}$ admits a continuous, strictly positive density, i.e., $p(\mathbf{x}) > 0$ for all $\mathbf{x} \in \mathbb{R}^d$. Further suppose that $f$ and $D$ are continuous. Then the set $\mathcal{S}$ is sufficient for $f$ at $\mathbf{x}$ if and only if $f(\mathbf{x}_{\mathcal{S}}, \mathbf{x}'_{\mathcal{S}^c}) = f(\mathbf{x})$ for all $\mathbf{x}'_{\mathcal{S}^c} \in \mathbb{R}^{|\mathcal{S}^c|}$.*

*Proof.* Showing that $f(\mathbf{x}_{\mathcal{S}}, \mathbf{x}'_{\mathcal{S}^c}) = f(\mathbf{x})$ for all $\mathbf{x}'_{\mathcal{S}^c} \in \mathbb{R}^{|\mathcal{S}^c|}$ implies $\mathcal{S}$ is sufficient for $f$ at $\mathbf{x}$ is trivial.

For the other direction, we will use continuity of $f$ and $D$ to first show that $D(f(\mathbf{x}'_{\mathcal{S}^c}, \mathbf{x}_{\mathcal{S}}), f(\mathbf{x})) = 0$ for all $\mathbf{x}'_{\mathcal{S}^c}$. From the assumption of sufficiency of $\mathcal{S}$ for $f$ at $\mathbf{x}$, we have

$$\mathbb{E}_{\mathbf{X}_{\mathcal{S}^c} \mid \mathbf{x}_{\mathcal{S}}}[D(f(\mathbf{X}_{\mathcal{S}^c}, \mathbf{x}_{\mathcal{S}}), f(\mathbf{x}))] = 0.$$

By the assumption of a positive, continuous density $p(\mathbf{x})$, we have

$$\int_{\mathbb{R}^{|\mathcal{S}^c|}} p(\mathbf{x}'_{\mathcal{S}^c} \mid \mathbf{x}_{\mathcal{S}}) D(f(\mathbf{x}'_{\mathcal{S}^c}, \mathbf{x}_{\mathcal{S}}), f(\mathbf{x})) \, \mathrm{d}\mathbf{x}'_{\mathcal{S}^c} = \int_{\mathbb{R}^{|\mathcal{S}^c|}} \frac{p(\mathbf{x}'_{\mathcal{S}^c}, \mathbf{x}_{\mathcal{S}})}{p(\mathbf{x}_{\mathcal{S}})} D(f(\mathbf{x}'_{\mathcal{S}^c}, \mathbf{x}_{\mathcal{S}}), f(\mathbf{x})) \, \mathrm{d}\mathbf{x}'_{\mathcal{S}^c} = 0$$

$$\implies \int_{\mathbb{R}^{|\mathcal{S}^c|}} \underbrace{p(\mathbf{x}'_{\mathcal{S}^c}, \mathbf{x}_{\mathcal{S}}) D(f(\mathbf{x}'_{\mathcal{S}^c}, \mathbf{x}_{\mathcal{S}}), f(\mathbf{x}))}_{\doteq \, \psi(\mathbf{x}'_{\mathcal{S}^c})} \, \mathrm{d}\mathbf{x}'_{\mathcal{S}^c} = 0.$$

Note that $\psi$ is continuous, since $\mathbf{x}'_{\mathcal{S}^c} \mapsto D(f(\mathbf{x}'_{\mathcal{S}^c}, \mathbf{x}_{\mathcal{S}}), f(\mathbf{x}))$ and $\mathbf{x}'_{\mathcal{S}^c} \mapsto p(\mathbf{x}'_{\mathcal{S}^c}, \mathbf{x}_{\mathcal{S}})$ are continuous. We have

$$\int_{\mathbb{R}^{|\mathcal{S}^c|}} \psi(\mathbf{z}) \, \mathrm{d}\mathbf{z} = 0.$$

By continuity and nonnegativity of $\psi$, this implies $\psi(\mathbf{z}) = 0$ for all $\mathbf{z} \in \mathbb{R}^{|\mathcal{S}^c|}$.[14] Therefore, for all $\mathbf{x}'_{\mathcal{S}^c} \in \mathbb{R}^{|\mathcal{S}^c|}$, we have

$$
\psi(\mathbf{x}'_{\mathcal{S}^c}) = p(\mathbf{x}'_{\mathcal{S}^c}, \mathbf{x}_{\mathcal{S}}) D(f(\mathbf{x}'_{\mathcal{S}^c}, \mathbf{x}_{\mathcal{S}}), f(\mathbf{x})) = 0
$$
$$
\implies D(f(\mathbf{x}'_{\mathcal{S}^c}, \mathbf{x}_{\mathcal{S}}), f(\mathbf{x})) = 0
$$
$$
\implies f(\mathbf{x}'_{\mathcal{S}^c}, \mathbf{x}_{\mathcal{S}}) = f(\mathbf{x}),
$$

where the last step follows from our assumption on $D$ that $D(\mathbf{u}, \mathbf{v}) = 0 \iff \mathbf{u} = \mathbf{v}$. $\qquad\square$

We are now ready to prove Proposition 1.

*Proof of Proposition 1.* If $\mathcal{S}$ is a MSS for $f$ at $\mathbf{x}$, then

$$
\mathop{\mathbb{E}}_{\mathbf{X}_{\mathcal{S}^c}|\mathbf{x}_{\mathcal{S}}} [D(f(\mathbf{X}_{\mathcal{S}^c}, \mathbf{x}_{\mathcal{S}}), f(\mathbf{x}))] = 0,
$$

and by Lemma 2, this occurs if and only if

$$
f(\mathbf{x}'_{\mathcal{S}^c}, \mathbf{x}_{\mathcal{S}}) = f(\mathbf{x}), \quad \forall \mathbf{x}'_{\mathcal{S}^c} \in \mathbb{R}^{|\mathcal{S}^c|}. \tag{8}
$$

We will define $\mathcal{S} \doteq \{i \in [d] : \mathbf{a}_i \neq \mathbf{0}\}$. We will prove the result in two parts.

Part 1: If $g$ is injective on $\mathrm{span}(A) + \mathbf{b}$, then the MSS is unique and equals the nonzero column indices of $A$. By Equation (8),

$$
f(\mathbf{x}'_{\mathcal{S}^c}, \mathbf{x}_{\mathcal{S}}) = f(\mathbf{x}), \quad \forall \mathbf{x}'_{\mathcal{S}^c} \in \mathbb{R}^{|\mathcal{S}^c|}
$$
$$
\iff g(A_{\mathcal{S}} \mathbf{x}_{\mathcal{S}} + A_{\mathcal{S}^c} \mathbf{x}'_{\mathcal{S}^c} + \mathbf{b}) = g(A\mathbf{x} + \mathbf{b}), \quad \forall \mathbf{x}'_{\mathcal{S}^c} \in \mathbb{R}^{|\mathcal{S}^c|}
$$
$$
\iff A_{\mathcal{S}} \mathbf{x}_{\mathcal{S}} + A_{\mathcal{S}^c} \mathbf{x}'_{\mathcal{S}^c} + \mathbf{b} = A_{\mathcal{S}} \mathbf{x}_{\mathcal{S}} + A_{\mathcal{S}^c} \mathbf{x}_{\mathcal{S}^c} + \mathbf{b}, \quad \forall \mathbf{x}'_{\mathcal{S}^c} \in \mathbb{R}^{|\mathcal{S}^c|}
$$
$$
\iff A_{\mathcal{S}^c} (\mathbf{x}'_{\mathcal{S}^c} - \mathbf{x}_{\mathcal{S}^c}) = \mathbf{0}, \quad \forall \mathbf{x}'_{\mathcal{S}^c} \in \mathbb{R}^{|\mathcal{S}^c|}
$$
$$
\iff A_{\mathcal{S}^c} \mathbf{u} = \mathbf{0}, \quad \forall \mathbf{u} \in \mathbb{R}^{|\mathcal{S}^c|}
$$
$$
\iff A_{\mathcal{S}^c} = \mathbf{0}.
$$

Therefore, $\mathcal{S}^c$ is the unique largest set for which Equation (8) holds: any superset thereof is not the complement of a sufficient set. Conclude that $\mathcal{S}$ is the unique MSS.

Part 2: If $g = \mathrm{softmax}$, then the MSS corresponds to the indices of columns of $A$ not proportional to $\mathbf{1}$. We note that for any $c \in \mathbb{R}$, and any $\mathbf{z} \in \mathbb{R}^d$, $g(\mathbf{z} + c\mathbf{1}) = g(\mathbf{z})$. Denote $Q \in \mathbb{R}^{d \times d-1}$ a matrix with orthonormal columns forming a basis for $\mathrm{span}\{\mathbf{1}\}^\perp$. It follows that $I - QQ^\top$ is an orthonormal projection onto $\mathrm{span}\{\mathbf{1}\}$, and therefore for any $\mathbf{z} \in \mathbb{R}^d$, there exists $c$ such that $(I - QQ^\top)\mathbf{z} = c\mathbf{1}$. Hence

$$
g(A\mathbf{x} + \mathbf{b}) = g\big(QQ^\top(A\mathbf{x} + \mathbf{b}) + (I - QQ^\top)(A\mathbf{x} + \mathbf{b})\big) = g\big(QQ^\top(A\mathbf{x} + \mathbf{b})\big) = \widetilde{g}\big(\widetilde{A}\mathbf{x} + \widetilde{b}\big),
$$

where $\widetilde{g}(\mathbf{z}) = g(Q\mathbf{z})$, $\widetilde{A} = Q^\top A$, and $\widetilde{b} = Q^\top \mathbf{b}$. We will now show that $\widetilde{g}$ admits a left inverse, and is thus injective. By construction, $Q$ is injective as a linear map, so it suffices to show that there exists a map $h$ such that for any $\mathbf{q} \perp \mathbf{1}$, $(h \circ g)(\mathbf{q}) = \mathbf{q}$, demonstrating injectivity of $g$ on the space orthogonal to $\mathbf{1}$. By construction as an orthogonal projection, $QQ^\top \mathbf{q} = \mathbf{q}$. Denote by $\log(\mathbf{x})_i = \log(x_i)$ the elementwise natural logarithm, and set $\beta \doteq \sum_{i \in [d]} \exp(q_i)$. Observe that

$$
\log(g(\mathbf{q}))_i = \log(\exp(q_i)/\beta) = q_i - \log(\beta),
$$

and so, $QQ^\top \log(g(\mathbf{q})) = QQ^\top(\mathbf{q} - \log(\beta)\mathbf{1}) = \mathbf{q}$. Hence $h(\mathbf{v}) = QQ^\top \log(\mathbf{v})$ is a left inverse of $g|_{\mathrm{span}\{\mathbf{1}\}^\perp}$. Therefore, $\widetilde{g}$ is a continuous and injective nonlinearity. From part 1 of the proof, it follows that a MSS for $f(\mathbf{x}) = g(A\mathbf{x} + \mathbf{b}) = \widetilde{g}(\widetilde{A}\mathbf{x} + \widetilde{b})$ is the set of nonzero column indices of $\widetilde{A}$, which are precisely the $i \in [d]$ such that $Q^\top \mathbf{a}_i \neq \mathbf{0}$, i.e., the set of $i \in [d]$ such that $\mathbf{a}_i$ is not a multiple of $\mathbf{1}$, $\mathcal{S} = \{i \in [d] : \forall c \in \mathbb{R}, \mathbf{a}_i \neq c\mathbf{1}\}$. $\qquad\square$

---

[14] The one-dimensional version of this statement is a common exercise in real analysis (see, e.g., (Rudin, 1976, Chapter 6, Exercise 2)).

## C   Proofs of Lemma 1 and Proposition 2

*Proof of Lemma 1.* First, it is trivial to see that if $N \leq d$ the statement is true. Now consider the case where $N > d$. Take the size of the active feature pool $|\mathcal{A}|$ as a progress measure. The sufficient $\mathcal{S}$ found by the MSO at each iteration satisfies $\mathcal{S} \cap \mathcal{A}^c = \emptyset$. Furthermore, by Definition 1, the empty set is not sufficient and thus $\mathcal{S} \neq \emptyset$. At each iteration, $\mathcal{S}$ is removed from $\mathcal{A}$. Therefore $|\mathcal{A}|$ decreases by at least one at each iteration, and LMEA runs for at most $d < \infty$ iterations. □

*Proof of Proposition 2.* By Lemma 1, LMEA terminates, so a set of explanations $\mathcal{E}$ is returned. Seeking contradiction, suppose that there is some $\varepsilon$-MSS $\mathcal{S}$ such that for all $\mathcal{S}' \in \mathcal{E}$, $\mathcal{S}' \cap \mathcal{S} = \emptyset$. Then $\mathcal{S} \subseteq \mathcal{A}$ in the final iteration of LMEA, since $\mathcal{A} = [d] \setminus \mathcal{U}$. With $N = d$, there are two possibilities: either (i) LMEA ran for $d$ iterations before terminating, or (ii) the MSO returned $\emptyset$ in the last iteration executed. In case (i), $\mathcal{U} = [d]$, so that $\mathcal{A} = [d] \setminus [d] = \emptyset$ which implies $\mathcal{S} \subseteq \mathcal{A} = \emptyset$. By Definition 1, sufficient sets cannot be empty, so we have reached a contradiction. In case (ii), we also have a contradiction: by Assumption 1 and Definition 1, the MSO returns $\emptyset$ if and only if $\mathcal{A}$ contains no sufficient sets. □

## D   Additional synthetic experiments

### D.1   Synthetic experiment: recovery of overlapping MSSs

In settings where MSSs are disjoint, LMEA will identify them all. However, in many real-world scenarios, MSSs overlap. Thus, we conduct a synthetic experiment to study LMEA's performance when MSSs overlap. Let $\mathcal{G}$ denote the set of ground-truth MSSs for a model $f$, instance $\mathbf{x}$, sufficiency function $D$, and sufficiency level $\varepsilon$. We represent the overlaps between MSSs using an *intersection graph* $\Gamma = (\mathcal{V}, \mathcal{I})$ (see, e.g., (McKee & McMorris, 1999, Ch. 1)). Each vertex $v \in \mathcal{V}$ corresponds to a MSS $\mathcal{S}$, and an edge $(\mathcal{S}_i, \mathcal{S}_j) \in \mathcal{I}$ exists iff the corresponding MSSs intersect, i.e., $\mathcal{S}_i \cap \mathcal{S}_j \neq \emptyset$. With this setup, when LMEA is run on an instance $\mathbf{x}$ with an overlap pattern $\Gamma = (\mathcal{V}, \mathcal{I})$, removing an explanation $\mathcal{S}$ from the active set $\mathcal{A}$ corresponds to removing the node $v$ and its neighbors (along with corresponding edges) from the graph $\Gamma$ to obtain a new intersection graph $\Gamma'$. LMEA will repeat until an empty graph is reached and return a set $\mathcal{F} \subseteq \mathcal{G}$ of explanations found. The number of edges and the overall structure of $\Gamma$ characterizes how MSSs overlap and, in turn, influences LMEA's ability to recover the full set $\mathcal{G}$.

To mimic a realistic computer vision problem, we consider a random geometric graph (RGG) (see, e.g., (Penrose, 2003, Ch. 1)), which is an particular kind of intersection graph. We construct an instance $\mathbf{x}$ with $n$ MSSs, $\mathcal{S}_1, \ldots, \mathcal{S}_n$, each a disk in $[0, 1]^2$: $\mathcal{S}_i = \{u \in \mathbb{R}^2 : \|u - \mathbf{C}_i\|_2 < r\}$, $\mathbf{C}_i \overset{\text{i.i.d.}}{\sim} \mathcal{U}([0, 1]^2)$, where $r$ is the (common) radius. Therefore, given the radius $r$ and centers $\{\mathbf{C}_i\}_{i=1}^n$, each vertex in vertex set $\mathcal{V}$ of the intersection graph $\Gamma = (\mathcal{V}, \mathcal{I})$ corresponds to a circle and $\mathcal{I} = \{(\mathcal{S}_i, \mathcal{S}_j) : i \neq j, \|\mathbf{C}_i - \mathbf{C}_j\|_2 < 2r\}$, so that an edge indicates that the two disks intersect. Examples of an instance $\mathbf{x}$ for two different $r$ are shown in Figure 5.

We simulate LMEA and track the percentage of MSSs recovered for $n \in [20]$ and $r \in \{0.05, 0.06, \ldots, 0.25\}$. For each setting, we run LMEA on 5,000 RGG instances. At each step, LMEA picks a MSS from the active set $\mathcal{A}$ uniformly at random. As both $n$ and $r$ increase, the number and degree of overlaps between explanations increases. As expected, LMEA recovers a smaller fraction of the true explanation set $\mathcal{G}$. The results, pictured in Figure 6, depict a graceful degradation in the number of recovered explanations as $n$ and $r$ (and thus the degree of overlap between MSSs) increase.

### D.2   Synthetic experiment: breaking Assumption 1

Here we conduct our final synthetic experiment that evaluates how violations of Assumption 1 impact LMEA's ability to recover the full set of true MSSs $\mathcal{G}$, as defined in Appendix D.1. We re-use the RGG setup in Appendix D.1 and consider two possible failure modes of an MSO being used in LMEA: (i) early termination and (ii) selection of spurious MSSs. Failure mode (i) corresponds to the MSO failing to find an MSS in the active set $\mathcal{A}$, while failure mode (ii) corresponds to the MSO finding an MSS that is not one of ground truth MSSs in $\mathcal{G}$. Such spurious MSS can often arise due to, e.g., off-manifold perturbations used by

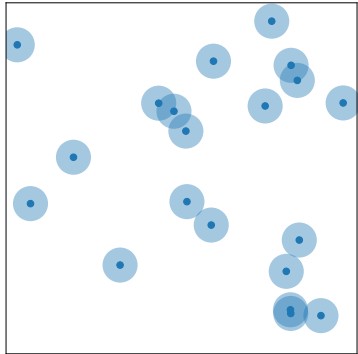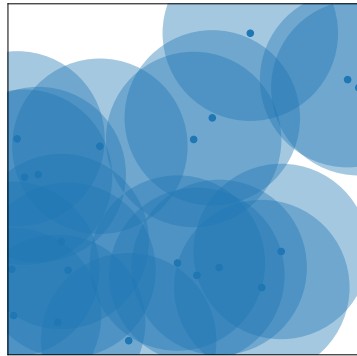

Figure 5: Two extremes of the random geometric graph (RGG) setup described in Appendix D.1 with $n = 20$ circular regions. Each circle represents an MSS. **Left**: $r = 0.05$. **Right**: $r = 0.25$. As the radius increases, more regions intersect, and the number of regions involved in these intersections increases.

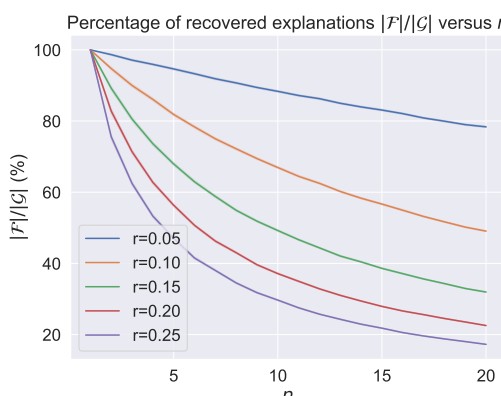

Figure 6: Recovered explanation percentage $100 \frac{|\mathcal{F}|}{|\mathcal{G}|}$ for the synthetic MSS overlap experiment described in Appendix D.1. The pictured quantities are averaged over 5,000 trials of each experiment.

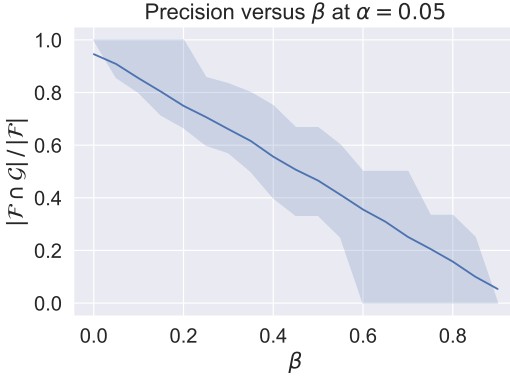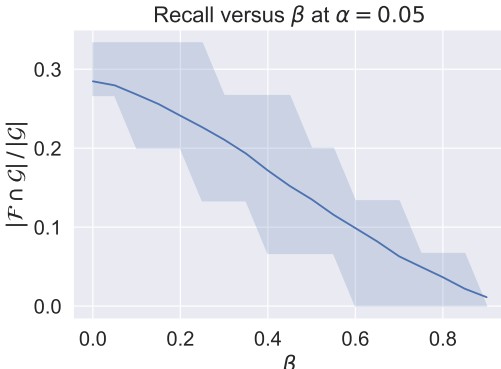

Figure 7: Precision and recall versus $\beta$ for the oracle quality ablation experiment described in Appendix D.2. The lines represent the mean precision over 5,000 trials of each setting, while shaded regions represent the 0.25-0.75 inter-quartile range.

the MSO to find explanations. At each step of the LMEA simulation as described in Appendix D.1, we pick from three possible MSO behaviors:

(a) The MSO fails to find any MSS and terminates early with probability $\alpha$,

(b) The MSO finds a spurious MSS with probability $\beta$, and

(c) The MSO finds a correct MSS in $\mathcal{G}$ with probability $1 - \alpha - \beta$.

Under this setup, $\beta$ quantifies the degree of violation of Assumption 1, with larger $\beta$ indicating a higher degree of violation (via more frequent selection of spurious explanations).

For an instance $\mathbf{x}$ with associated RGG with $n$ vertices, we partition $\{\mathcal{S}_i\}_{i=1}^n = \mathcal{G} \cup \mathcal{B}$ into two sets: $\mathcal{G}$ (the set of "good," or true, MSSs) and $\mathcal{B}$ (the set of "bad," or spurious, MSSs). We simulate LMEA on the intersection graph $\Gamma$ formed by $\{\mathcal{S}_i\}_{i=1}^n$ and their overlaps. If the MSO enters case (b) and there are no remaining spurious explanations, then LMEA terminates. Likewise, if the MSO enters case (c) and there are no remaining true MSSs, then LMEA terminates.

We run this experiment on an RGG with $n = 20$, $r = 0.15$, and $\alpha = 0.05$, sweeping the spurious MSS selection probability $\beta \in \{0.05, 0.1, \ldots, 0.9\}$. We set $|\mathcal{G}| = 15$ and $|\mathcal{B}| = 5$, and track the precision and recall of LMEA's returned set of explanations $\mathcal{F}$ (introduced in Appendix D.1) with respect to the set of true MSSs $\mathcal{G}$. The results are pictured in Figure 7. We see a gradual decrease in the quality of the MSSs recovered by LMEA. As $\beta \to 1$ (meaning LMEA is more likely to find a spurious MSS at each iteration), precision decreases, meaning that a higher fraction of the MSSs found by LMEA are spurious. Similarly, as $\beta \to 1$, the recall decreases, meaning that LMEA finds a smaller subset $\mathcal{E}$ of the true MSSs $\mathcal{G}$. In conclusion, while Assumption 1 is strong, this experiment reveals that LMEA's performance in practice degrades gracefully as the degree of violation of Assumption 1 increases.

# E  Baseline hyperparameters

## E.1  MultiReX: background and hyperparameters

MultiReX operates on an objective landscape defined by a ranking $\phi$ of each pixel's causal contribution toward the classification. This causal responsibility ranking is computed via an iterative refinement procedure with number of refinement steps $n_{\text{iter}}$. Given $\phi$, MultiReX first finds a global explanation $\mathcal{T}_0$, before proceeding to a stochastic local search procedure for further explanations.

Table 4: MultiReX hyperparameter settings used in the Shapes, Malaria, and CelebAMask-HQ experiments. The MultiReX package's default values for each hyperparameter are shown in parentheses. A $\uparrow$ indicates that a larger value is more generous, while a $\downarrow$ indicates that a smaller value is more generous.

| $s$ $\uparrow$ | $n_{\text{step}}$ $\uparrow$ | $r$ $\downarrow$ | $\alpha$ $\downarrow$ | $n_{\text{exp}}$ $\uparrow$ | $\tau$ $\uparrow$ | $n_{\text{iter}}$ $\uparrow$ |
|---|---|---|---|---|---|---|
| 20 (10) | 80 (40) | 5 (25) | 0.2 (0.2) | 25 (4) | 0.1 (0.0) | 100 (20) |

For this local search, MultiReX initializes a number $s$ of circles $\mathcal{C}_i \subseteq [d]$, $i \in [s]$, each of radius $r$, which serve as initial candidate explanations to be refined. We will first describe the execution of MultiReX for a single circle $\mathcal{C}_i$. After initializing $\mathcal{C}_i$, MultiReX checks whether it is sufficient. If not, then $\mathcal{C}_i$ is expanded (its radius increased) by a factor of $1 + \alpha$, and its sufficiency is re-checked. If the expanded circle $(1+\alpha)\mathcal{C}_i$ is sufficient, then it is added to a set of candidate explanations, and the search for circle $i$ terminates. Otherwise, this process is repeated a maximum of $n_{\text{exp}}$ times. If after all of these expansions, $\mathcal{C}_i$ is still not sufficient, it is re-initialized in a neighboring position via an accept-reject method that moves to areas with higher mean responsibility $\frac{1}{|\mathcal{C}_i|} \sum_{i=1}^{|\mathcal{C}_i|} \phi_i$. This process is repeated until a sufficient $\mathcal{C}_i$ is found, or a maximum number of steps $n_{\text{step}}$ is reached. If the final $\mathcal{C}_i$ is sufficient, then it is pruned to an explanation $\mathcal{T}_i$. The aforementioned procedure is executed for all $\mathcal{C}_i$, leading to a set of at most $s + 1$ candidate explanations (including the global explanation) $\{\mathcal{T}_i\}_{i=0}^{k}$, $k \leq s$. Finally, this set of candidate explanations is pruned to minimize overlap; the only hyperparameter for this step is the allowed Sørensen-Dice coefficient (Dice, 1945; Sørensen, 1948) threshold $\tau$ between any two explanations. The output of this final stage is a set of MSSs $\{\mathcal{S}_i\}_{i=0}^{k'}$, $k' \leq s$.

The implementation of MultiReX[15] allows the configuration of many hyperparameters. We set each of these to increase the computational budget over the default configuration[16] for a generous comparison. Table 4 reports the hyperparameters used in our experiments, which are held fixed across datasets. We note that, due to rounding in the implementation of the circle expansion step, $\alpha$ is the smallest allowable radius expansion parameter for $r = 5$. MultiReX returns multiple sets of explanations satisfying the overlap bound $\tau$. We pick the one that has the smallest total area, which yields the smallest sparsity metric in our experiments. For the experiments in Section 6, we use MultiReX's default masking value of zero.

### E.2 Beam search: background and hyperparameters

The method proposed by Shitole et al. (2021) finds multiple MSSs in an image via beam search (BS) over a $r \times r$ grid of image patches. To rank beam candidates, the algorithm pre-computes a saliency map $I \in [0,1]^{r \times r}$. The beam search procedure has multiple hyperparameters: the beam width $B \in \mathbb{N}$, the number of classes to be explained $c \in \mathbb{N}$, the maximum allowable number of overlapping patches between MSSs $P \in \mathbb{N}$, the maximum number of patches allowed in a single MSS $M \in \mathbb{N}$, maximum number of explanations $E \in \mathbb{N}$, and the number of candidates to consider in each iteration of beam search $q \in \mathbb{N}$, which must satisfy $q \geq B$. BS is capable of explaining the top $c$ classes predicted by the model $f$; for fair comparison with LMEA and MultiReX we set $c = 1$, which generates explanations for the top class only. For a generous comparison, we set each of these parameters to their maximum "suggested value" in the public implementation,[17] with the exception of $E$, which we set to 20 for consistency with our other baselines. These values are listed in Table 5. To ensure a fair comparison with LMEA and MultiReX, we set BS's probability threshold parameter to ensure that each MSS output by BS is such that the model's estimated for the probability of the "positive" class satisfies $f_{y^\star}(\widetilde{\mathbf{x}}_{\mathbf{s}}) \geq 1/2$, where $y^\star = 1 = \arg\max_y f_y(\widetilde{\mathbf{x}}_{\mathbf{s}})$ for all images $\mathbf{x}$ in the experiments (which as described in Section 6.4, are positive samples with estimated probabilities $f_{y^\star}(\widetilde{\mathbf{x}}_{\mathbf{s}}) \geq 0.8$), $\widetilde{\mathbf{x}}_{\mathbf{s}} = \mathbf{s} \odot \mathbf{x} + (\mathbf{1} - \mathbf{s}) \odot \mathbf{b}$ is the perturbed input for the MSS mask $\mathbf{s}$ output by BS, and $\mathbf{b}$ is the BS perturbation background.

Shitole et al. (2021) use $r = 7$ in their paper and in their public implementation. We leave hyperparameters related to computing the saliency map $I$ at their default values, and we use their Gaussian blur option for

---

[15]https://github.com/ReX-XAI/ReX
[16]The default arguments reported are those in this file.
[17]https://github.com/viv92/structured-attention-graphs. Defaults are those reported in this file.

the background $\mathbf{b}$ for the experiments in Section 6. With $r = 7$, the maximal value of $M$ is $r^2 = 49$, which we leave unmodified. Setting $P \geq 1$ results in MSSs that can overlap substantially, which would be unfair to MultiReX and LMEA in our experiments, so we set $P = 0$. This value approximates a requirement of approximately disjoint MSSs; however, we observe that some features may be shared among explanations due to BS's upsampling $7 \times 7$ grids to the original $224 \times 224$ image size (which introduces bleed between patch regions).

Table 5: BS hyperparameters used in CelebAMask-HQ experiments. The public implementation's default values for each hyperparameter are shown in parentheses. A $\uparrow$ indicates that a larger value is more generous, while a $\downarrow$ indicates that a smaller value is more generous. Starred values indicate that setting that hyperparameter differently is either nontrivial to implement, impossible given other hyperparameter values, or would result in an unfair comparison.

| $r \downarrow$ | $B \uparrow$ | $c \uparrow$ | $P \uparrow$ | $E \uparrow$ | $M \uparrow$ | $q \uparrow$ |
|---|---|---|---|---|---|---|
| 7* (7) | 15 (3) | 1* (1) | 0* (1) | 20 (10) | 49* (49) | 15 (15) |

# F   Further examples

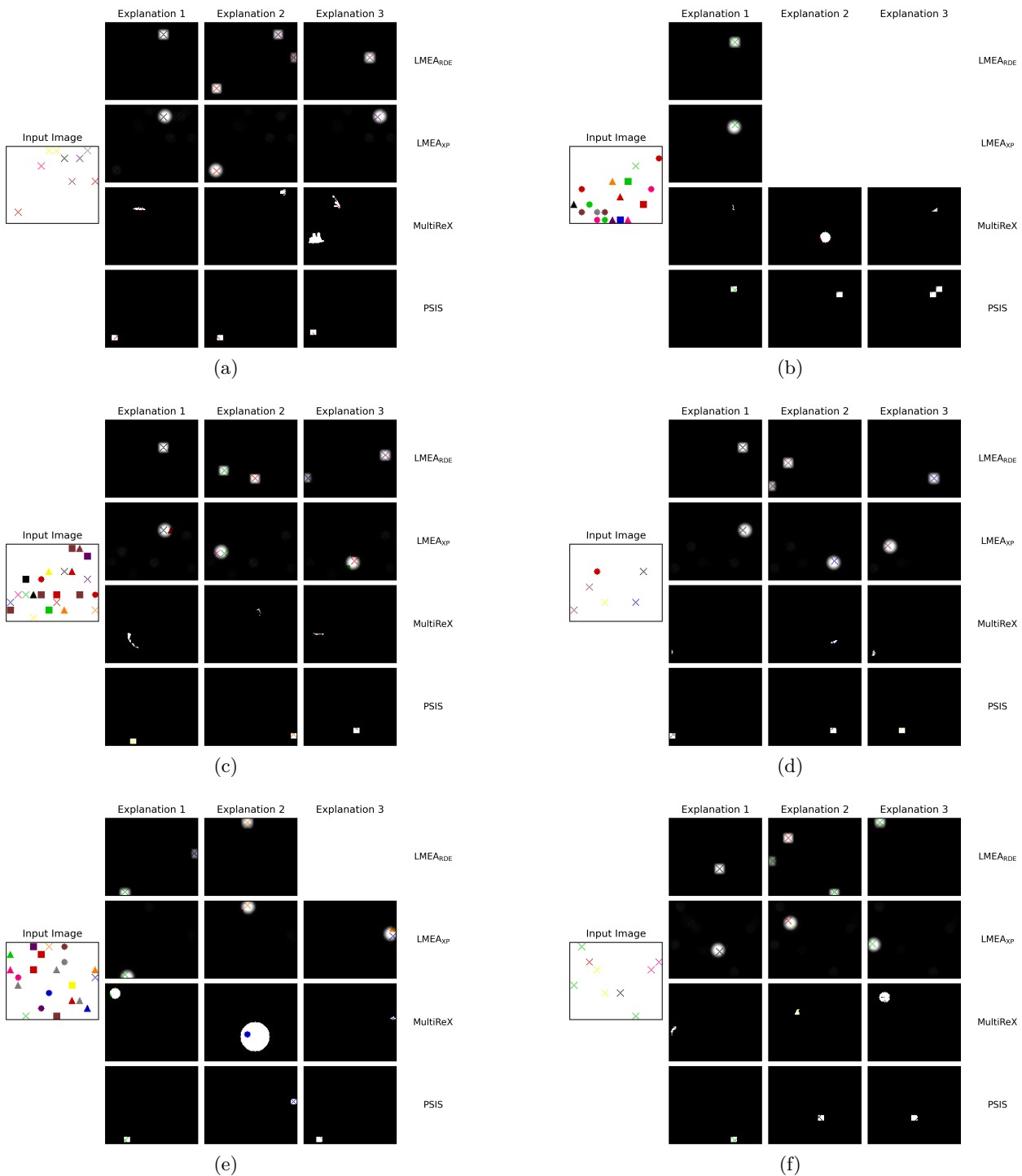

Figure 8: Randomly sampled runs of LMEA and baselines without SBP on images from the shapes dataset, where the model is trained to predict the presence of at least one "X". Black regions are unselected by each MEM. *Top row of each subfigure:* LMEA with RDE MSO. *Middle row of each subfigure:* LMEA with XP MSO. *Bottom two rows of each subfigure:* MultiReX and PSIS.

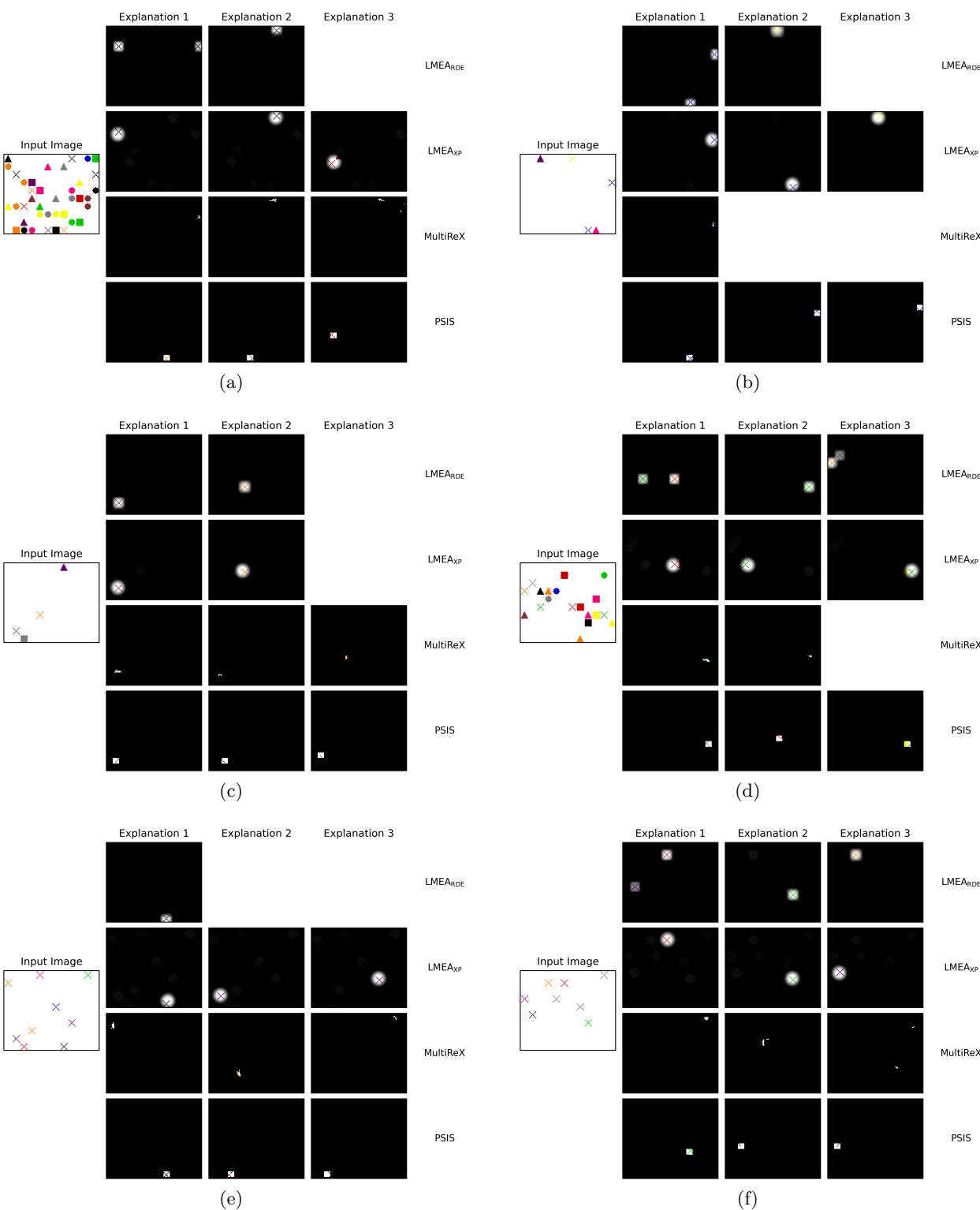

Figure 9: Randomly sampled runs of LMEA and baselines with SBP on images from the shapes dataset, where the model is trained to predict the presence of at least one "X". Black regions are unselected by each MEM. *Top row of each subfigure:* LMEA with RDE MSO. *Middle row of each subfigure:* LMEA with XP MSO. *Bottom two rows of each subfigure:* MultiReX and PSIS.

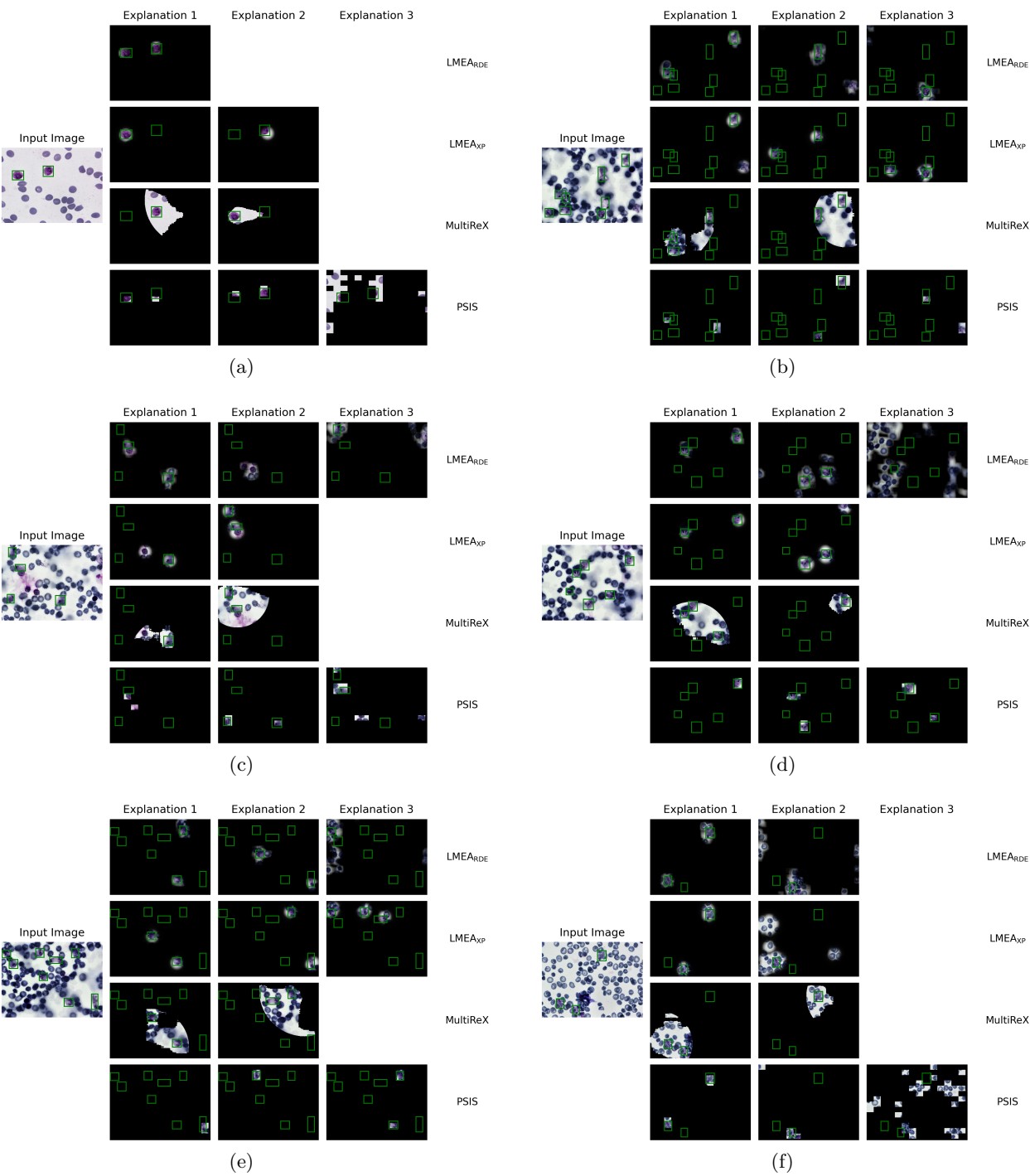

Figure 10: Randomly sampled runs of LMEA and baselines without SBP on images from the BBBC041 dataset, where the classifier is trained to predict the presence of at least one trophozoite in the image. Green squares indicate the bounding box labels corresponding to infected (trophozoite) cells. *Top row of each subfigure:* LMEA with RDE MSO. *Middle row of each subfigure:* LMEA with XP MSO. *Bottom two rows of each subfigure:* MultiReX and PSIS.

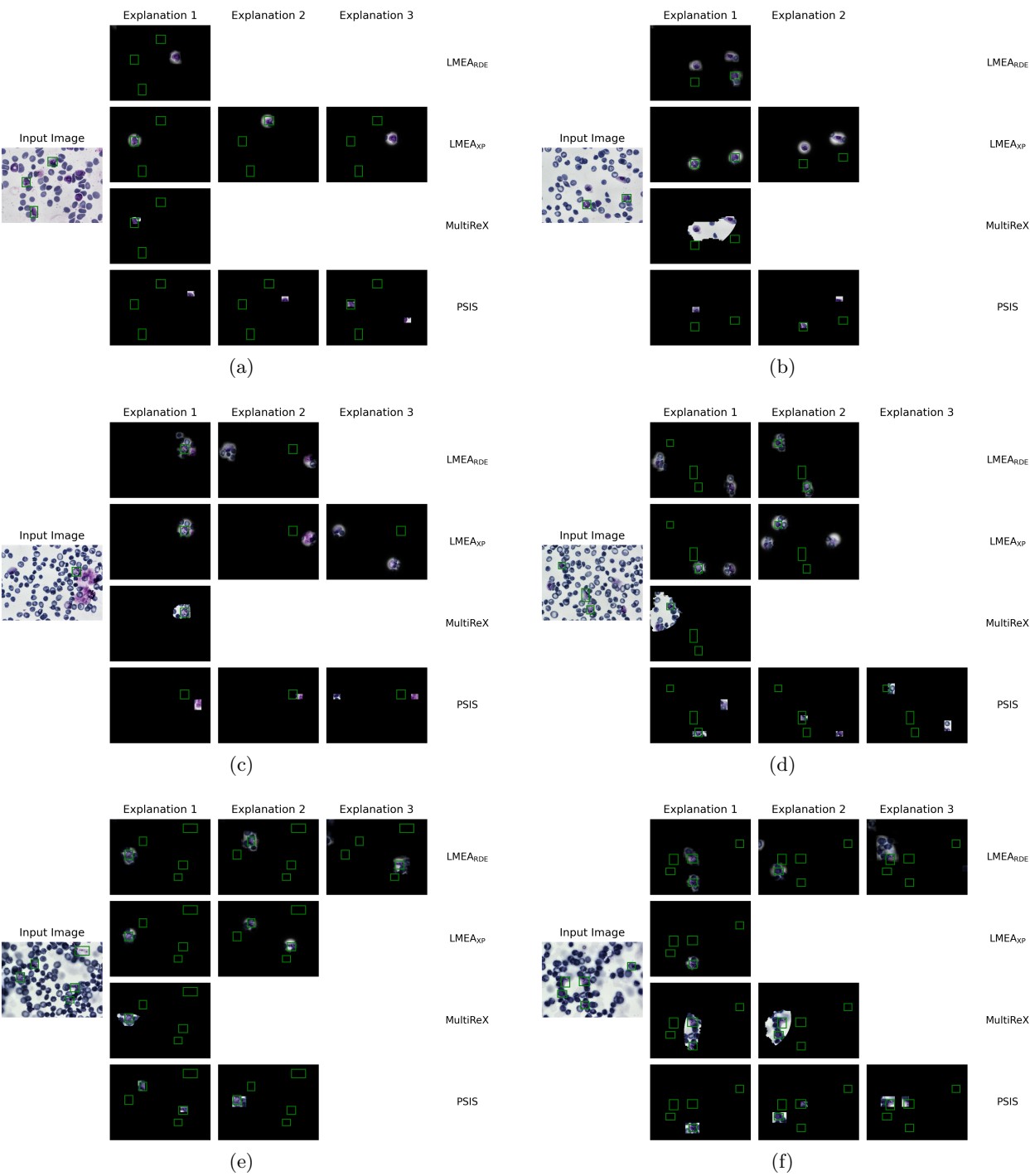

Figure 11: Randomly sampled runs of LMEA and baselines with SBP on images from the BBBC041 dataset, where the classifier is trained to predict the presence of at least one trophozoite in the image. Green squares indicate the bounding box labels corresponding to infected (trophozoite) cells. *Top row of each subfigure:* LMEA with RDE MSO. *Middle row of each subfigure:* LMEA with XP MSO. *Bottom two rows of each subfigure:* MultiReX and PSIS.

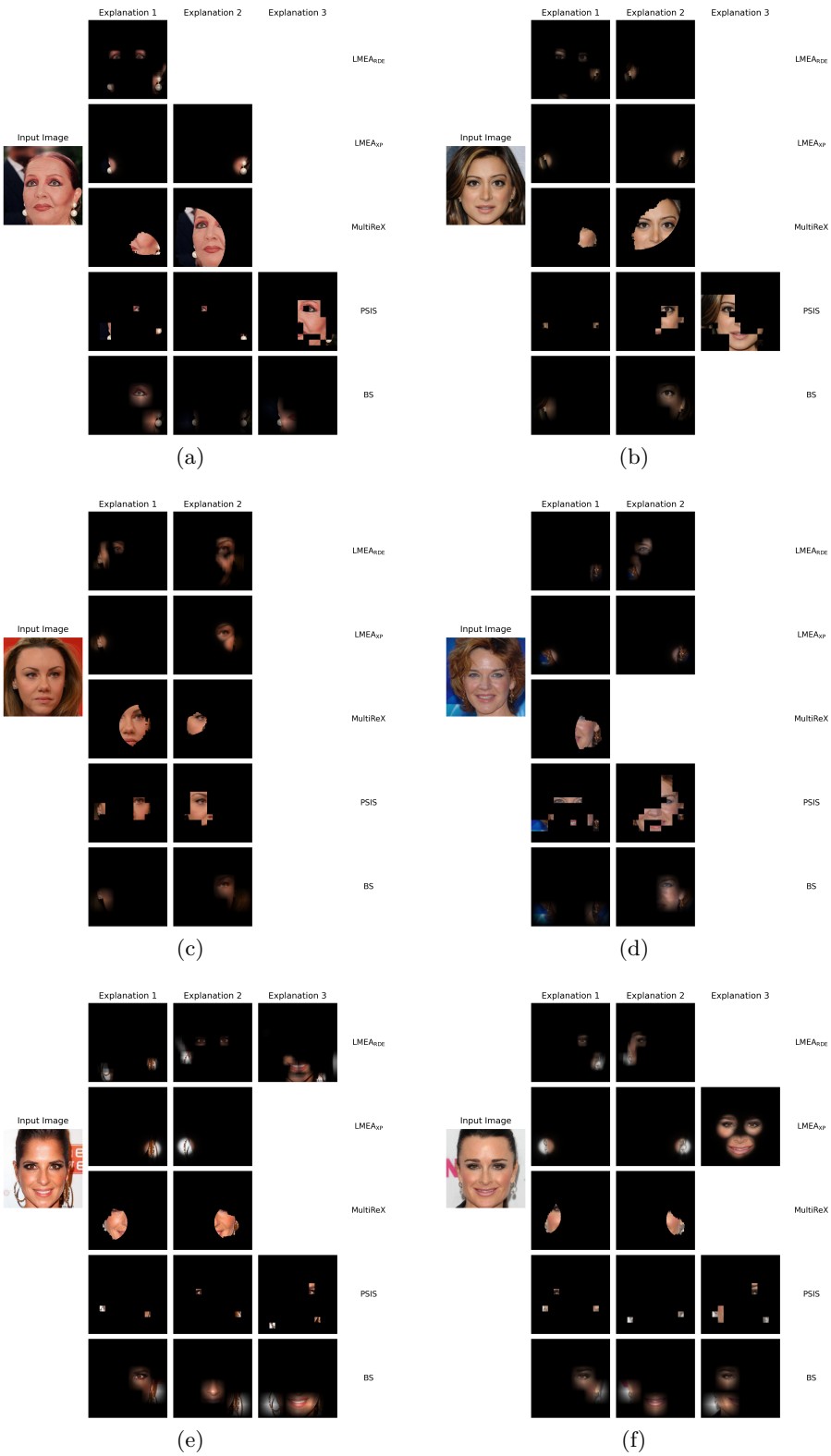

Figure 12: Randomly sampled runs of LMEA and baselines without SBP on images from the CelebAMask-HQ dataset, where the classifier is trained to predict the presence of the "wearing earrings" label. *Top row of each subfigure:* LMEA with RDE MSO. *Middle row of each subfigure:* LMEA with XP MSO. *Bottom three rows of each subfigure:* MultiReX, PSIS, and BS.

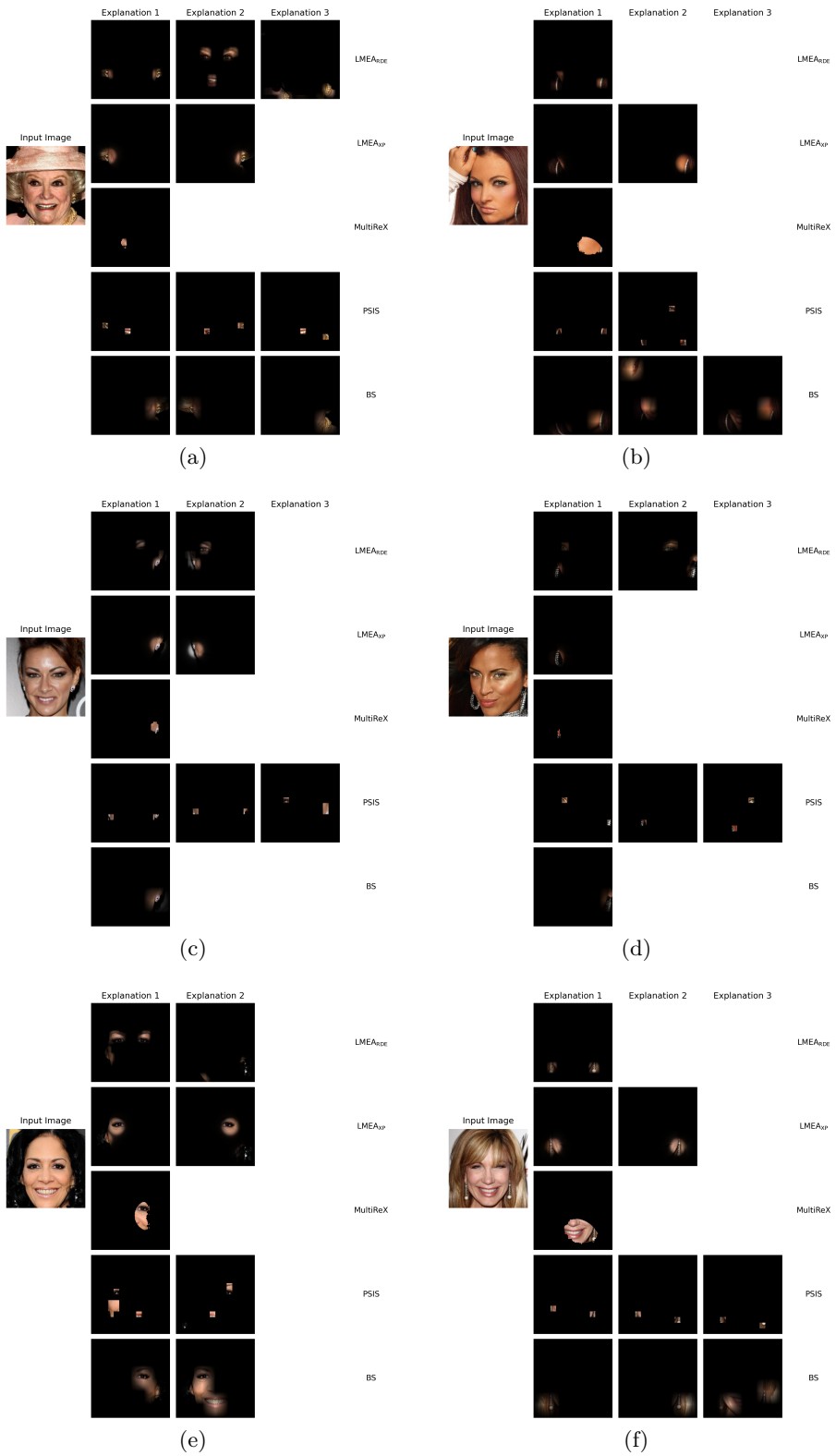

Figure 13: Randomly sampled runs of LMEA and baselines with SBP on images from the CelebAMask-HQ dataset, where the classifier is trained to predict the presence of the "wearing earrings" label. *Top row of each subfigure:* LMEA with RDE MSO. *Middle row of each subfigure:* LMEA with XP MSO. *Bottom two rows of each subfigure:* MultiReX, PSIS, and BS.

# G   Generation of example figures

To generate the examples in Figures 2, 3, 4, 8, 9, 10, 11, 12, and 13 we used the following procedure. For each experiment, after generating explanations for each image in the test set, we filtered the set of candidate images to those that had at least one explanation for each method (to rule out atypical examples where one of the methods fails to find an explanation). Next, we filtered to those images for which at least one of $\{\text{LMEA}_{\text{RDE}}, \text{LMEA}_{\text{XP}}, \text{MultiReX}, \text{SIS}\}$ had at least two explanations. Finally, we took a random permutation of the images that remained and took the first seven images for each dataset for each of the figures mentioned above. We then picked representatives of these random sets for Figures 2, 3, and 4 in the main body of the paper. These original random sets, which were generated before experiments on the BS baseline were performed, were pinned and used to generate later figures that include the BS baseline, e.g., Figure 4.

# H   Experimental details

## H.1   Further LMEA implementation details

**Explanation binarization.** Both RDE and XP MSOs return soft masks $\mathbf{s} \in [0, 1]^d$. To convert these into MSSs $\mathcal{S}$, we threshold them according to Otsu's method (Otsu, 1979).

**Termination criterion.** All of our experiments involve explaining binary classifiers. Therefore, to recover the most explanations possible with LMEA, we dynamically set the sufficiency criterion to $\varepsilon = |f_1(\mathbf{x}) - 1/2|$, where $f_1(\mathbf{x})$ is the model's estimated probability that $Y = 1$ given $\mathbf{X} = \mathbf{x}$. After each call to the MSO, which yields a soft attribution mask $\mathbf{s} \in [0, 1]^d$, we additionally check the sufficiency of the solution according to

$$\|f(\mathbf{s} \odot \mathbf{x} + (\mathbf{1} - \mathbf{s}) \odot \mathbf{b}_{\text{suff}}) - f(\mathbf{x})\|_{\text{TV}} \leq \varepsilon,$$

where $\mathbf{b}_{\text{suff}}$ is a background value, which will be set to zero or the empirical mean image computed over the training set, depending on the dataset. These are standard choices of background inputs in the feature attribution literature (Fong et al., 2019; Teneggi et al., 2022). This setup causes LMEA to terminate once the predicted label changes: $1 = f(\mathbf{x}) \neq \text{argmax}_{y \in \mathcal{L}} f_y(\mathbf{s} \odot \mathbf{x} + (\mathbf{1} - \mathbf{s}) \odot \mathbf{b}_{\text{suff}})$. If the above condition does not hold, then LMEA exits early. A similar sufficiency check is performed on the active set using its binary characteristic mask vector $\mathbf{1}_{\mathcal{A}}$, as specified in Algorithm 1. We also terminate early whenever an explanation is found that overlaps with previously found explanations, as this tends to correspond to an absence of informative features in $\mathcal{A}$, and LMEA with the XP MSO may find MSSs that overlap with explanations from previous iterations.

**Masked stop-gradient.** The following simple Python wrapper, based on a similar trick in the Neural-Sort (Grover et al., 2019) repository (https://github.com/ermongroup/neuralsort) suffices to prevent re-selection of previously selected features in LMEA for RDE and XP MSOs.

---

**Listing 1** Masked stop-gradient wrapper.

```python
class MaskedStopGradient(nn.Module):
    def __init__(self, model, mask):
        super().__init__()
        self._model = model
        self._mask = mask

    def forward(self, x):
        return self._model((1 - self._mask) * x + (self._mask * x).detach())
```

---

The mask in Listing 1 represents the coordinates in $\mathcal{A}^c$ (`1 - active_set`) that should not be re-selected. Passing `MaskedStopGradient(model, 1 - active_set)` to the MSO means that gradients corresponding to $\mathcal{A}^c$ will not backpropagate, as described in Section 6.3.

**Dilation.** For details on the dilation operation used in our experiments, including hyperparameters, see Appendix I.

## H.2   Synthetic dataset

**Model training.** We trained the simple convolutional network described in section 6.5.1 using Adam (Kingma & Ba, 2015) with learning rate $10^{-3}$ and batch size of 64 for 40 epochs, keeping the best model checkpoint based on validation (cross-entropy) loss. The images are preprocessed during training by subtracting the background $\mathbf{b} = \mathbf{1}$ and dividing by the approximate channel-wise standard deviations $\widehat{\boldsymbol{\sigma}} = (0.1323, 0.165, 0.17)$.

**LMEA.** For LMEA sufficiency checks, we used the background value $\mathbf{b}_{\text{suff}} = \mathbf{0}$, described in Section 6.3, which due to feature normalization corresponds to the white background of the shapes images.

**RDE.** In RDE, we set the smoothmax temperature to $T = 1$. We set the perturbation distribution to $\mathcal{N}(\widehat{\boldsymbol{\mu}}, \frac{1}{8}\widehat{\boldsymbol{\Sigma}})$, where $\widehat{\boldsymbol{\mu}} = \frac{1}{N}\sum_{i=1}^{N}\mathbf{x}^{(i)}$ is the empirical mean image and $\widehat{\boldsymbol{\Sigma}}$ is diagonal with $\widehat{\Sigma}_{jj} = \frac{1}{N-1}\sum_{i=1}^{N}(x_j^{(i)} - \widehat{\mu}_j)^2$, i.e., $\widehat{\Sigma}$ is the per-pixel-and-channel empirical variance image computed over the set of training images $\{\mathbf{x}^{(i)}\}_{i=1}^{N}$. To obtain the RDE explanations, we solved Equation (4) on the grid $\lambda \in \{1, 5, 10, 15, 20\}$ via 3000 iterations of Adam with learning rate $10^{-3}$.

**XP.** We use the implementation of XP in the TorchRay package.[18], solving the "preservation" (Fong et al., 2019) (sufficiency) objective with a "fade-to-black" (Fong et al., 2019) (zero) background. Due to feature normalization, this corresponds to setting the unselected pixels to white in the corresponding shapes images. We use the contrastive reward $r(\mathbf{x}) \doteq a_1(\widetilde{\mathbf{x}}) - a_0(\widetilde{\mathbf{x}})$, where $a_1(\widetilde{\mathbf{x}})$ is the activation (logit) corresponding to class one for input $\widetilde{\mathbf{x}}$ and $a_0(\widetilde{\mathbf{x}})$ is the activation (logit) corresponding to class zero for perturbed input $\widetilde{\mathbf{x}}$. This is because

$$
\begin{aligned}
p_1 &= \frac{\exp(r(\widetilde{\mathbf{x}}) + a_0(\widetilde{\mathbf{x}}))}{\exp(a_0(\widetilde{\mathbf{x}})) + \exp(r(\widetilde{\mathbf{x}}) + a_0(\widetilde{\mathbf{x}}))} \\
&= \frac{\exp(r(\widetilde{\mathbf{x}}))}{1 + \exp(r(\widetilde{\mathbf{x}}))} \\
&= \sigma(r(\widetilde{\mathbf{x}})),
\end{aligned}
$$

where $\sigma$ is the sigmoid functon. Since $\sigma$ is monotonically increasing, maximizing $r$ over perturbed inputs $\widetilde{\mathbf{x}}$ is equivalent to maximizing the model's probability of the positive class $f_1(\widetilde{\mathbf{x}})$, which is also equivalent to minimizing the sufficiency objective $\|f(\widetilde{\mathbf{x}}) - \widehat{\mathbf{y}}_{\text{conf}}\|_{\text{TV}}$, where $\widehat{\mathbf{y}}_{\text{conf}} = (0, 1)$ is the perfectly confident positive prediction and $\|\cdot\|_{\text{TV}}$ denotes the TV norm between distributions.

We run the XP solver for 1500 iterations on the grid of area constraint parameters $\alpha \in \{0.05, 0.1, 0.15, 0.2\}$, check $\varepsilon$-sufficiency according to Section 6.3, and choose the $\varepsilon$-sufficient solution with the smallest $\alpha$. For this dataset, we set the mask smoothing parameter `sigma` to 7, and we also disable the jitter option. All the other parameters are left at their default values.

## H.3   BBBC041 dataset

**Model training.** We fine-tune the pretrained ResNet18 model via 30 epochs of Adam with a learning rate of $10^{-4}$, a batch size of 4, and a learning rate decay factor of 0.2 applied after each 10 epochs, similarly to Teneggi et al. (2022) (we train for five more epochs than Teneggi et al. (2022)). In addition to Teneggi et al. (2022)'s random horizontal flip augmentations, we use JPEG compression and random vertical flip augmentations during training. Due to heterogeneity between training, validation, and test sets, we had difficulty training a model to perform well on the test set provided by Teneggi et al. (2022).[19] As a result, we randomly re-split the training and validation subsets of the dataset into training, validation, and test sets. As in the shapes experiment, we keep the best-performing model checkpoint according to the validation loss.

---

[18]https://github.com/facebookresearch/TorchRay
[19]We use the trophozoite dataset provided by Teneggi et al. (2022): https://zenodo.org/records/5914342.

**LMEA.** For LMEA sufficiency checks, we used the background value $\mathbf{b}_{\mathrm{suff}} = \widehat{\boldsymbol{\mu}}$.

**RDE.** For this dataset, we set the perturbation distribution to $\mathcal{N}(\widehat{\boldsymbol{\mu}}, \frac{1}{4}\widehat{\boldsymbol{\Sigma}})$, where once again the mean and covariance are estimated from the training set. We use the same grid for the sparsity penalty as in Appendix H.2, and solve RDE for each $\lambda$ via 500 iterations of Adam with learning rate $10^{-2}$.

**XP.** For this dataset, we use the "blur" background, which is a blurred version of the original input image, and we leave the mask smoothing parameter `sigma` at its default value of 21. The rest of the details are the same as in Appendix H.2.

### H.4    CelebAMask-HQ dataset

**Model training.** The ImageNet-pretrained ResNet18 model was fine-tuned for 25 epochs of Adam using a learning rate of $10^{-4}$ and a batch size of 64. For consistency with the pretrained model's preprocessing pipeline, we resize each image to $256 \times 256$, center crop to $224 \times 224$, and normalize by subtracting $\widetilde{\boldsymbol{\mu}} = (0.485, 0.456, 0.406)$ from R, G, and B channels, respectively, and dividing by $\widetilde{\boldsymbol{\sigma}} = (0.229, 0.224, 0.225)$, again, channel-wise. We keep the best model checkpoint according to validation loss.

**LMEA.** For LMEA sufficiency checks, we used the background value $\mathbf{b}_{\mathrm{suff}} = \widehat{\boldsymbol{\mu}}$.

**RDE.** We set the perturbation distribution to $\mathcal{N}(\widehat{\boldsymbol{\mu}}, \frac{1}{4}\widehat{\boldsymbol{\Sigma}})$, with empirical (training set) mean and covariance, as before. We again use the same grid for the sparsity penalty as in Appendix H.2, and solve RDE for each $\lambda$ via 500 iterations of Adam with learning rate $10^{-2}$.

**XP.** We use the same parameters as in Appendix H.3.

## I    More detail on the dilation step

### I.1    Motivation behind the dilation step

The dilation step for removal in LMEA is motivated by the structure of objectives such as Equation 4, which consist of terms that encourage sufficiency of the solution (like RDE's expected distortion term), and terms that encourage sparsity of the resulting masks (like RDE's $\ell_1$ penalty). The goal of the model wrapper method described in Section 6.3 and Appendix H.1 is to enforce that features are not re-selected by the MSO after their removal in LMEA (and to do so without needing to modify the optimization loop of the MSO). When there is no smoothing mask $h$, the idea described in Section 6.3 accomplishes this objective without modification. However, when there is a smoothing mask $h$, enforcing against re-selection becomes more complicated, as we explain here. For simplicity of exposition (and without loss of generality), we assume that the input $\mathbf{x}$ has one-dimensional structure, rather than higher-dimensional structure (as images do).

Recall that the form of the RDE objective is

$$L(\mathbf{s}) \doteq \underbrace{\frac{1}{2}\, \mathbb{E}\left[\left\| f(h(\mathbf{s}) \odot \mathbf{x} + (\mathbf{1} - h(\mathbf{s})) \odot \widetilde{\mathbf{X}}) - f(\mathbf{x})\right\|^2\right]}_{\doteq\, \mathsf{suff}(\mathbf{s})} + \lambda\, \underbrace{\|h(\mathbf{s})\|_1}_{\doteq\, R(\mathbf{s})},$$

where $\mathsf{suff}(\mathbf{s})$ is a sufficiency term and $R(\mathbf{s})$ is a sparsity-encouraging regularization term.[20]

To see how the smoothing kernel affects the application of the model wrapper method described in Appendix H.1, we must examine the structure of the gradient $\nabla L(\mathbf{s})$. Recall that we want to achieve $\nabla\mathsf{suff}(\mathbf{s})_{\mathcal{A}^c} = 0$, so that previously selected features are driven to zero as described in Section 6.3. To do this, we apply the chain rule (leaving un-expanded the Jacobian of the regularizer $R(\mathbf{s})$):

$$J_L(\mathbf{s}) = \mathbb{E}\left[J_{\frac{1}{2}\|\cdot\|^2}\big(\Delta f(\widetilde{\mathbf{X}}_{\mathbf{s}})\big) J_f(\widetilde{\mathbf{X}}_{\mathbf{s}}) J_{\widetilde{\mathbf{X}}_{\mathbf{s}}}(\mathbf{s}) + J_R(\mathbf{s})\right], \tag{9}$$

---

[20]Throughout this section, we will assume that interchange of expectation and differentiation is justified.

where we recall that our convention is to write the Jacobian of a function $g$ evaluated at a point $\mathbf{z}$ as $J_g(\mathbf{z})$, and

$$\Delta f(\widetilde{\mathbf{X}}_{\mathbf{s}}) \doteq f(\widetilde{\mathbf{X}}_{\mathbf{s}}) - f(\mathbf{x}),$$

$$\widetilde{\mathbf{X}}_{\mathbf{s}} \doteq h(\mathbf{s}) \odot \mathbf{x} + (1 - h(\mathbf{s})) \odot \widetilde{\mathbf{X}}.$$

Computing both terms in the above chain rule expansion besides the $J_f$ term, we have

$$J_{\frac{1}{2}\|\cdot\|^2}\left(\Delta f(\widetilde{\mathbf{X}}_{\mathbf{s}})\right) = \Delta f(\widetilde{\mathbf{X}}_{\mathbf{s}})^\top, \text{ and}$$

$$J_{\widetilde{\mathbf{X}}_{\mathbf{s}}}(\mathbf{s}) = \text{diag}(\mathbf{x} - \widetilde{\mathbf{X}})J_h(\mathbf{s}).$$

Plugging these back into Equation 9, we get

$$J_L(\mathbf{s}) = \mathbb{E}\left[\Delta f(\widetilde{\mathbf{X}}_{\mathbf{s}})^\top J_f(\widetilde{\mathbf{X}}_{\mathbf{s}})\text{diag}(\mathbf{x} - \widetilde{\mathbf{X}})J_h(\mathbf{s}) + J_R(\mathbf{s})\right].$$

Taking the transpose, we obtain the gradient

$$\nabla L(\mathbf{s}) = \mathbb{E}\left[J_h(\mathbf{s})^\top \text{diag}(\mathbf{x} - \widetilde{\mathbf{X}}_{\mathbf{s}})J_f(\widetilde{\mathbf{X}}_{\mathbf{s}})^\top \Delta f(\widetilde{\mathbf{X}}_{\mathbf{s}}) + J_R(\mathbf{s})\right]. \tag{10}$$

To simplify the above expression, we will assume that $h(\mathbf{s})$ is a convolution with a kernel $\kappa$. For simplicity of presentation, we assume that $\kappa$ does not depend on $\mathbf{s}$. Then, neglecting boundary issues,

$$h_i(\mathbf{s}) = \sum_{\ell=-r}^{r} \kappa_\ell s_{i+\ell},$$

where $r$ is the radius of the kernel $\kappa$.[21] Set $A \doteq J_h(\mathbf{s})^\top \text{diag}(\mathbf{x} - \widetilde{\mathbf{X}})$. Then

$$A_{ij} = \sum_\ell [J_h(\mathbf{s})]_{\ell i}\text{diag}(\mathbf{x} - \widetilde{\mathbf{X}})_{\ell j}$$

$$= \sum_\ell [J_h(\mathbf{s})]_{\ell i}(x_\ell - \widetilde{X}_\ell)\delta_{ij}$$

$$= [J_h(\mathbf{s})]_{ji}(x_j - \widetilde{X}_j)$$

$$= \kappa_{i-j}(x_j - \widetilde{X}_j),$$

where $\delta_{ij}$ is the Kronecker delta. Let $\widetilde{J}_f$ denote the modified version of the Jacobian of $f$ with columns $j \in \mathcal{A}$ set to zero, as described in Section 6.3. To ensure that previously selected features receive zero gradient from the sufficiency term $\mathsf{suff}(\mathbf{s})$ during optimization, we want to modify the columns of $J_f(\widetilde{\mathbf{X}}_{\mathbf{s}})$ to produce a new matrix $\widetilde{J}_f(\widetilde{\mathbf{X}}_{\mathbf{s}})$ which forces the rows of the matrix $B \doteq J_h(\mathbf{s})^\top \text{diag}(\mathbf{x} - \widetilde{\mathbf{X}}_{\mathbf{s}})\widetilde{J}_f(\widetilde{\mathbf{X}}_{\mathbf{s}})^\top = A\widetilde{J}_f(\widetilde{\mathbf{X}}_{\mathbf{s}})^\top$ (compare with $AJ_f(\widetilde{\mathbf{X}}_{\mathbf{s}})^\top$ in Equation 10) to be zero. The entries of $B$ are given by

$$B_{ij} = \sum_\ell A_{i\ell}[\widetilde{J}_f(\widetilde{\mathbf{X}}_{\mathbf{s}})^\top]_{\ell j}$$

$$= \sum_\ell \kappa_{i-\ell}(x_\ell - \widetilde{X}_\ell)[\widetilde{J}_f(\widetilde{\mathbf{X}}_{\mathbf{s}})]_{j\ell}. \tag{11}$$

Since $\kappa_{i-\ell}(x_\ell - \widetilde{X}_\ell) \neq 0$ in general, we can see that the $i$th row of $B$ includes contributions from columns $\ell$ of the modified Jacobian $\widetilde{J}_f(\widetilde{\mathbf{X}}_{\mathbf{s}})$ within the kernel radius $r$ of the smoothing kernel $\kappa$. This means that by setting the $i$th column of $\widetilde{J}_f(\widetilde{\mathbf{X}}_{\mathbf{s}})$ to zero using the wrapper (where $i \in \mathcal{A}^c$), we are not guaranteed to eliminate gradient contributions from $\mathsf{suff}(\mathbf{s})$, since other features within the kernel radius of feature $i$ will contribute to the sum in Equation 11.

To eliminate these contributions, we want to ensure that the components of $\widetilde{J}_f(\widetilde{\mathbf{X}}_{\mathbf{s}})$ that are set to zero also exclude spatial neighborhoods of radius $r$ of each selected explanation $\mathcal{S}$. To achieve this, we employ a morphological dilation step (see, e.g., (Haralick et al., 1987)) using the `morphology.dilation` function of the scikit-image (Van der Walt et al., 2014) package using a disk element of radius $\rho$. This results in an enlarged version $\widetilde{\mathcal{S}}$ of the explanation $\mathcal{S}$, which is then removed from $\mathcal{A}$ in LMEA: $\mathcal{A} \leftarrow \mathcal{A} \setminus \widetilde{\mathcal{S}}$.

---

[21]Here we use the image processing convention of $i + \ell$ rather than $i - \ell$; this makes no difference in the analysis.

### I.2 Practical guidance on setting the dilation radius

From the above analysis, it is clear that the dilation radius $\rho$ is intimately related to the radius $r$ of the smoothing kernel $\kappa$. We recommend setting $\rho$ to be some multiple of $r$ that guarantees the contributions from inactive features in $\mathcal{A}^c$ in Equation 11 are mitigated. For a given smoothing kernel $h$ (possibly infinite in size), this can be done by finding a value of $\ell$ such that $\kappa_\ell \approx 0$. In our experiments, we set $\rho = 1.2\varsigma$ where $\varsigma$ is the kernel size in the mask generator function for XP experiments, or the radius of the smoothmax operation in RDE experiments.

## J Hyperparameter sensitivity analysis

Here we investigate the impact of the sufficiency hyperparameter $\varepsilon$ and the sparsity-based postprocessing (SBP) parameter $\delta$ on LMEA's performance, as measured by Expl., IoU, IoGT, IoE, sparsity, and sufficiency (i.e., $\|f(\mathbf{s} \odot \mathbf{x} + (\mathbf{1} - \mathbf{s}) \odot \mathbf{b}) - f(\mathbf{x})\|_{\mathrm{TV}}$). For the sufficiency level sweep, we set a maximum allowable total variation (TV) sufficiency level $\widetilde{\varepsilon}$. For each input $\mathbf{x}$, we set $\varepsilon = \min\{\widetilde{\varepsilon},\ \|f(\mathbf{x}) - (1/2, 1/2)\|_{\mathrm{TV}}\}$, i.e., we take $\varepsilon$ to be the minimum of the maximum sufficiency level $\widetilde{\varepsilon}$ and the TV distance to the uniform distribution over two classes. This means that LMEA stops as soon as the sufficiency level $\widetilde{\varepsilon}$ is violated or the prediction flips, whichever comes first. We choose $\widetilde{\varepsilon} \in \{0.1, 0.2, 0.3, 0.4, 0.5\}$. Sweeping $\delta$ is more straightforward: we simply apply each $\delta \in \{1.0, 3.0, 5.0, \infty\}$ in the SBP procedure. The $\delta = \infty$ case corresponds to no SBP, and is included for ease of comparison. We run the $\widetilde{\varepsilon}$ sweep without SBP and the $\delta$ sweep with the default rule for setting $\varepsilon$ described in Appendix H.1. The results are pictured in Figures 14–17.

Generally speaking, as $\delta$ increases, Expl. increases, IoU decreases, IoGT increases, IoE decreases, sparsity increases slightly, and the sufficiency (as measured by the TV norm between the masked input and the full input) increases slightly. The decrease in IoU and IoE is less pronounced for the XP MSO than the RDE MSO. As $\widetilde{\varepsilon}$ increases, Expl. increases, IoU decreases, IoGT increases, IoE decreases, sparsity increases slightly, and TV increases. These results are expected: smaller $\delta$ values impose a stricter requirement on the sparsity of the resulting solutions, producing $\varepsilon$-MSSs that are fewer in number but more specific to the ground truth object regions. These $\varepsilon$-MSSs therefore have higher IoU and IoE scores, and since there are fewer of them, their coverage of the union of ground truth regions as measured by IoGT score is lower. Likewise, a smaller $\widetilde{\varepsilon}$ means a stricter requirement on the definition of an $\varepsilon$-MSS. As $\widetilde{\varepsilon} \to 0$, we expect each of these to be individually fully sufficient for the prediction, and thus for the number of them to be fewer but their concordance with ground truth regions to increase. This matches the trends observed in Expl. and IoGT (which decrease with smaller $\widetilde{\varepsilon}$ due to the stricter requirement) and IoE/IoU (which increase with smaller $\varepsilon$ due to the stricter sufficiency requirement).

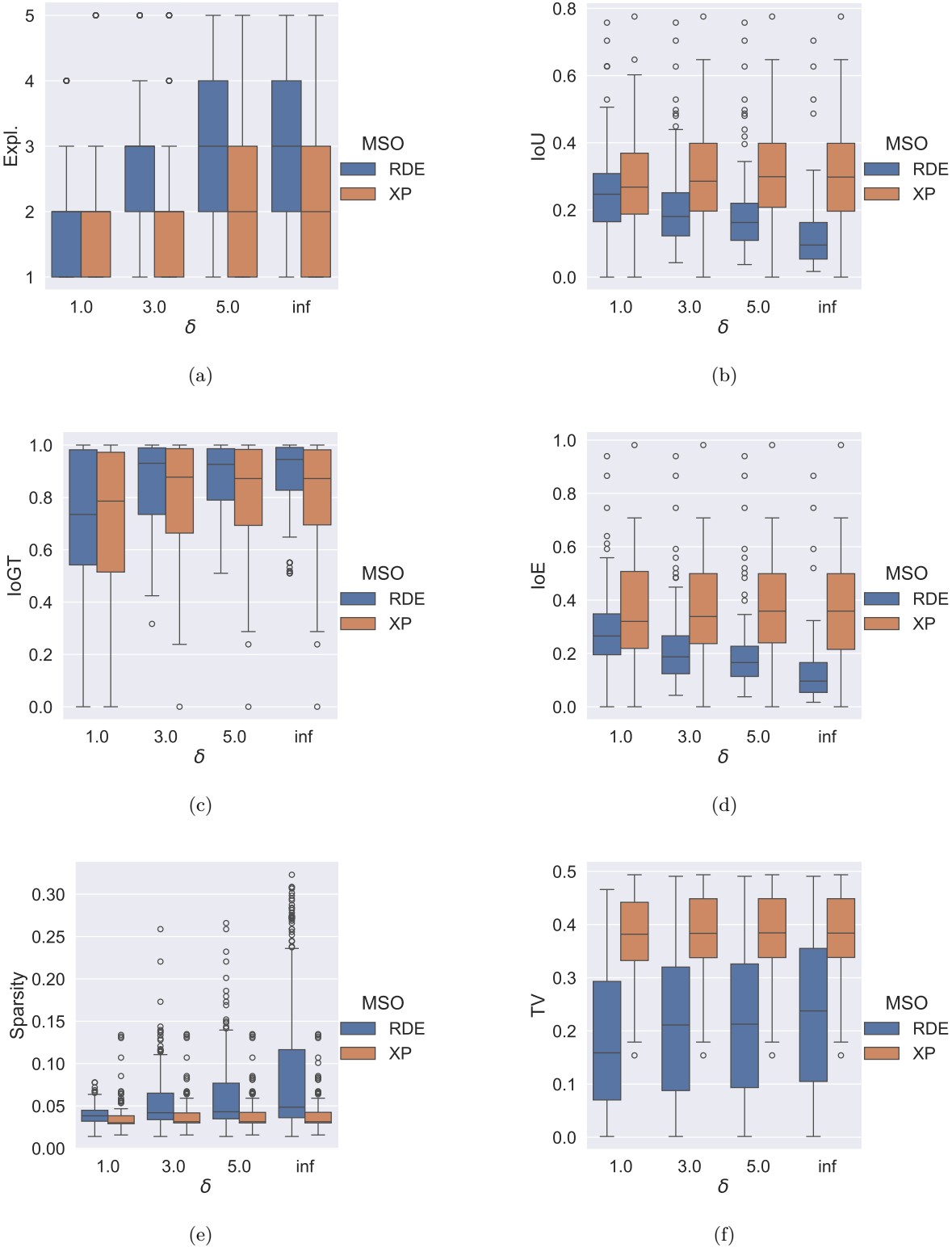

Figure 14: $\delta$ hyperparameter sweep results for the BBBC041 dataset.

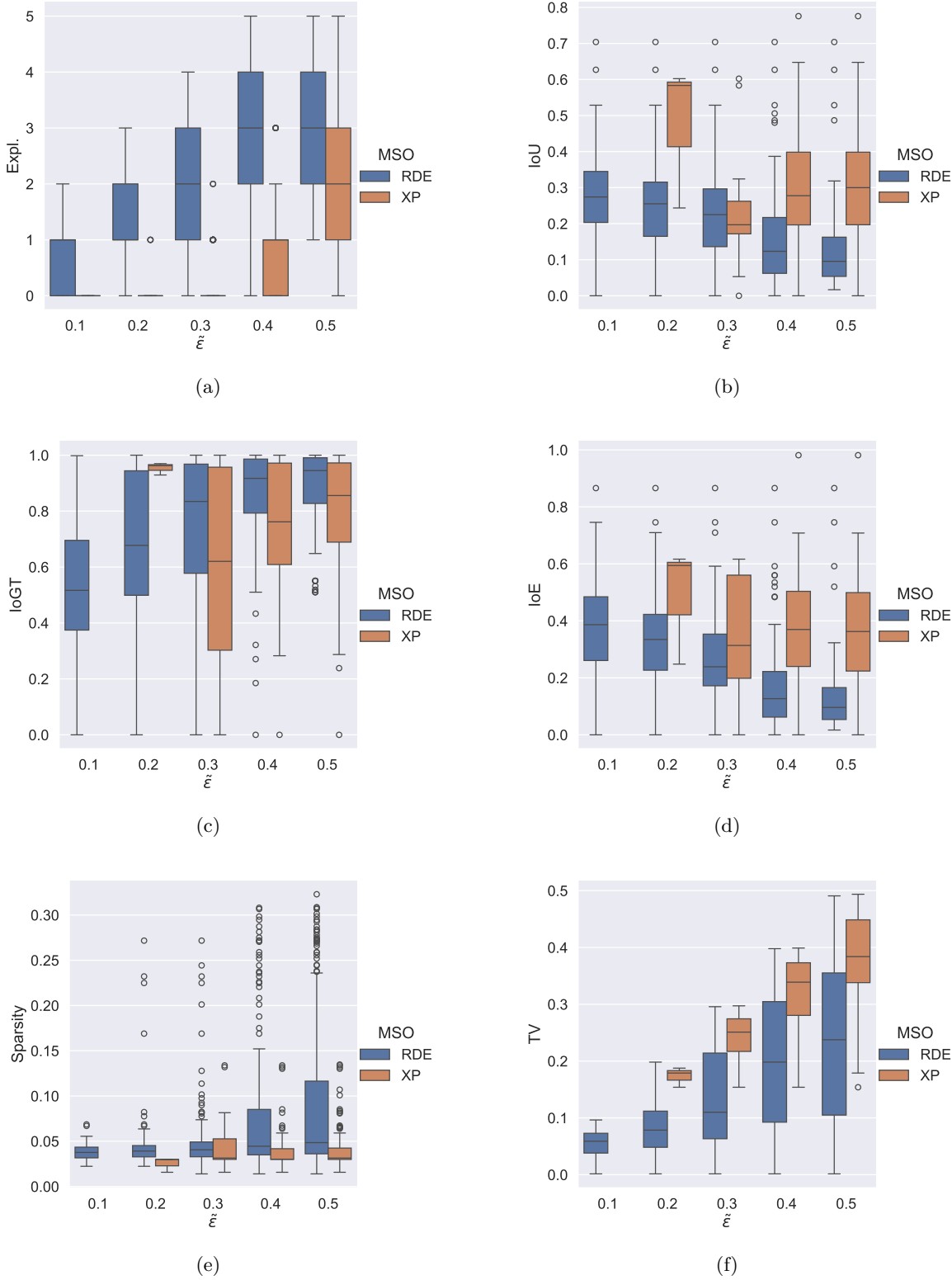

Figure 15: $\varepsilon$ hyperparameter sweep results for the BBBC041 dataset.

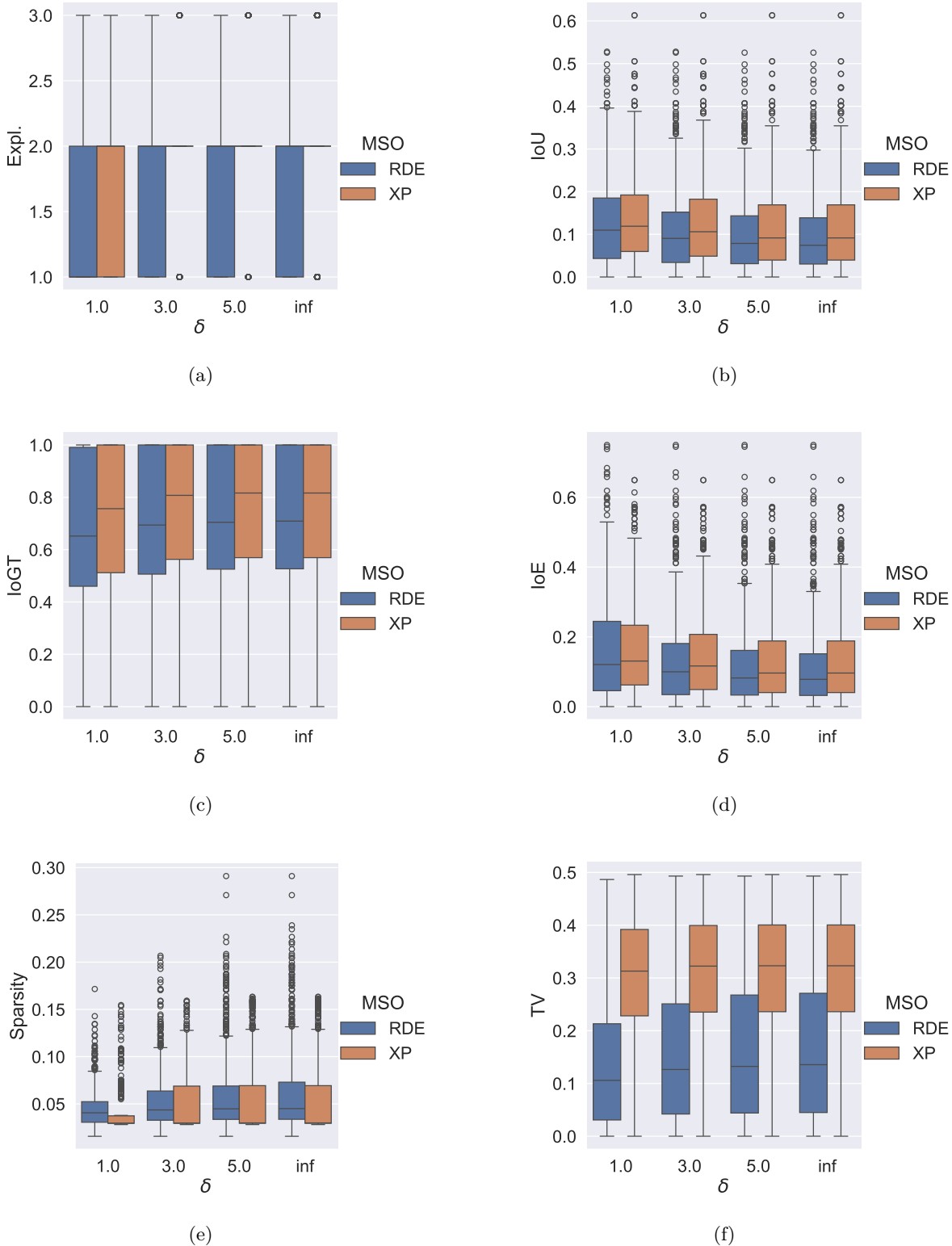

Figure 16: $\delta$ hyperparameter sweep results for the CelebAMask-HQ dataset.

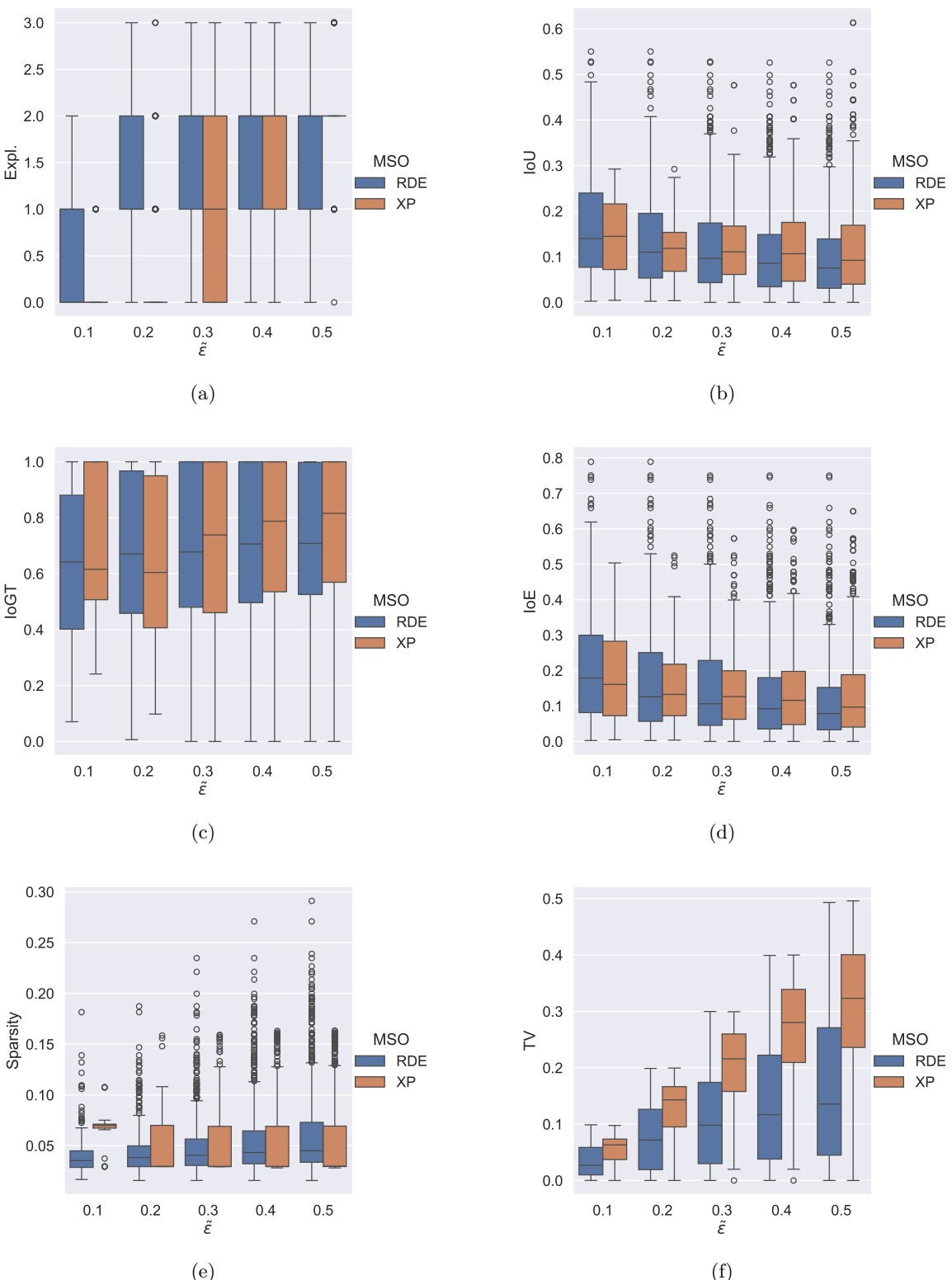

Figure 17: $\varepsilon$ hyperparameter sweep results for the CelebAMask-HQ dataset.

# K   Additional baseline background parameter experiments

In the experiments presented in Section 6, we set the background for baselines according to the defaults exposed in each baseline implementation (or those in other existing implementations, in the case of PSIS). In our experiments in Section 6, however, LMEA uses the mean image (computed over the training set) as a background for the sufficiency checks described in Appendix H.1.[22] To study the impact of background values on results, we re-run the experiments with the baselines configured to use this mean background. The results are reported in Tables 6–8. In these tables, we keep the rest of the configuration options the same as before and report the LMEA results again side-by-side for convenience.

On the shapes and BBC041 datasets, we find that the MultiReX and PSIS baselines achieve similar performance (but MultiReX finds more explanations on BBBC041 with the mean background, and IoU/IoE metrics degrade slightly relative to the default configuration). These datasets are such that relevant objects are spread throughout the image, and thus the mean image background is not much more structured than a constant background (such as MultiReX's zero default or PSIS' shared mean pixel background). On both datasets, as before, SBP improves the IoU and IoE scores of both baselines.

For the CelebAMask-HQ dataset, however, the faces in the images are aligned, and the mean image is thus more structured. For this dataset, each baseline finds more MSSs on average with the mean background than with their respective default settings. For the BS and PSIS baselines without SBP, the IoU and IoE scores are about the same, IoGT improves slightly, and the MSSs become smaller, relative to their default configurations. For the MultiReX baseline, IoU remains about the same while IoE improves and IoGT worsens substantially, relative to the default background. The MSSs that MultiReX finds with the mean background are also much smaller than under the default configuration. On this dataset with SBP applied, the aforementioned patterns for the BS baseline are largely the same. For the PSIS baseline with SBP applied, IoU decreases slightly, IoGT improves substantially, and IoE decreases. For MultiReX with SBP applied, IoGT worsens while IoE improves substantially.

Overall, while the baselines find more MSSs than LMEA most of the time (which is only substantially more pronounced for the mean background configuration than for the default configurations on the CelebAMask-HQ dataset), these MSSs have lower IoE and IoU scores than the MSSs found by LMEA, indicating that the MSSs found by the baselines lie outside of regions that are known to be relevant to the prediction task to a greater degree.

---

[22]Note that this differs from the default for PSIS, which is a shared mean pixel value, which produces a constant background image.

Table 6: Shapes results. Baselines use the mean image background. LMEA setup and results are unchanged from the main text but provided for ease of comparison.

| | Metric | | Single-MSS | | Multi-MSS | | | |
| | | | RDE | XP | LMEA$_{RDE}$ | LMEA$_{XP}$ | MultiReX | PSIS |
|---|---|---|---|---|---|---|---|---|
| **Without SBP** | Expl./Obj. | | 0.31 (0.27) | 0.31 (0.27) | 0.63 (0.16) | 0.93 (0.10) | 1.76 (0.55) | 2.78 (0.77) |
| | Expl. | ↑ | 1.00 (0.00) | 1.00 (0.00) | 3.51 (1.15) | 5.74 (2.01) | 9.46 (2.72) | **15.83** (4.74) |
| | IoU | ↑ | 0.30 (0.15) | 0.19 (0.08) | **0.55** (0.05) | 0.36 (0.02) | 0.15 (0.10) | 0.33 (0.10) |
| | IoGT | ↑ | 0.41 (0.29) | 0.32 (0.26) | **0.95** (0.07) | 0.92 (0.08) | 0.38 (0.10) | **0.95** (0.06) |
| | IoE | ↑ | **0.60** (0.09) | 0.39 (0.06) | **0.57** (0.05) | 0.37 (0.02) | 0.23 (0.20) | 0.34 (0.10) |
| | Sparsity | ↓ | **1.64** (0.54) | 1.85 (0.05) | 1.87 (0.59) | 1.81 (0.22) | 1.11 (2.24) | **0.81** (0.97) |
| | Missing | ↓ | **0.00** | **0.00** | **0.00** | **0.00** | **0.00** | **0.00** |
| **With SBP** | Expl./Obj. | | — | — | 0.59 (0.20) | 0.93 (0.11) | 1.03 (0.53) | 2.36 (0.34) |
| | Expl. | ↑ | — | — | 3.10 (1.10) | 5.72 (2.02) | 4.68 (1.64) | **14.58** (4.89) |
| | IoU | ↑ | — | — | **0.50** (0.10) | 0.36 (0.02) | 0.21 (0.11) | 0.42 (0.04) |
| | IoGT | ↑ | — | — | 0.84 (0.21) | **0.92** (0.09) | 0.23 (0.13) | 0.89 (0.06) |
| | IoE | ↑ | — | — | 0.57 (0.06) | 0.37 (0.03) | **0.84** (0.12) | 0.44 (0.05) |
| | Sparsity | ↓ | — | — | 1.76 (0.51) | 1.81 (0.15) | **0.18** (0.07) | 0.58 (0.20) |
| | Missing | ↓ | — | — | **0.00** | **0.00** | **0.00** | **0.00** |

Table 7: BBBC041 results. Baselines use the mean image background. LMEA setup and results are unchanged from the main text but provided for ease of comparison.

| | Metric | | Single-MSS | | Multi-MSS | | | |
| | | | RDE | XP | LMEA$_{RDE}$ | LMEA$_{XP}$ | MultiReX | PSIS |
|---|---|---|---|---|---|---|---|---|
| **Without SBP** | Expl./Obj. | | 0.50 (0.32) | 0.50 (0.32) | 1.26 (0.80) | 0.77 (0.54) | 0.99 (0.65) | 1.82 (1.81) |
| | Expl. | ↑ | 1.00 (0.00) | 0.96 (0.19) | 3.21 (1.06) | 2.07 (1.06) | 2.38 (0.80) | **4.19** (1.83) |
| | IoU | ↑ | 0.27 (0.15) | 0.27 (0.16) | 0.11 (0.09) | **0.31** (0.14) | 0.08 (0.09) | 0.08 (0.06) |
| | IoGT | ↑ | 0.57 (0.32) | 0.58 (0.34) | **0.90** (0.12) | 0.80 (0.21) | 0.69 (0.25) | 0.83 (0.12) |
| | IoE | ↑ | 0.35 (0.18) | **0.39** (0.20) | 0.12 (0.10) | **0.36** (0.17) | 0.10 (0.17) | 0.08 (0.08) |
| | Sparsity | ↓ | 3.69 (0.96) | **3.63** (2.47) | 9.21 (8.38) | **3.91** (2.43) | 17.64 (18.60) | 10.72 (13.78) |
| | Missing | ↓ | **0.00** | 3.53 | **0.00** | 3.53 | **0.00** | **0.00** |
| **With SBP** | Expl./Obj. | | — | — | 0.75 (0.49) | 0.73 (0.57) | 0.60 (0.42) | 1.05 (1.21) |
| | Expl. | ↑ | — | — | 2.01 (0.84) | 1.78 (0.99) | 1.25 (0.43) | **2.41** (1.36) |
| | IoU | ↑ | — | — | 0.26 (0.12) | **0.29** (0.14) | 0.17 (0.16) | **0.29** (0.13) |
| | IoGT | ↑ | — | — | **0.74** (0.24) | 0.72 (0.28) | 0.44 (0.31) | 0.59 (0.24) |
| | IoE | ↑ | — | — | 0.29 (0.15) | 0.37 (0.18) | 0.38 (0.35) | **0.41** (0.21) |
| | Sparsity | ↓ | — | — | 3.97 (1.15) | 3.54 (2.16) | 7.28 (10.36) | **2.10** (1.21) |
| | Missing | ↓ | — | — | **0.00** | 3.53 | **0.00** | **0.00** |

Table 8: CelebA results. Baselines use the mean image background. LMEA setup and results are unchanged from the main text but provided for ease of comparison.

| | Metric | | Single-MSS | | Multi-MSS | | | | |
| | | | RDE | XP | LMEA$_{\text{RDE}}$ | LMEA$_{\text{XP}}$ | MultiReX | PSIS | BS |
|---|---|---|---|---|---|---|---|---|---|
| **Without SBP** | Expl./Obj. | | 0.62 (0.22) | 0.62 (0.22) | 1.11 (0.46) | 1.15 (0.43) | 3.49 (1.39) | 9.40 (3.77) | 2.46 (0.92) |
| | Expl. | ↑ | 0.99 (0.09) | 1.00 (0.05) | 1.90 (0.65) | 1.98 (0.53) | 6.18 (2.14) | **15.61** (3.67) | 4.52 (1.77) |
| | IoU | ↑ | 0.14 (0.11) | 0.12 (0.08) | 0.10 (0.09) | **0.12** (0.10) | 0.03 (0.03) | 0.07 (0.06) | 0.06 (0.05) |
| | IoGT | ↑ | 0.55 (0.30) | 0.51 (0.28) | 0.71 (0.25) | 0.76 (0.24) | 0.11 (0.16) | 0.89 (0.15) | **0.94** (0.13) |
| | IoE | ↑ | **0.19** (0.16) | 0.16 (0.14) | 0.12 (0.12) | **0.13** (0.12) | 0.08 (0.13) | 0.07 (0.07) | 0.06 (0.06) |
| | Sparsity | ↓ | **4.21** (1.78) | 4.25 (2.37) | 6.19 (4.39) | 5.24 (3.59) | **1.06** (3.68) | 1.76 (3.90) | 4.69 (2.04) |
| | Missing | ↓ | 0.88 | **0.22** | 0.88 | 0.22 | **0.00** | **0.00** | **0.00** |
| **With SBP** | Expl./Obj. | | — | — | 0.88 (0.39) | 0.94 (0.30) | 1.97 (1.06) | 7.37 (3.21) | 2.18 (0.90) |
| | Expl. | ↑ | — | — | 1.49 (0.53) | 1.68 (0.48) | 3.42 (1.65) | **12.32** (3.79) | 4.04 (1.77) |
| | IoU | ↑ | — | — | 0.13 (0.10) | **0.14** (0.10) | 0.04 (0.05) | 0.11 (0.09) | 0.08 (0.06) |
| | IoGT | ↑ | — | — | 0.66 (0.28) | 0.72 (0.27) | 0.05 (0.10) | 0.80 (0.22) | **0.91** (0.16) |
| | IoE | ↑ | — | — | 0.16 (0.15) | 0.17 (0.13) | **0.30** (0.28) | 0.13 (0.11) | 0.08 (0.07) |
| | Sparsity | ↓ | — | — | 4.56 (2.11) | 4.07 (2.24) | **0.07** (0.22) | 0.87 (0.28) | 4.17 (1.61) |
| | Missing | ↓ | — | — | 0.88 | 0.22 | **0.00** | **0.00** | **0.00** |

