# OpenReview forum: "Let Me Explain, Again: Multiplicity in Local Sufficient Explanations"
_TMLR — Accepted by TMLR_

### Review · Reviewer_5gfh · 2026-01-13

**Summary Of Contributions:**

Summary

This paper analyzes the multiplicity of local sufficient explanations. First, from various perspectives including logic, conditional independence, and non-linearity, the paper demonstrates the non-uniqueness of minimal sufficient subsets (MSS), which are defined as a minimal subset of features sufficient to approximately explain the model prediction. Then, the paper proposes a meta-algorithm LMEA which extends a previous work finding a single MSS to find multiple MSSs. The proposed algorithm is tested on various image classification tasks to show that it achieves competitive performance compared to existing multiple-MSS algorithms.

Strengths
- The paper is clearly written. While I am not an expert in this area, I could easily understand the background and follow the logical flow of the paper.
- The paper addresses the non-uniqueness of MSS from extensive perspectives with concrete examples.
- The experiment setup is concrete with various metrics.

Weaknesses
- The paper explicitly states the similarity of LMEA with Byra & Skibbe (2025): "Byra & Skibbe (2025), which is similar to LMEA with extremal perturbations (XP) (Fong et al., 2019) as the MSO", "For the XP MSO, LMEA is similar to the method of Byra & Skibbe (2025) with an infinite SørensenDice (Dice, 1945; Sørensen, 1948) penalty coefficient". However, in Section 6, the evaluation of LMEA against Byra & Skibbe (2025) is missing.

**Additional Comments:**

None.

**Audience:**

Yes

**Audience Explanation:**

Explainable/interpretable machine learning has gained lots of attention in recent years. The paper addresses an important question in this area and provides several interesting findings.

**Broader Impact Concerns:**

The paper is missing a Broader Impact Statement section. The authors are encouraged to write a short section regarding the ethical implications of this work.

**Claims And Evidence:**

Yes

**Claims Explanation:**

First, the paper analyzes the non-uniqueness of MSS from various perspectives. For each perspective, the authors provide an intuition, concrete examples, and the proof related to the existence of non-uniqueness of MSS.

Second, the authors prove that the proposed algorithm can provably retrieve all MSSs (under specific assumptions). Then, the proposed algorithm is evaluated under a concrete experiment setup to demonstrate that it can retrieve almost all MSSs in practical scenarios. The algorithm is compared to the baselines under various metrics and various datasets with a specific example.

**Requested Changes:**

(major) While the authors explicitly acknowledge the similarity of the proposed algorithm with Byra & Skibbe (2025), it is not included as a baseline in Section 6. The authors are encouraged to compare the experimental results against Byra & Skibbe (2025), report any similarities/differences in the experimental results, and more clearly state the differences between the two methods.

(major) Adding more empirical analyses would strengthen the contributions of the paper. For example, does LMEA work well when the model becomes larger (and thus the number of MSS may increase due to the increased nonlinearity of the function)? Does LMEA work well when the input dimension becomes larger (i.e., more features, e.g., high-resolution image)?

(minor) It would be better to include hyperparameter sensitivity analysis of the proposed algorithm, for example, about the maximum number of explanations $N$, sufficiency level $\epsilon$, and $\delta$ for SBP.

(minor) Section 6.8 could be strengthened by providing some insights for the practitioners, e.g., which algorithm (RDE or XP) should be used for MSO under a specific setting.

(minor) In Section 4, it seems that some of the perspectives (and the related examples) have already been addressed in the literature (e.g., Sec 4.1). It would be better to explicitly state which perspective (and the related examples) have already been addressed in the literature and which have not.

(minor) There are some parts where \citep should change to \citet (for example in Sec 1.1).

---

> ### Author Response · Authors · 2026-04-07
> **Response**
>
> We thank the reviewer for their thoughtful feedback. We first address the questions the reviewer had about the manuscript. Then we explain the changes we made to the manuscript as requested by the reviewer.
>
> # Weaknesses
>
> ---
>
> ### **Evaluation against Byra \& Skibbe**
>
> We do state that LMEA shares similarities with the method of Byra & Skibbe (2025). Unfortunately, we do not include an evaluation against their method because there is still no publicly available implementation. While the paper (published January 2025) states that the implementation is available at `github.com/mbyr/INR-EXP`, the repository does not contain code for the proposed method (as of now). We also emailed the authors to request access to the implementation, but did not receive it.
>
> Moreover, we did not think it would be fair to construct our own implementation of their method. Even small differences in details that are not fully specified in the manuscript (e.g., training schedule and optimizers) could significantly affect results, and a third‑party reimplementation could therefore lead to an inaccurate or potentially misleading comparison. In the event code becomes available, we will gladly include this comparison and add it to an arXiv version of this work.
>
> # Changes to paper
>
> ---
>
> ### **Evaluation of Byra \& Skibbe**
> While we are not able to provide a comparison to Byra \& Skibbe (2025), we do agree with the reviewer that we need to more clearly state the differences between our method and theirs. We provide a more detailed discussion in **section 6.1** of the revised manuscript.
>
> ### **Additional empirical analysis for LMEA**
>
> We agree that additional empirical analysis of LMEA would strengthen this work. The reviewer suggests two experiments: (1) using bigger models to evaluate how LMEA performs (since bigger models may have more MSSs), and (2) evaluating how LMEA works when the input dimension is larger. We kindly disagree that these are the correct axes along which to further evaluate LMEA. With a larger model (and potentially more MSSs), LMEA will simply take longer: it may require more iterations, and every evaluation of the MSO (e.g., XP and RDE) will also be slower. With larger images, LMEA will similarly take longer because each MSO evaluation becomes more expensive. In both cases, these experiments mainly measure computational cost rather than providing insight into when LMEA succeeds or fails.
>
> That said, the reviewer’s comment did make us consider different ways to evaluate LMEA’s strengths and weaknesses. Thus, we add additional synthetic experiments that evaluate LMEA’s performance when (1) MSSs overlap, and (2) the MSO is not an oracle and does not always return a true MSS. These experiments are included in **Appendix D.1 & D.2**. To summarize, LMEA’s performance naturally degrades when MSSs overlap and when the MSO is imperfect. We believe these findings are important to report, as they can help inform future work aimed at improving LMEA.
>
> ### **Insights for practitioners**
>
> The main benefit of LMEA is that it allows practitioners to use any MSO of their choosing. We do not believe we are in the position to comment on which MSO (XP or RDE) should be used for a specific setting. The papers that introduced these methods provide comparisons. We have added a comment encouraging readers to read these works themselves to make an informed decision.
>
> ### **Edits to section 4**
>
> We agree with the reviewer. We have added new explicit comments throughout section 4 that clarify which perspectives/examples have (and have not) been addressed in previous literature.
>
> ### **Other minor points**
>
> We have changed instances of \citep to \citet where needed. Thank you for pointing this out.
>
> ### **Hyperparameter sensitivity analysis**
>
> We appreciate the suggestion and agree that a hyperparameter sensitivity analysis would strengthen the manuscript. We are currently conducting this analysis for LMEA and will include the results and discussion in the revised manuscript.

---

> > ### Comment · Reviewer_5gfh · 2026-04-07
> > **Reply to rebuttal**
> >
> > I thank the authors for their detailed response. However, I cannot find the updated parts in the paper (Sec 6.2 -- differences between Byra & Skibbe and your method, App. D.1,D.2 -- additional experiments). Could you check this?

---

> > > ### Author Response · Authors · 2026-04-07
> > >
> > > The paper should be updated now! Sorry for the inconvenience.

---

> > > > ### Comment · Reviewer_5gfh · 2026-04-08
> > > > **Reply to rebuttal**
> > > >
> > > > Thanks for the update. I have read the updated manuscript and do not have any remaining concerns.

---

> > > > > ### Author Response · Authors · 2026-04-16
> > > > > **Hyperparameter sensitivity analysis**
> > > > >
> > > > > As suggested, we ran the hyperparameter sensitivity analysis on the sufficiency parameter $\varepsilon$ and the SBP parameter $\delta.$ Overall, we find that IoU and IoE improve with stricter (i.e., smaller) $\varepsilon$ and $\delta$, while IoGT improves and Expl. increases with looser (i.e., bigger) $\varepsilon$ and $\delta$. For figures and a complete description of the setup and discussion, see Appendix J in the updated revision.

---

### Review · Reviewer_7V1r · 2026-01-19

**Summary Of Contributions:**

The authors present a discussion on the presence of multiple sufficient explanations and when we might expect to see them wrt data and model. They argue that, because multiple sufficient subsets (MSSs) are more likely than not, XAI tools should start from this as a default. I agree. They then proceed to present an algorithm, LMEA, which wraps a class of XAI tool to enable them to discover MSSs. They introduce a post-processing step to this algorithm, SBP, which the authors apply to all of the tools evaluated.

# Strengths
This is undeniably an interesting problem. I like the attempt to lift single MSS producing tools to multiple-MSS tools. This is genuinely useful.

The paper is well written, well proof-read and flows logically. For the most part, it is a self sufficient read (though see later comments). I've included some updated bibtex for the authors to use.

Overall, while I think the contribution is relatively modest, it *is* a contribution and deserves to be read.

# Weaknesses
I think we need to be careful about treating 'explanation' and 'MSS' as if they were synonyms and I don't believe that they are. I'll discuss this from the point of view of actual causality (Halpern 2019) where (to give a very brief summary) an explanation is a minimal subset of features $s$ sufficient to bring about outcome $o=O$ in a set of contexts $K$. It's the set of contexts which is vital, as it codifies the `explanatory power' of $s$. Without $K$ we really only have an actual cause, not an explanation. Now, MultiReX computes its MSSs the same way as ReX (Chockler et al. 2024), so $K$ is (loosely) defined as the set of all permutations of the original image with the masking value. So, MultiReX has some claim to provide explanations wrt theory. Can we say the same about the other tools? Perhaps just refer to MSS or explain what you mean by the word explanation.

Sparsity-based postprocessing feels very ad-hoc. How does it ensure sufficiency? Perhaps I'm being unfair but it appears that the claim is: if it looks a little large, it must be wrong. But this is surely wrong. As a simple example, image an object like a fork, in which each individual tine is an MSS, as is the handle. By this logic, wouldn't you likely delete the MSS that corresponds with the handle? MultiReX in particular, returns explanations of greatly varying size, but your rule will throw most of them out.

The experimental design and metrics are a little odd and difficult to interpret (see below). This is possibly because a comparison is being made between tools that don't really have very much in common, beyond claiming to find MSSs (but each has its own definition). Would it not be simpler just to compare tools pre- and post- LMEA to show that your algorithm does what it claims?


# Bibliography

@inproceedings{kelly2025big,
  title={I am big, You are Little; I am Right, You are Wrong},
  author={Kelly, David A and Chanchal, Akchunya and Blake, Nathan},
  booktitle={Proceedings of the IEEE/CVF International Conference on Computer Vision},
  pages={817--826},
  year={2025}
}

@inproceedings{chockler2025,
   title     = "Multiple Different Black Box Explanations for Image Classifiers",
   author    = "Hana Chockler and David A. Kelly and Daniel Kroening",
   year      = "2025",
   doi       = "10.3233/FAIA250869",
   volume    = "413",
   series    = "Frontiers in Artificial Intelligence and Applications",
   pages     = "699 -- 706",
   booktitle = "ECAI 2025 - 28th European Conference on Artificial Intelligence, including 14th Conference on Prestigious Applications of Intelligent Systems, PAIS 2025 - Proceedings",
}

@article{chockler2024causal,
  title={Causal explanations for image classifiers},
  author={Chockler, Hana and Kelly, David A and Kroening, Daniel and Sun, Youcheng},
  journal={arXiv preprint arXiv:2411.08875},
  year={2024}
}

@inproceedings{deepcover,
  title={Explanations for occluded images},
  author={Chockler, Hana and Kroening, Daniel and Sun, Youcheng},
  booktitle={Proceedings of the IEEE/CVF International Conference on Computer Vision},
  pages={1234--1243},
  year={2021}
}

**Audience:**

Yes

**Audience Explanation:**

I found it interesting and I enjoyed reading it. It highlights an important, but virtually unexplored, area of XAI which is multiple explanations. I do not think, and the authors do not claim, that this paper is the last word on the subject, but I expect that many of your readers will not have given the subject any thought at all. This is definitely a topic of interest to TMLR readers.

**Broader Impact Concerns:**

I don't see any ethical concerns or implications in this paper.

**Claims And Evidence:**

Yes

**Claims Explanation:**

I could not see any obvious errors in the paper. The discussion on the existence of multiple explanations (section 4) is clear and convincing.
I do not really like the experimental design or evaluation, but nor can I see anything wrong with it *per se*. The results look believable, though I feel that a bit of `massaging' has been done (via SBP and other techniques) to make the numbers appear as they do. Appendices appear complete with some nice extra examples.

**Requested Changes:**

** Introduction (minor changes)
   - paragraph 2 should probably cite either DeepCover or ReX as you use MultiReX as a baseline. MultiReX is really just a wrapper around ReX/DeepCover (bibtex above)
   - I've provided an up-to-date bibtex for MultiReX, which was presented at ECAI 2025
   - p2 "representative subset" is a bit vague at this point.
   - p3 to be more precise, Chockler and Shitole allow you to choose disjoint or overlapping, not merely overlapping

** Setting and Background (minor changes)
   - Problem Setting "we seek a small sufficient set". Why small? Do you not mean minimal? You go on later to say that small and minimal are used as synonyms, but this is not correct. Minimal does not imply small, though in practice this might be the case.

** Minimal Sufficient Subsets (minor changes)
   - I don't like the use of the word "informative" in section 3. In terms of information theory, they probably have low entropy (depending on context). In terms of image classification, they might be so stripped back as to give you little human-interpretable information. See "I am big, You are Little; I am Right, You are Wrong" (Kelly et al, 2025) bibtex below.

** Examples (minor changes)
   - Example 1. Why include f(x) = P in it? It isn't used or referred to again for this example. I know that it is relevant later on, but I feel that as this is the simplest example of the lot, it should not contain extraneous information. When I read it I just found myself wondering why it was there.
   - minor quibble, but why lump OR logic example and vision problems together? These are probably the two
     easiest and most intuitive examples; they could be fleshed out a little.
  - Perhaps this indicates and important gap in my knowledge, but I had to look up "CI relationship". Please dedicate a few lines to this for readers who are not familiar.
  - this entire section is one of your contributions, but feels a little underpowered. I recognise that that is not a very helpful description, but I would like to see more detail/discussion or even a more complex example.

** Mega-Algorithm
   - what does it mean to remove S from the active feature pool A? You gloss over this like it's a trivial thing, and discuss only much later in your experimental section (where it seems you just 0 things out?). But this is a really important point. Doesn't removal just push each input (and therefore each MSS) more and more out of distribution? Consider a case of a binary classifier, given that it has to return something,  regardless of how much "real" information it receives, how do you know when the MSS is being determined mostly by your removal technique or by the information contained in x? There really aren't any neutral values for a model: some ImageNet models return `moped' when sent a matrix of 0.

   - consider a case where your MSO finds a large sufficiency which intersects with two, much smaller sufficiencies. It finds the large one first. Does that mean that your algorithm must now reject the two smaller sufficiencies as they are no longer sufficient given the removal of some of their features? Couldn't you employ some backtracking to take this into account? You do address this point, but you might want to bring that forward to here to make reading easier.

 - It concerns me how many times you refer to "strong assumption" so far in the paper (and we are only on p9).

  - Overall, I think your algorithm too simplistic for a complex problem. With that said, your evaluation seems to support the case that it works reasonably well. So, I do not really like the LMEA algorithm, but it will not act as a blocker to acceptance. You could address design limitations more and why you kept it so simple and discuss possible improvements in future work.

** Experiments (I'd like to see much more discussion of choices, and evaluation to be redesigned)

 - I found the experimental effort to be sufficient, but the experimental evaluation is a little odd.
   - You shouldn't be relying on segmentations or ground truth labels, but the model itself as oracle. Essentially, are you not penalising tools which (correctly) find MSSs outside of the segmentation? By doing this, you exclude the type of incorrect reasoning that MultiReX is really designed to find. Are explanations there to make a human feel better (segmentation agreement) or to show what the model is doing?
  - bottom p9, are you missing a comma after y* = argmax_y ?
   - why not use Shitole as a baseline as well?
   - is MultiReX the only blackbox method you use?
   - I understand why you've done IoGT and things like this: it's to keep reviewers happy. But they are meaningless here, no? Surely the only test that matters is whether the MSS is sufficient or not, and I don't see any evidence that you pass the MSS back to the model to double check. You are comparing apples with oranges here.
 - you're beating around the bush onf Expl./Obj. If an object is large, it is still only 1 salient object (with one gt and segmentation) but might have many MSSs. Does this number tell us anything useful at all? Why not just report Expl?
   - you do not need "a measure of quality an reliability" other than the model itself providing the classification (a la MultiReX).

 - I've already discussed SBP above, so won't do it again here.

   -- Section 6.5
      aren't you in danger of encoding shortcuts by training to 100% accuracy? These shortcuts would then (hopefully) be detected by the MEMs but would then be penalised for those detections by applying SBP and your other metrics.

   -- p16 "shows that LMEA improves the ability of XP": a strange sentence, as they have no such ability without LMEA.

   -- "other metrics besides the number of explanations found"... isn't that the whole point of MeMs?

** Appendices
   Appendix E appears to be empty. Is this just a problem with floating figures?

---

> ### Author Response · Authors · 2026-04-07
> **Response**
>
> # Reviewer 7V1r
> We thank the reviewer for their thoughtful feedback. We first address the weaknesses you point out in our work. Then we explain the changes we made to the manuscript as requested by the reviewer.
>
> ## Weaknesses
>
> ### Explanation vs. MSS
>
> ### Sparsity based post-processing
> - The reviewer makes a good point. However, we would like to stress that sparsity-based postprocessing (SBP) does not itself ensure sufficiency. Instead, sufficiency is checked after the explanation is returned by the MSO. While this step is implicit in Algorithm 1, we do check explicitly for sufficiency at each step of LMEA; see Appendix G.1.
>
>   Also, the reviewer is correct that valid MSSs may vary in size and that MultiReX returns a set of explanations of varying size. LMEA without SBP does the same and SBP is an optional component of LMEA. We believe SBP can be useful in settings where domain knowledge suggests that sufficient regions will approximately be small and of the same size (the malaria experiment being a prime example). In conclusion, a user can choose to use SBP or not. We do not advocate for or against the use of SBP. That is why we report results both with and without SBP.
>
> ### Experimental design \& metrics
> - We start by addressing your question about evaluating tools pre- and post-LMEA. We evaluate LMEA along two axes. The first is its ability to augment MSOs to output multiple MSSs. We already report these results in Tables 1-3. We could have just stopped here; however, LMEA is a method to find multiple MSSs. Therefore, we also evaluate LMEA along this second axis by comparing it to the baselines in our experiments.
> - We appreciate the reviewer's perspective on evaluation; we think they raised a very nice point here. However, we respectfully disagree with their philosophy of treating the model itself as an oracle. Ideally, yes, we would like to investigate what's true for the algorithm, and for this ideal, ground truth labels are irrelevant. However, at the end of the day, most methods are trying to solve a nonconvex (or otherwise computationally difficult) problem, and their correctness often can't be guaranteed. With this in mind, we believe that it is important to evaluate some notion of correctness of the explanations. One way to do this is to set up an experiment, like the shapes example in Section 6.5.1, where the features needed for the prediction are known in advance and the model is able to achieve perfect accuracy. While our experiments on CelebA in Section 6.7 and the malaria dataset in Section 6.6 depart from this ideal, the model accuracy is still high, and we still believe that these experiments provide value in evaluating the correctness of the explanations obtained from LMEA and the baselines.
>
> ## Changes to paper
>
> ### Section 1: Introduction
> - Thank you for bringing up DeepCover and ReX. We have cited them in **paragraph 2**.
> - We agree that that term "representative subset" is vague. We have removed it from the **final paragraph** and have made slight modifications.
> - We have refined our language in **paragraph 2 of the related work** to reflect that Chockler and Shitole allow for disjoint or overlapping explanations.
>
> ### Section 2: Setting and Background
> - We agree with the reviewer that small and minimal are not the same. We used "small" because the definition of "minimal" has not been introduced yet. We have modified the sentence to say "minimal".
>
> ### Section 3: Minimal Sufficient Subsets
> - We agree we loosely used the term "informative". We have removed this and modified the paragraph.
>
> ### Section 4: Examples
> - You are right that it is not necessary to define $f(X) = \mathbb{P}(Y=1|X).$ However, to be precise, our definition of sufficiency is with respect to a model f. We do say after the Definition 1: "When f, x, or D are clear from context we will omit them". In this example we were not sure if it would be immediately clear to everyone, so we include $f(X) = \mathbb{P}(Y=1|X=x)$. We have rephrased Example 1  to highlight the importance of $f$.
>
> - We highlight the `OR` logic and vision problem examples to show that the `OR` logic is not a special or contrived case for multiple MSSs; vision problems, in particular, provide strong evidence that this pattern, and thus multiple MSSs, are common in real-world situations. We have reworded the paragraph to emphasize this.
>
> - We agree that CI relationships should have been properly defined. We have added a few sentences to clearly them. See **paragraph 2 of Section 4.2**.
>
> - The reviewer is correct in noting that this section represents one of our main contributions. We acknowledge that we may have understated its significance in the original manuscript. Our goal was to present this section in a clear and measured way, avoiding grandiose claims or overstating the impact of our results. However, we have now added new text throughout this section (in blue) to better highlight the importance of these examples and results.

---

> > ### Author Response · Authors · 2026-04-07
> > **Response (continued)**
> >
> > ### Section 5: Meta-algorithm
> > - In Algorithm 1, LMEA defines removal via a set difference operation. The reviewer is right to question what this actually means in practice. In fact, how to best remove features is a key question in the interpretability literature. There are multiple such removal methods, some of which are expensive for high-dimensional inputs such as images. We follow prior work and use inexpensive approximations to "on-manifold" inputs, such as the mean image or a blurred version of the input. In LMEA, removal is achieved by constraining the MSO not to re-select features, *not* by passing a perturbed input to the MSO. In particular, for XP and RDE, we accomplish this by modifying the gradients used by the optimizer, which does not in itself push the input out of distribution for the model.
> >
> > - The reviewer brings up the following example:
> >   > Consider a case where your MSO finds a large sufficiency which intersects with two, much smaller sufficiencies. It finds the large one first. Does that mean that your algorithm must now reject the two smaller sufficiencies as they are no longer sufficient given the removal of some of their features. Couldn't you employ some backtracking to take this into account??*
> >
> >   The reviewer is correct: when MSSs are intersecting (as in the example), the order in which LMEA selects MSSs directly impacts which subsets are included in the set of MSSs, $\mathcal{E}$, that LMEA returns. While $\mathcal{E}$ may not contain every MSS, in Proposition 2 we demonstrate that the union of all subsets in $\mathcal{E}$ intersects *every* existing MSS. (This holds true in the reviewer's example as well. LMEA could return either the large MSS or the two smaller MSSs. If $\mathcal{E}$ contains the large MSS, it intersects the two smaller ones. Conversely, if $\mathcal{E}$ contains the two smaller MSSs, both intersect the large MSS). Although this result does not guarantee that every MSS is returned, it does ensure that $\mathcal{E}$ is still representative of all MSSs.
> >
> >   We agree that some form of backtracking algorithm could potentially resolve the issue discussed. However, we believe that one would need to assume additional structure on the set of MSSs to devise a principled backtracking method that, combined with LMEA, would return all MSSs. This is an interesting direction for future work, and we elaborate on this point in **section 5.2**.
> >
> > - We have taken the reviewers advice and dropped some instances of "strong assumption" where we feel they are not needed.

---

> > > ### Author Response · Authors · 2026-04-07
> > > **Response (continued)**
> > >
> > > ### Experiments:
> > >
> > > - We appreciate the reviewer's suggestion that a comparison to Shitole's method would improve the paper. We are currently working toward this comparison, and we will include results and discussion as soon as possible in a revised manuscript.
> > >
> > > - The reviewer asks, "bottom p9, are you missing a comma after y* = argmax_y ?" No, we do not believe that this is a typo.
> > >
> > > - The reviewer asks, "is MultiReX the only blackbox method you use?" No. Both MultiReX and our PSIS baseline are black-box methods.
> > >
> > > - We disagree that "the only test that matters is whether the MSS is sufficient or not," as we elaborate on in our earlier response under "Experimental Design & Metrics." Our IoGT, IoE, and IoU, and #Expl./Obj. metrics are defined to measure how well the MSSs match regions of the input that are sufficient for the label according to the task itself. When the model performs well, we can expect that identifying MSSs for the prediction is similar to identifying MSSs for the label (and if $f_y(X) = \mathbb{P}(Y=y|X),$ then this correspondence is exact). IoE and IoGT measure quantities analogous to precision and recall of the MSSs obtained, as outlined in our earlier comment and Section 6.4 of the paper, while IoU measures a combination of the two.
> > >
> > > - The reviewer rightly points out that there is not necessarily a 1-1 correspondence between the MSSs recovered by LMEA and the number of objects in the image in our experiments. However, it is of interest to ask how many MSSs are recovered for each object in the image (each of which is individually sufficient for the prediction) which motivates our inclusion of the Expl./Obj. metric. We concede, though, that the raw number of MSSs is also of interest. We have therefore added an Expl. metric to Tables 1-3 in the paper.
> > >
> > > - The reviewer asks,
> > > > Section 6.5 aren't you in danger of encoding shortcuts by training to 100% accuracy? These shortcuts would then (hopefully) be detected by the MEMs but would then be penalised for those detections by applying SBP and your other metrics.
> > >
> > > The dataset on which our model achieves 100% accuracy is synthetic, and the 100% reported is on held-out data. The model is able to achieve perfect performance due to the simplicity of the task, and the fact that 100% accuracy is achieved on the validation set is evidence of the *absence* of shortcut reasoning, rather than the presence of such reasoning. We therefore disagree with this critique.
> > >
> > > - We thank the reviewer for pointing out the awkward sentence on p. 16. We have updated the draft (see paragraph preceding Section 6.6) with more natural phrasing.
> > >
> > > - The reviewer asks,
> > > > "other metrics besides the number of explanations found"... isn't that the whole point of MeMs?
> > >
> > > As discussed in our other responses, we do not believe that this is the only valid axis of evaluation for MeMs. Hence we do not elaborate on this issue further here.

---

> > > > ### Comment · Reviewer_7V1r · 2026-04-09
> > > > **thank you**
> > > >
> > > > Thanks to the authors for making many suggested changes. I feel like they improve the readability and clarity of the paper. I'm happy to agree to disagree on the valid axis of evaluation for MeMs.

---

> > > > ### Author Response · Authors · 2026-04-16
> > > > **Comparison to Shitole et al. (2021)**
> > > >
> > > > We thank the reviewer for their suggestion to compare to Shitole et al. (2021). We have done this and incorporated the corresponding changes into the manuscript. However, this comparison was difficult to do for all datasets, as we elaborate on below.
> > > >
> > > > The beam search (BS) of Shitole et al. (2021) finds attribution masks defined with respect to a grid of $r \times r$ image patches. In their paper and in their public implementation, they set  $r=7.$ We keep this default $7 \times 7$ grid setup in our experiments, as it is hardcoded certain places in the implementation, and modifying this value has the potential to introduce bugs.
> > > >
> > > > The difficulty of configuring the $r$ parameter makes running experiments on the shapes and BBBC041 experiments challenging. Both of these datasets have rectangular images (100x120 for shapes and 300x400 for BBBC041), and the networks for these experiments were trained without resizing augmentations. This means that resizing to the 224x224 image dimensions used by BS would result in off-manifold inputs for our networks. Furthermore, the resulting MSSs would be sufficient for these resized images, and not necessarily for the originals, which makes apples-to-apples comparison difficult. We therefore run BS on the CelebAMask-HQ experiment only. The results of this experiment are provided in Table 3, and the discussion in Section 6.7 has been updated.

---

> > > > > ### Comment · Reviewer_7V1r · 2026-04-20
> > > > >
> > > > > Yes, the available code for Shitole et al. has a great deal that is hardcoded and a lot of code that appears to be dead. It seems the public code was only ever intended to run the experiments presented in their paper, and nothing else. I'm of the opinion that it is not your job as authors to make other people's tools work well.

---

### Review · Reviewer_LSso · 2026-03-23

**Summary Of Contributions:**

This paper makes three main contributions to the XAI literature. It gives a unified explanation of why minimal sufficient subsets (MSS) are not unique. It covers several perspectives, including Boolean logic, context-specific conditional independence, nonlinearity, approximate sufficiency (ε > 0), and degenerate feature distributions. The authors introduces a meta-algorithm called LMEA. This method wraps any single-MSS explanation approach and produces multiple, disjoint MSSs. The paper validates LMEA on three image classification benchmarks. These include synthetic shapes, BBBC041 malaria detection, and CelebAMask-HQ earring detection. The results show competitive or better performance than MultiReX and PSIS across multiple metrics. The strengths are clear. The theory is good. The algorithm is simple and elegant. It is easy to apply in practice. The limitations are also discussed. That said, there are some weaknesses that I would like to mention. The experiments focus only on vision tasks with disjoint MSSs. The completeness guarantee depends on a strong oracle assumption. The effect of ε-MSS overlap is not explored much.

**Audience:**

Yes

**Audience Explanation:**

The paper is quite interesting and focuses on a topic that has been gaining a lot of attention recently.

**Claims And Evidence:**

Yes

**Claims Explanation:**

# Strengths

- The paper examines MSS multiplicity through five distinct lenses, building a good case that unique MSSs are the exception.
- LMEA is O(d) in oracle calls, trivially terminates (Lemma 1), and requires minimal change to existing MSO implementations.
- Sparsity-based postprocessing (SBP) reduces spurious explanations and its analysis is done well in the experimental results. From the experiments, it seems like SBP either improves or does not affect the performance.
- The notation is heavy but introduced carefully. Key ideas are explained well.

# Weaknesses

- Assumption 1 is very strong. Proposition 2 requires that the MSO returns an ε-MSS whenever one exists in the active pool. The authors acknowledge this but do not quantify how badly violations affect completeness in practice. A sensitivity analysis or empirical ablation on oracle quality would significantly strengthen the paper and provide a better understanding of the algorithm's robustness.
- The authors acknowledge in Section 7 that LMEA degrades when ε-MSSs overlap, but no experiments stress-test this failure mode. All three evaluation datasets were deliberately chosen because they have disjoint MSSs (as mentioned in the paper: "Finally, our experiments involve MIL datasets which we know a priori to have several non-overlapping MSSs, and these may not be reflective of other machine learning scenarios. That said, in other structured domains such as text, we also anticipate that LMEA will succeed in extending existing single-MSS methods, and these settings are left as a matter of future work."). This is a circular experimental design choice that limits generalizability.
- The authors introduce morphological dilation as a heuristic, however, there is no clear justification for choosing the radius. Also the idea of morphological dilation is not discussed that much in the paper, which makes it hard to understand the rationale behind the choice.

**Requested Changes:**

1. Add an ablation that shows how breaking Assumption 1 affects LMEA in practice. Right now, the gap between theory and experiments is not clear.

2. Include at least one setup/dataset where ε-MSSs overlap. This can be synthetic or real. It directly tests the failure case discussed earlier and helps assess how well LMEA generalizes beyond disjoint settings.

3. Explain the dilation step more clearly. Also provide details about hyperparameter tuning for hparams like the radius. These hyperparameters play an important role but are not well motivated, which makes reproduction harder.

---

> ### Author Response · Authors · 2026-04-07
> **Response**
>
> We thank the reviewer for their thoughtful feedback. We first address the questions the reviewer had about the manuscript. Then we explain the changes we made to the manuscript as requested by the reviewer.
>
> # Weaknesses
>
> ---
>
> ### **Assumption 1**
> We agree that we do not quantify how violations affect completeness in practice, and that a sensitivity analysis of oracle quality would strengthen the paper. We have therefore added an experiment (see "Changes to paper" below).
>
> ### **Overlap of MSSs**
> We also agree that an experiment evaluating LMEA when MSSs overlap would strengthen the paper. We have added a synthetic experiment that accomplishes this precisely (see "Changes to paper" below).
>
> ### **Dilation as a heuristic**
>
> Dilation is used because the masks for the RDE and XP MSOs are smoothed, which complicates the removal step of LMEA. For instance, if one were to clamp mask values in $\mathcal{A}^c$ (the complement of the active set of features) to zero in the optimization loop of either of these algorithms, one would obtain a $d$-dimensional mask parameter vector $\hat{\mathbf{s}}$ (see Eq. 4) that, once smoothed to yield $\mathbf{s}^{\star} = h(\hat{\mathbf{s}}),$ may include previously selected features (i.e., features in $\mathcal{A}^c$). To prevent such re-selection of features outside of the active set $\mathcal{A},$ it is necessary to adjust the removal step to compensate. Dilating explanations before removal achieves this compensation. A detailed discussion explaining the dilation step and how to tune the radius hyperparameter is included in **Appendix I** of the revised manuscript.
>
>
> # Changes to paper
>
> ---
>
> ### **Sensitivity Analysis of Assumption 1**
> In **Appendix D.2**, we have added a synthetic experiment that evaluates the impact of violating Assumption 1 on the performance of LMEA. We provide a short summary here.
>
> In the experiment, we let $\mathcal{G}$ denote the set of ground-truth MSSs for a model $f$ and instance $x$. To model the violation of Assumption 1, we simulate an MSO being used in LMEA such that it:
>
> a) Fails to find an MSS and terminates early with probability $\alpha$.
>
> b) Finds a spurious MSS (a non-ground truth MSS) with probability $\beta$.
>
> c) Returns a correct MSS with probability $1-\alpha-\beta$.
>
> Thus, $\beta$ directly quantifies the violation of Assumption 1, as a larger $\beta$ indicates a higher degree of violation since spurious MSSs are being selected more often. In **Figure 7**, we report the precision and recall of the sets returned by LMEA with respect to $\mathcal{G}$. As expected, we see both precision and recall gradually decrease as $\beta$ increases. For a complete description, with additional figures, see **Appendix D.2**.
>
> ### **Example of overlapping MSSs**
> In **Appendix D.1**, we have added a synthetic experiment studying the performance of LMEA when $\epsilon$-MSSs overlap. We provide a short summary here.
>
> In this experiment, we let $\mathcal{G}$ denote the set of ground-truth MSSs for a model $f$ and instance $x$. We model the MSSs in an image as a set of $n$ disks with common radius $r$. The number of intersections and their structure are controlled by $n$ and $r$, with the number of intersections increasing for larger $n$ and $r$. In **Figure 6**, we plot the fraction of MSSs recovered by LMEA as a function of $n$ and $r$.  As expected, as more MSSs intersect, the fraction of MSSs recovered by LMEA decreases. For a complete description, with additional figures, see **Appendix D.1**.
>
> ### **Explaining the dilation step**
> We have fully explained the reasoning behind the dilation step (including a detailed mathematical derivation) and provided practical guidance on setting dilation hyperparameters in **Appendix I**.

---

> > ### Comment · Reviewer_LSso · 2026-04-11
> >
> > Thank you for incorporating these revisions and additional experiments. I have reviewed the updated manuscript and have only a few minor concerns.
> >
> > Some minor comments:
> > - I recommend refining the subsection titles for the newly added appendix material to improve clarity and descriptiveness (e.g., instead of “D.2 Synthetic Experiment 3”,  consider more informative, content-specific titles).
> >
> > - I found only a brief mention of the sensitivity analysis for Assumption 1 in Appendix D.2. While the rebuttal provides additional details, this discussion does not appear to be fully integrated into the manuscript. Could you please incorporate a more complete description of this analysis into the paper? For reference, the current text states: “We see a gradual decrease of the set of explanations recovered by LMEA $\cdots$”

---

> > > ### Author Response · Authors · 2026-04-16
> > > **Response to minor comments**
> > >
> > > We thank the reviewer for pointing out these opportunities to further improve the paper. We have added more descriptive subsection titles to Section 6.5.1 and Appendices D.1-D.2. We have also added commentary further clarifying the synthetic oracle quality experiment setup and results to the final paragraph of Appendix D.2.

---

### Author Response · Authors · 2026-04-07
**Global response by authors**

We thank the reviewers for the thoughtful and helpful comments. We have incorporated the requested changes throughout the manuscript and uploaded a revised version. For convenience, all revisions are highlighted in blue.

---

> ### Author Response · Authors · 2026-04-16
> **Baseline improvements and further background parameter experiments**
>
> We thank the reviewers again for their many comments and questions that have raised the quality of the paper. To improve the work further, we ran additional experiments beyond those requested. For our PSIS baseline, which previously used a zero-valued background $\mathbf{b},$ we implemented the mean pixel background used in the [original](https://github.com/b-carter/SufficientInputSubsets) and [Google Research](https://github.com/google-research/google-research/tree/master/sufficient_input_subsets) SIS implementations, which tends to give better results for that baseline. These changes are reflected in Section 6.2, Figures 2-4 and 8-13, and Tables 1-3. To eliminate differing background values between methods as a source of variation, we also conducted an experiment with LMEA and all baselines configured to use the same background; see Appendix K for these results and the associated discussion. While the baselines often find a higher number of explanations than LMEA, the agreement of LMEA with ground truth regions as measured by IoU and IoE exceeds those of the baselines in most cases, which is consistent with the patterns observed in the original set of experiments.

---

### Decision · Action_Editor_RMQA · 2026-04-23

**Recommendation:** Accept with minor revision

**Additional Comments:**

The authors addressed most major concerns in their rebuttal and revision. All reviewers were therefore in favor of acceptance, and I support this consensus.

For the camera-ready version, I ask the authors to further revise the manuscript to address points that I feel could be better discussed. It is possible that the revision already addresses some of these points, in which case the authors could simply highlight how it does so.
- Most importantly, I think there should be more discussion on the "tension" that I mentioned above between the goal of MSSs, i.e., faithfulness to the explained model, and the metrics used that measure faithfulness/sufficiency in terms of ground truth or human labels.
- The two goals of the experiments (again as mentioned above) could be better clarified, perhaps with subsections or paragraphs of Section 6 identified with either the first or the second goal.
- It would be good to ensure that the claims regarding SBP are in line with how it was characterized in the rebuttal (optional, helpful in certain situations).

**Audience:**

Yes

**Audience Explanation:**

The criterion of interest was a strong point for the submission. Reviewers found the work to be quite interesting, in part due to the lack of attention to the issue of MSS multiplicity. The work also strongly motivates the problem by showing that MSS multiplicity is the usual situation and providing multiple perspectives on how it can arise.

**Claims And Evidence:**

Yes

**Claims Explanation:**

This submission puts forth the following sets of claims/contributions regarding minimal sufficient subsets (MSS), which are minimal subsets of features (feature-value assignments) that are sufficient to (approximately) account for a model's prediction.

**Explanation of the non-uniqueness of MSSs from various perspectives**: Reviewers thought that this contribution was very solid. Five different perspectives were clearly presented, each with intuition, examples, and mathematical justification.

**Let Me Explain Again (LMEA) meta-algorithm**:
- Reviewers appreciated LMEA's ability to lift single-MSS methods ("minimal sufficiency oracle," MSO) to find multiple MSSs with minimal changes and a linear number of calls to the MSO.
- Regarding the claim that LMEA can provably find all MSSs, Reviewer LSso felt that Assumption 1 (availability of an "ideal" MSO) is very strong and calls for a sensitivity analysis to determine the impact of not meeting this assumption in practice. The authors addressed this during the rebuttal with an experiment featuring MSOs that find spurious MSSs, showing natural degradation.
- Opinions regarding the sparsity-based post-processing (SBP) were mixed: Reviewer LSso appreciated its ability to reduce spurious explanations, while Reviewer 7V1r felt that it was ad hoc and could fail. The rebuttal clarified that SBP is optional and can be helpful in certain settings but not others.

**Empirical evaluation of LMEA**: Reviewers had the following concerns:
- Reviewer LSso commented that the chosen datasets do not sufficiently test LMEA in the setting of overlapping MSSs. The authors addressed this during the rebuttal with such an experiment, showing natural degradation.
- Reviewers 5gfh and 7V1r asked about comparisons with existing methods that find multiple MSSs: Byra & Skibbe (2025) and Shitole et al. (2021). Regarding Byra & Skibbe, the authors responded that no code is available, despite requests made to Byra & Skibbe. Regarding Shitole et al., the authors attempted the comparison but found that the hard-coding of an algorithm parameter made the comparison too difficult for some datasets, so they report only a partial comparison. The reviewers were satisfied with these responses.
- Additional requests by Reviewer 5gfh (larger models, higher input dimension, hyperparameter sensitivity) were responded to or addressed with experiments to the reviewer's satisfaction.
- Reviewer 7V1r had questions about the experimental goals and design. As far as I understand, these center around two issues:
  - There is some tension between MSSs on the one hand, which are oriented toward faithfulness to the model being explained, and some of the metrics used in the experiments, which measure faithfulness/sufficiency in terms of ground truth or human labels.
  - The experiments have two goals: 1) showing LMEA's ability to extend existing MSOs to output multiple MSSs and 2) evaluating LMEA's performance as a method for finding multiple MSSs, separate from how it achieves this.